# Efficient Reinforcement Learning for Global Decision Making in the Presence of Local Agents at Scale

## Abstract

We study reinforcement learning for global decision-making in the presence of local agents, where the global decision-maker makes decisions affecting all local agents, and the objective is to learn a policy that maximizes the joint rewards of all the agents. Such problems find many applications, e.g. demand response, EV charging, and queueing. In this setting, scalability has been a long-standing challenge due to the size of the joint state space which can be exponential in the number of agents. This work proposes the SUBSAMPLE-Q algorithm, where the global agent subsamples $k \leq n$ local agents to compute a policy in time that is polynomial in $k$. We show that this learned policy converges to the optimal policy on the order of $\tilde{O}(1/\sqrt{k} + \epsilon_{k,m})$ as the number of subsampled agents $k$ increases, where $\epsilon_{k,m}$ is the Bellman noise. Finally, we validate our theoretical results through numerical simulations in demand-response and queueing settings.

## 1 Introduction

Global decision-making, where a global agent makes decisions that affect a large number of local agents, is a classical problem that has been widely studied in many forms (Foster et al., 2022; Qin et al., 2023; Foster et al., 2023) and can be found in many applications, e.g. network optimization, power management, and electric vehicle (EV) charging (Kim & Giannakis, 2017; Zhang & Pavone, 2016; Molzahn et al., 2017). A critical challenge is the uncertain nature of the underlying system, which is often difficult to model precisely. Reinforcement Learning (RL) has demonstrated an impressive performance in a wide array of applications, such as the game of Go (Silver et al., 2016), autonomous driving (Kiran et al., 2022), and robotics (Kober et al., 2013). More recently, RL has emerged as a powerful tool for learning to control unknown systems (Ghai et al., 2023; Lin et al., 2023; 2024a;b), and thus holds significant potential for decision-making in multi-agent systems, including global decision making for local agents.

However, RL for multi-agent systems becomes intractable as the number of agents increases, due to the curse of dimensionality. For instance, classical RL algorithms, such as tabular $Q$-learning and temporal difference learning, require storing a $Q$-function (Bertsekas & Tsitsiklis, 1996; Powell, 2007) that scales with the size of the state-action space. Even if the individual agents' state space is small, the global state space can take values from a set of size exponentially large in the number of agents. When the system's rewards are not discounted, reinforcement learning for multi-agent systems is provably NP-hard (Qu et al., 2020a; Blondel & Tsitsiklis, 2000), and this scalability issue has been observed in a variety of settings Guestrin et al. (2003); Papadimitriou & Tsitsiklis (1999). A promising line of research over recent years focuses on networked instances, where interactions are restricted to local neighborhoods of agents (Lin et al., 2020; 2021; Qu et al., 2020b; Jing et al., 2022; Chu et al., 2020). This approach has led to scalable algorithms where each agent only considers the agents in its neighborhood to derive approximately optimal solutions. However, these results do not apply to our setting, where one global agent interacts with many local agents. This can be viewed as a star graph, where the neighborhood of the central decision-making agent is large.

Beyond the networked formulation, another exciting line of work addressing this intractability is mean-field RL (Yang et al., 2018). Mean-field RL assumes that all agents are homogeneous in their state and action spaces, enabling interactions to be approximated by a representative "mean"

agent. This significantly reduces the complexity of $Q$-learning to a polynomial dependence on the number of agents, and learns an approximately optimal policy where the approximation error decays with the number of agents (Gu et al., 2021; 2022a). However, mean-field RL does not directly transfer to our setting since the global decision-making agent violates the homogeneity assumption. Moreover, when the number of local agents is large, storing a polynomially-large $Q$-table (where the polynomial's degree depends on size of the state space for a single agent) can still be infeasible. This motivates the following fundamental question: *can we design a fast and competitive policy-learning algorithm for a global decision-making agent in a system with many local agents?*

**Contributions.** We answer this question affirmatively. Our key contributions are outlined below.

- **Subsampling Algorithm.** We propose SUBSAMPLE-Q, an algorithm designed to address the challenge of global decision-making in systems with a large number of local agents. We model the problem as a Markov Decision Process with a global decision-making agent and $n$ local agents. SUBSAMPLE-Q (Algorithms 1 to 3) begins by selecting $k \leq n$ local agents to learn a deterministic policy $\hat{\pi}^{\text{est}}_{k,m}$, where $m$ is the number of samples used to update the estimates of the $Q$-function, by applying value iteration and mean-field value iteration on the $k$ local agents to learn $\hat{Q}^{\text{est}}_{k,m}$, which can be viewed as a smaller $Q$ function. It then deploys a stochastic policy $\hat{\pi}_{k,m}$ that uniformly samples $k$ local agents at each step and uses $\hat{\pi}_{k,m}$ to determine an action for the global agent.

- **Sample Complexity and Theoretical Guarantee.** As the number of local agents increases, the size of $\hat{Q}_{k,m}$ scales polynomially with $k$, rather than polynomially with $n$ as in mean-field RL. When the size of the local agent's state space grows, the size of $\hat{Q}_{k,m}$ scales exponentially with $k$, instead of exponentially with $n$ as in traditional $Q$-learning). Theorem 3.4 demonstrates that the performance gap between $\pi^{\text{est}}_{k,m}$ and the optimal policy $\pi^*$ is $O(1/\sqrt{k} + \epsilon_{k,m})$, where $\epsilon_{k,m}$ represents the Bellman noise in $\hat{Q}^{\text{est}}_{k,m}$. The choice of $k$ reveals a fundamental trade-off between the size of the $Q$-table and the optimality of $\pi^{\text{est}}_{k,m}$. As $n$ scales, setting $k = O(\log n)$ achieves a runtime that is polylogarithmic in $n$, representing an exponential speedup over the previously best-known polytime mean-field RL methods, while maintaining a decaying optimality gap.

- **Numerical Simulations.** We evaluate the effectiveness of SUBSAMPLE-Q in two scenarios: a power system demand-response problem (Example 5.1) and a queueing problem (Example 5.2). A key inspiration for our approach is the *power-of-two-choices* from queueing theory (Mitzenmacher & Sinclair, 1996), where a dispatcher subsamples two queues to make decisions. Our work generalizes this principle to a broader decision-making problem.

While our results are theoretical in nature, it is our hope that SUBSAMPLE-Q will lead to further exploration into the potential of subsampling in Markov games and networked multi-agent reinforcement learning, and inspire the development of practical algorithms for multi-agent settings.

## 2 PRELIMINARIES

**Notation.** For $k, n \in \mathbb{N}$ where $k \leq n$, let $\binom{[n]}{k}$ denote the set of $k$-sized subsets of $[n] = \{1, \dots, n\}$. For any vector $z \in \mathbb{R}^d$, let $\|z\|_1$ and $\|z\|_\infty$ denote the standard $\ell_1$ and $\ell_\infty$ norms of $z$ respectively. Let $\|\mathbf{A}\|_1$ denote the matrix $\ell_1$-norm of $\mathbf{A} \in \mathbb{R}^{n \times m}$. Given a collection of variables $s_1, \dots, s_n$ the shorthand $s_\Delta$ denotes the set $\{s_i : i \in \Delta\}$ for $\Delta \subseteq [n]$. We use $\tilde{O}(\cdot)$ to suppress polylogarithmic factors in all problem parameters except $n$. For a discrete measurable space $(\mathcal{X}, \mathcal{F})$, the total variation distance between probability measures $\rho_1, \rho_2$ is given by $\text{TV}(\rho_1, \rho_2) = \frac{1}{2} \sum_{x \in \mathcal{X}} |\rho_1(x) - \rho_2(x)|$.

**Problem Statement.** We consider a system of $n + 1$ agents given by $\mathcal{N} = \{0\} \cup [n]$. Let agent 0 be the "global agent" decision-maker, and agents $[n]$ be the "local" agents. In this model, each agent $i \in [n]$ is associated with a state $s_i \in \mathcal{S}_l$, where $\mathcal{S}_l$ is the local agent's state space. The global agent is associated with a state $s_g \in \mathcal{S}_g$ and action $a_g \in \mathcal{A}_g$, where $\mathcal{S}_g$ is the global agent's state space and $\mathcal{A}_g$ is the global agent's action space. The global state of all agents is given by $(s_g, s_1, \dots, s_n) \in \mathcal{S} := \mathcal{S}_g \times \mathcal{S}_l^n$. At each time-step $t$, the next state for all the agents is independently generated by stochastic transition kernels $P_g : \mathcal{S}_g \times \mathcal{S}_g \times \mathcal{A}_g \to [0, 1]$ and $P_l : \mathcal{S}_l \times \mathcal{S}_l \times \mathcal{S}_g \to [0, 1]$ as follows:

$$s_g(t+1) \sim P_g(\cdot|s_g(t), a_g(t)), \tag{1}$$

$$s_i(t+1) \sim P_l(\cdot|s_i(t), s_g(t)), \forall i \in [n] \tag{2}$$

The global agent selects $a_g(t) \in \mathcal{A}_g$. Next, the agents receive a structured reward $r : \mathcal{S} \times \mathcal{A}_g \to \mathbb{R}$, given by Equation (3), where the choice of functions $r_g$ and $r_l$ is flexible and application-specific.

$$r(s, a_g) = \underbrace{r_g(s_g, a_g)}_{\text{global component}} + \frac{1}{n} \sum_{i \in [n]} \underbrace{r_l(s_i, s_g)}_{\text{local component}} \tag{3}$$

We define a policy $\pi : \mathcal{S} \to \mathcal{P}(\mathcal{A}_g)$ as a map from states to distributions of actions such that $a_g \sim \pi(\cdot|s)$. When a policy is executed, it generates a trajectory $(s^0, a_g^0, r^0), \ldots, (s^T, a_g^T, r^T)$ via the process $a_g^t \sim \pi(s^t), s^{t+1} \sim (P_g, P_l)(s^t, a_g^t)$, initialized at $s^0 \sim d_0$. We write $\mathbb{P}^\pi[\cdot]$ and $\mathbb{E}^\pi[\cdot]$ to denote the law and corresponding expectation for the trajectory under this process. The goal of the problem is to then learn a policy $\pi$ that maximizes the *value function* $V : \pi \times \mathcal{S} \to \mathbb{R}$, the expected discounted reward for each $s \in \mathcal{S}$ given by

$$V^\pi(s) = \mathbb{E}^\pi \left[ \sum_{t=0}^\infty \gamma^t r(s(t), a_g(t)) | s(0) = s \right], \tag{4}$$

where $\gamma \in (0, 1)$ is a discounting factor. We define $\pi^*$ as the optimal deterministic policy, which maximizes $V^\pi(s)$ at all states. This model characterizes a crucial decision-making process in the presence of multiple agents where the information from all local agents is concentrated towards the decision maker, the global agent. The objective of the problem is to learn an approximately optimal policy that jointly minimizes the sample and computational complexities of learning the policy.

We make the following standard assumptions:

**Assumption 2.1** (Finite state/action spaces). We assume that the state spaces of all the agents and the action space of the global agent are finite: $|\mathcal{S}_l|, |\mathcal{S}_g|, |\mathcal{A}_g| < \infty$.

**Assumption 2.2** (Bounded rewards). The global and local components of the reward function are bounded. Specifically, $\|r_g(\cdot, \cdot)\|_\infty \leq \tilde{r}_g$, and $\|r_l(\cdot, \cdot)\|_\infty \leq \tilde{r}_l$. Then, $\|r(\cdot, \cdot)\|_\infty \leq \tilde{r}_g + \tilde{r}_l := \tilde{r}$.

**Definition 2.1** ($\epsilon$-optimal policy). Given a policy simplex $\Pi$, a policy $\pi \in \Pi$ is $\epsilon$-optimal if for all $s \in \mathcal{S}, V^\pi(s) \geq \sup_{\pi^* \in \Pi} V^{\pi^*}(s) - \epsilon$.

**Remark 2.2.** Heterogeneity among the local agents can be captured by modeling agent types as part of the agent state. Specifically, assign a type to each local agent by letting $\mathcal{S}_l = \mathcal{E} \times \bar{\mathcal{S}}_l$, where $\mathcal{E}$ represents a set of different possible agent types, which are treated as part of the agent's state. This type remains fixed throughout the transitions, allowing the transition and reward functions to vary depending on the agent's type, and enabling the global agent to uniquely signal agents of each type.

**Related Work.** This paper relates to two major lines of work which we describe below.

*Multi-agent RL (MARL).* MARL has a rich history, starting with early works on Markov games used to characterize the decision-making process (Shapley, 1953; Littman, 1994), which can be regarded as a multi-agent extension of the Markov Decision Process (MDP). MARL has since been actively studied (Zhang et al., 2021) in a broad range of settings, such as cooperative and competitive agents. MARL is most similar to the category of "succinctly described" MDPs (Blondel & Tsitsiklis, 2000), where the state/action space is a product space formed by the individual state/action spaces of multiple agents, and where the agents interact to maximize an objective function. Our work, which can be viewed as an essential stepping stone to MARL, also shares the curse of dimensionality.

A line of celebrated works (Qu et al., 2020b; Chu et al., 2020; Lin et al., 2020; 2021; Jing et al., 2022) constrain the problem to networked instances to enforce local agent interactions and find policies that maximize the objective function, which is the expected cumulative discounted reward. By exploiting Gamarnik's spatial exponential decay property from combinatorial optimization (Gamarnik et al., 2009), they overcome the curse of dimensionality by truncating the problem to only search over the policy space derived from the local neighborhood of agents that are at most $\kappa$ away from each other to find an $O(\rho^{k+1})$-approximation of the maximized objective function for $\rho \in (0, 1)$. However, since their algorithms have a complexity that is exponential in the size of the neighborhood, they are only tractable for sparse graphs. Therefore, these algorithms do not apply to our decision-making problem, which can be viewed as a dense star graph (see Appendix A). The recently popular work on V-learning (Jin et al., 2021) reduces the dependence of the product action space to an additive dependence. However, since our work focuses on the action of the global decision-maker,

the complexity in the action space is already minimal. Instead, our work focuses on reducing the complexity of the joint state space which has not been previously accomplished for dense networks.

*Mean-Field RL.* Under assumptions of homogeneity in the state/action spaces of the agents, the problem of densely networked multi-agent RL was partially resolved in Yang et al. (2018); Gu et al. (2021; 2022a;b); Subramanian et al. (2022) which approximates the learning problem through mean-field control, where the approximation error scales as $O(1/\sqrt{n})$. To overcome the problem of designing algorithms on probability measure spaces, they study MARL under Pareto optimality and use the (functional) strong law of large numbers to consider a lifted state/action space with a representative agent, where the rewards and dynamics of the system are aggregated. Cui & Koeppl (2022); Hu et al. (2023); Carmona et al. (2023) introduce heterogeneity to the mean-field approach using graphon mean-field games; however, there is a loss of topological information when using graphons to approximate finite graphs, as graphons correspond to infinitely large adjacency matrices. Additionally, graphon mean-field RL imposes a critical assumption of the existence of graphon sequences that converge in cut-norm to the problem instance. Another mean-field RL approach that partially introduces heterogeneity is in a line of work considering major and minor agents. This has been well studied in the competitive setting (Carmona & Zhu, 2016; Carmona & Wang, 2016). In the cooperative setting, Mondal et al. (2022); Cui et al. (2023) are most related to our work, as they collectively consider a setting with $k$ classes of homogeneous agents, but their mean-field analytic approaches do not converge to the optimal policy upon introducing a global decision-making agent. Furthermore, these works require Lipschitz continuity assumptions on the reward functions which we relax in our work. Finally, the algorithms underlying mean-field RL have a runtime that is polynomial in $n$, whereas our `SUBSAMPLE-Q` algorithm has a runtime that is polylogarithmic in $n$.

*Other Related Works.* A line of works has similarly exploited the star-shaped network in cooperative multi-agent systems. Min et al. (2023); Chaudhari et al. (2024) studied the communication complexity and mixing times of various learning settings with purely homogeneous agents, and Do et al. (2023) studied the setting of heterogeneous linear contextual bandits to yield a no-regret guarantee. We extend this work to the more challenging setting of reinforcement learning.

**Q-learning.** To provide background for the analysis in this paper, we review a few key technical concepts in RL. At the core of the standard Q-learning framework (Watkins & Dayan, 1992) for offline-RL is the $Q$-function $Q: \mathcal{S} \times \mathcal{A}_g \to \mathbb{R}$. $Q$-learning seeks to produce a policy $\pi^*(\cdot|s)$ that maximizes the expected infinite horizon discounted reward. For any policy $\pi$, $Q^\pi(s, a_g) = \mathbb{E}^\pi[\sum_{t=0}^\infty \gamma^t r(s(t), a_g(t))|s(0) = s, a_g(0) = a]$. One approach to learning the optimal policy $\pi^*(\cdot|s)$ is dynamic programming, where the $Q$-function is iteratively updated using value-iteration: $Q^0(s, a_g) = 0$, for all $(s, a_g) \in \mathcal{S} \times \mathcal{A}_g$. Then, for all $t \in [T]$, $Q^{t+1}(s, a) = \mathcal{T}Q^t(s, a_g)$, where $\mathcal{T}$ is the Bellman operator defined as

$$\mathcal{T}Q^t(s, a_g) = r(s, a_g) + \gamma \mathbb{E}_{s'_g \sim P_g(\cdot|s_g, a_g), s'_i \sim P_l(\cdot|s_i, s_g), \forall i \in [n]} \max_{a' \in \mathcal{A}_g} Q^t(s', a'_g). \tag{5}$$

The Bellman operator $\mathcal{T}$ satisfies a $\gamma$-contractive property, implying the existence of a unique fixed-point $Q^*$ such that $\mathcal{T}Q^* = Q^*$, by the Banach-Caccioppoli fixed-point theorem (Banach, 1922). Here, the optimal policy is the deterministic greedy policy $\pi^*: \mathcal{S}_g \times \mathcal{S}_l^n \to \mathcal{A}_g$, where $\pi^*(s) = \arg\max_{a_g \in \mathcal{A}_g} Q^*(s, a_g)$. However, the complexity of a single update to the $Q$-function is $O(|\mathcal{S}_g||\mathcal{S}_l|^n|\mathcal{A}_g|)$, which grows exponentially with $n$. As the number of local agents increases ($n \gg |\mathcal{S}_l|$), this exponential update complexity renders $Q$-learning impractical (see Example 5.2).

**Mean-field Transformation.** To address this, Yang et al. (2018) developed a mean-field approach which, under homogeneity assumptions, considers the distribution function $F_{s_{[n]}}: \mathcal{S}_l \to \mathbb{R}$ given by

$$F_{s_{[n]}}(x) := \frac{1}{n} \sum_{i=1}^n \mathbf{1}\{s_i = x\}, \quad \forall x \in \mathcal{S}_l. \tag{6}$$

Let $\mu_n(\mathcal{S}_l) = \{\frac{b}{n}|b \in \{0, \ldots, n\}\}^{|\mathcal{S}_l|}$ be the space of $|\mathcal{S}_l|$-length vectors where each entry is an element of $\{0, \frac{1}{n}, \frac{2}{n}, \ldots, 1\}$. In this space, $F_{s_{[n]}} \in \mu_n(\mathcal{S}_l)$ where $F_{s_{[n]}}$ represents the proportion of agents in each state. The $Q$-function is permutation-invariant in the local agents as they are homogeneous, and permuting the labels of local agents with the same state will not change the global agent's decision. Thus, the $Q$-function only depends on the states $s_{[n]}$ through the distribution function $F_{s_{[n]}}$:

$$Q(s_g, s_{[n]}, a_g) = \hat{Q}(s_g, F_{s_{[n]}}, a_g). \tag{7}$$

Here, $\hat{Q} : \mathcal{S}_g \times \mu_n(\mathcal{S}_l) \times \mathcal{A}_g \to \mathbb{R}$ is a reparameterized $Q$-function learned by mean-field value iteration. We initialize $\hat{Q}^0(s_g, F_{s_{[n]}}, a_g) = 0, \forall(s, a_g) \in \mathcal{S}_g \times \mathcal{A}_g$. For all $t$, we update $\hat{Q}$ as $\hat{Q}^{t+1}(s, F_{s_{[n]}}, a_g) = \hat{\mathcal{T}}\hat{Q}^t(s_g, F_{s_{[n]}}, a_g)$, where $\hat{\mathcal{T}}$ is the Bellman operator in distribution space:

$$\hat{\mathcal{T}}\hat{Q}^t(s_g, F_{s_{[n]}}, a_g) = r(s, a_g) + \gamma \mathbb{E}_{\substack{s'_g \sim P_g(\cdot|s_g,a_g), \\ s'_i \sim P_l(\cdot|s_i,s_g), \forall i \in [n]}} \max_{a'_g \in \mathcal{A}_g} \hat{Q}^t(s', F'_{s_{[n]}}, a'_g). \qquad (8)$$

$\hat{\mathcal{T}}$ is $\gamma$-contractive; hence, it has a unique fixed-point $\hat{Q}^*$ where $\hat{Q}^*(s_g, F_{s_{[n]}}, a_g) = Q^*(s_g, s_{[n]}, a_g)$, and the deterministic optimal (greedy) policy $\hat{\pi}^*$ is $\hat{\pi}^*(s_g, F_{s_{[n]}}) = \arg\max_{a_g \in \mathcal{A}_g} \hat{Q}^*(s_g, F_{s_{[n]}}, a_g)$. The update complexity to the $\hat{Q}$-function is $O(|\mathcal{S}_g||\mathcal{A}_g||\mathcal{S}_l|n^{|\mathcal{S}_l|})$, which scales polynomially in $n$.

**Remark 2.3.** The solution offered by mean-field value iteration and standard $Q$-learning requires a sample complexity of $\min\{\tilde{O}(|\mathcal{S}_g||\mathcal{A}_g||\mathcal{S}_l|^n), \tilde{O}(|\mathcal{S}_g||\mathcal{A}_g||\mathcal{S}_l|n^{|\mathcal{S}_l|})\}$, where one uses $Q$-learning if $|\mathcal{S}_l|^{n-1} < n^{|\mathcal{S}_l|}$, and mean-field value iteration otherwise. In each of these regimes, as $n$ scales, the update complexity can become incredibly computationally intensive. Therefore, we introduce the SUBSAMPLE-Q algorithm in Section 3 to mitigate the cost of scaling the number of local agents.

## 3 METHOD AND THEORETICAL RESULTS

### 3.1 PROPOSED METHOD: SUBSAMPLE-Q

In this work, we propose the SUBSAMPLE-Q algorithm to overcome the polynomial (in $n$) sample complexity of mean-field value iteration and the exponential (in $n$) sample complexity of traditional $Q$-learning. In our algorithm, the global agent randomly samples a subset of local agents $\Delta \subseteq [n]$ such that $|\Delta| = k$, for $k \leq n$. It ignores all other local agents $[n] \setminus \Delta$, and performs value iteration to learn the $Q$-function $\hat{Q}^*_k$ and policy $\hat{\pi}^*_{k,m}$ for this surrogate subsystem of $k$ local agents, where $m$ is the sample size in each iteration. When $|\mathcal{S}_l|^{k-1} < k^{|\mathcal{S}_l|}$, the algorithm uses traditional value-iteration, and when $|\mathcal{S}_l|^{k-1} > k^{|\mathcal{S}_l|}$, it switches to mean-field value iteration. The surrogate reward gained by the system at each time step is $r_\Delta : \mathcal{S} \times \mathcal{A}_g \to \mathbb{R}$, given by Equation (9):

$$r_\Delta(s, a_g) = r_g(s_g, a_g) + \frac{1}{|\Delta|} \sum_{i \in \Delta} r_l(s_g, s_i). \qquad (9)$$

To convert the optimality of the global agent's action on the $k$ local-agent subsystem to an approximate optimality on the full $n$-agent system, we use a randomized policy $\pi^{\text{est}}_{k,m}$ which samples $\Delta \in \mathcal{U}\binom{[n]}{k}$ at each time-step to derive the action $a_g \leftarrow \hat{\pi}^{\text{est}}_{k,m}(s_g, s_\Delta)$. Finally, Theorem 3.4 shows that the policy $\pi^{\text{est}}_{k,m}$ converges to the optimal policy $\pi^*$ as $k \to n$ and $m \to \infty$.

We present Algorithms 1 and 2 (SUBSAMPLE-Q: Learning) and Algorithm 3 (SUBSAMPLE-Q: Execution), which we describe below. We first characterize the notion of the empirical distribution:

**Definition 3.1** (Empirical Distribution Function). For any population $(s_1, \ldots, s_n) \in \mathcal{S}_l^n$, define the empirical distribution function $F_{s_\Delta} : \mathcal{S}_l \to \mathbb{R}$ for $\Delta \subseteq [n]$ such that $|\Delta| = k$ by:

$$F_{s_\Delta}(x) := \frac{1}{|\Delta|} \sum_{i \in \Delta} \mathbf{1}\{s_i = x\}. \qquad (10)$$

Let $\mu_k(\mathcal{S}_l) := \left\{ \frac{b}{k} | b \in \{0, \ldots, k\} \right\}^{|\mathcal{S}_l|}$ be the space of $|\mathcal{S}_l|$-length vectors where each entry in a vector is an element of $\{0, \frac{1}{k}, \frac{2}{k}, \ldots, 1\}$ such that $F_{s_\Delta} \in \mu_k(\mathcal{S}_l)$. Here, $F_{s_\Delta}$ is the proportion of agents in the $k$-local-agent subsystem at each state.

**Algorithms 1 and 2** (Offline learning). Let $m \in \mathbb{N}$ denote the sample size for the learning algorithm with sampling parameter $k \leq n$. When $|\mathcal{S}_l|^{k-1} \leq k^{|\mathcal{S}_l|}$, we empirically learn the optimal $Q$-function for a subsystem with $k$-local agents denoted by $\hat{Q}^{\text{est}}_{k,m} : \mathcal{S}_g \times \mathcal{S}_l^k \times \mathcal{A}_g \to \mathbb{R}$: set $\hat{Q}^0_{k,m}(s_g, s_\Delta, a_g) = 0$ for all $(s_g, s_\Delta, a_g) \in \mathcal{S}_g \times \mathcal{S}_l^k \times \mathcal{A}_g$. At time step $t$, set $\hat{Q}^{t+1}_{k,m}(s_g, s_\Delta, a_g) = \tilde{\mathcal{T}}_{k,m}\hat{Q}^t_{k,m}(s_g, s_\Delta, a_g)$, where $\tilde{\mathcal{T}}_{k,m}$ is the *empirically adapted Bellman operator* in Equation (11).

---

**Algorithm 1** SUB-SAMPLE-Q: Learning (if $|\mathcal{S}_l|^{k-1} \leq k^{|\mathcal{S}_l|}$)

---

**Require:** A multi-agent system as described in Section 2. Parameter $T$ for the number of iterations in the initial value iteration step. Sampling parameters $k \in [n]$ and $m \in \mathbb{N}$. Discount parameter $\gamma \in (0, 1)$. Oracle $\mathcal{O}$ to sample $s_g' \sim P_g(\cdot|s_g, a_g)$ and $s_i' \sim P_l(\cdot|s_i, s_g, a_i)$ for all $i \in [n]$.

1: Uniformly sample $\Delta \subseteq [n]$ such that $|\Delta| = k$.
2: Initialize $\hat{Q}_{k,m}^0(s_g, s_\Delta, a_g) = 0$ for $(s_g, s_\Delta, a_g) \in \mathcal{S}_g \times \mathcal{S}_l^k \times \mathcal{A}_g$.
3: **for** $t = 1$ to $T$ **do**
4:    **for** $(s_g, s_\Delta, a_g) \in \mathcal{S}_g \times \mathcal{S}_l^k \times \mathcal{A}_g$ **do**
5:        $\hat{Q}_{k,m}^{t+1}(s_g, s_\Delta, a_g) = \tilde{\mathcal{T}}_{k,m}\hat{Q}_{k,m}^t(s_g, s_\Delta, a_g)$
6: Return $\hat{Q}_{k,m}^T$. For all $s_g, s_\Delta \in \mathcal{S}_g \times \mathcal{S}_l^k$, let $\hat{\pi}_{k,m}^{\text{est}}(s_g, s_\Delta) = \arg\max_{a_g \in \mathcal{A}_g} \hat{Q}_{k,m}^T(s_g, s_\Delta, a_g)$.

---

When $|\mathcal{S}_l|^{k-1} > k^{|\mathcal{S}_l|}$, we empirically learn the optimal mean-field $Q$-function for a $k$ local agent system, denoted (with abuse of notation) by $\hat{Q}_{k,m}^{\text{est}} : \mathcal{S}_g \times \mu_k(\mathcal{S}_l) \times \mathcal{A}_g \to \mathbb{R}$. For $(s_g, F_{s_\Delta}, a_g) \in \mathcal{S}_g \times \mu_k(\mathcal{S}_l) \times \mathcal{A}_g$, set $\hat{Q}_{k,m}^0(s_g, F_{s_\Delta}, a_g) = 0$. At time $t$, set $\hat{Q}_{k,m}^{t+1}(s_g, F_{s_\Delta}, a_g) = \hat{\mathcal{T}}_{k,m}\hat{Q}_{k,m}^t(s_g, F_{s_\Delta}, a_g)$, where $\hat{\mathcal{T}}_{k,m}$ is the *empirically adapted mean-field Bellman operator* in Equation (12).

$\mathcal{T}_{k,m}$ and $\hat{\mathcal{T}}_{k,m}$ draws $m$ random samples $s_g^j \sim P_g(\cdot|s_g, a_g)$ and $s_i^j \sim P_l(\cdot|s_i, s_g)$ for $j \in [m]$, $i \in \Delta$:

$$\tilde{\mathcal{T}}_{k,m}\hat{Q}_{k,m}^t(s_g, s_\Delta, a_g) = r_\Delta(s, a_g) + \frac{\gamma}{m} \sum_{j \in [m]} \max_{a_g' \in \mathcal{A}_g} \hat{Q}_{k,m}^t(s_g^j, s_\Delta^j, a_g'). \tag{11}$$

$$\hat{\mathcal{T}}_{k,m}\hat{Q}_{k,m}^t(s_g, F_{s_\Delta}, a_g) = r_\Delta(s, a_g) + \frac{\gamma}{m} \sum_{j \in [m]} \max_{a_g' \in \mathcal{A}_g} \hat{Q}_{k,m}^t(s_g^j, F_{s_\Delta^j}, a_g'). \tag{12}$$

As in Equation (7), $\hat{Q}_{k,m}^t$ only depends on $s_\Delta$ through $F_{s_\Delta}$:

$$\hat{Q}_{k,m}^t(s_g, s_\Delta, a_g) = \hat{Q}_{k,m}^t(s_g, F_{s_\Delta}, a_g). \tag{13}$$

$\tilde{\mathcal{T}}_{k,m}$ and $\hat{\mathcal{T}}_{k,m}$ are $\gamma$-contractive by Lemma A.10. Algorithms 1 and 2 apply value iteration with their Bellman operator until $\hat{Q}_{k,m}$ converges to a fixed point $\hat{Q}_{k,m}^{\text{est}}$ satisfying $\tilde{\mathcal{T}}_{k,m}\hat{Q}_{k,m}^{\text{est}} = \hat{Q}_{k,m}^{\text{est}}$ and $\hat{\mathcal{T}}_{k,m}\hat{Q}_{k,m}^{\text{est}} = \hat{Q}_{k,m}^{\text{est}}$, giving equivalent deterministic policies $\hat{\pi}_{k,m}^{\text{est}}(s_g, s_\Delta) = \arg\max_{a_g \in \mathcal{A}_g} \hat{Q}_{k,m}^{\text{est}}(s_g, s_\Delta, a_g)$ and $\hat{\pi}_{k,m}^{\text{est}}(s_g, F_{s_\Delta}) = \arg\max_{a_g \in \mathcal{A}_g} \hat{Q}_{k,m}^{\text{est}}(s_g, F_{s_\Delta}, a_g)$.

**Algorithm 3** (Online implementation). Here, Algorithm 3 (SUBSAMPLE-Q: Execution) randomly samples $\Delta \sim \mathcal{U}\binom{[n]}{k}$ at each time step and uses action $a_g \sim \hat{\pi}_{k,m}^{\text{est}}(s_g, F_{s_\Delta})$ to get reward $r(s, a_g)$. This procedure of first sampling $\Delta$ and then applying $\hat{\pi}_{k,m}^{\text{est}}$ is denoted by a stochastic policy $\pi_{k,m}^{\text{est}}(a_g|s)$:

$$\pi_{k,m}^{\text{est}}(a_g|s) = \frac{1}{\binom{n}{k}} \sum_{\Delta \in \binom{[n]}{k}} \mathbf{1}(\hat{\pi}_{k,m}^{\text{est}}(s_g, F_{s_\Delta}) = a_g). \tag{14}$$

Then, each agent transitions to their next state based on Equation (1).

**Remark 3.2.** Algorithm 2 assumes the existence of a generative model $\mathcal{O}$ (Kearns & Singh, 1998) to sample $s_g' \sim P_g(\cdot|s_g, a_g)$ and $s_i \sim P_l(\cdot|s_i, s_g)$. This may generalize to the online RL setting using cold-start and no-regret techniques from (Jin et al., 2018), which we leave for future investigations.

### 3.2 Theoretical Guarantee

This subsection shows that the value of the expected discounted cumulative reward produced by $\pi_{k,m}^{\text{est}}$ is approximately optimal, where the optimality gap decays as $k \to n$ and $m \to \infty$.

**Bellman noise.** We introduce the notion of Bellman noise, which is used in the main theorem. Consider $\hat{\mathcal{T}}_{k,m}$. Clearly, it is an unbiased estimator of the generalized adapted Bellman operator $\hat{\mathcal{T}}_k$,

$$\hat{\mathcal{T}}_k\hat{Q}_k(s_g, F_{s_\Delta}, a_g) = r_\Delta(s, a_g) + \gamma \mathbb{E}_{s_g' \sim P_g(\cdot|s_g, a_g), s_i' \sim P_l(\cdot|s_i, s_g), \forall i \in \Delta} \max_{a_g' \in \mathcal{A}_g} \hat{Q}_k(s_g', F_{s_\Delta'}, a_g'). \tag{15}$$

---

**Algorithm 2** SUBSAMPLE-Q: Learning (if $k^{|\mathcal{S}_l|} < |\mathcal{S}_l|^k$)

---

**Require:** A multi-agent system as described in Section 2. Parameter $T$ for the number of iterations in the initial value iteration step. Sampling parameters $k \in [n]$ and $m \in \mathbb{N}$. Discount parameter $\gamma \in (0, 1)$. Oracle $\mathcal{O}$ to sample $s_g' \sim P_g(\cdot|s_g, a_g)$ and $s_i \sim P_l(\cdot|s_i, s_g)$ for all $i \in [n]$.

1: Uniformly choose $\Delta \subseteq [n]$ such that $|\Delta| = k$.
2: Set $\hat{Q}^0_{k,m}(s_g, F_{s_\Delta}, a_g) = 0$, for $(s_g, F_{s_\Delta}, a_g) \in \mathcal{S}_g \times \mu_k(\mathcal{S}_l) \times \mathcal{A}_g$
3: **for** $t = 1$ to $T$ **do**
4:     **for** $(s_g, F_{s_\Delta}, a_g) \in \mathcal{S}_g \times \mu_k(\mathcal{S}_l) \times \mathcal{A}_g$ **do**
5:         $\hat{Q}^{t+1}_{k,m}(s_g, F_{s_\Delta}, a_g) = \hat{\mathcal{T}}_{k,m} \hat{Q}^t_{k,m}(s_g, F_{s_\Delta}, a_g)$
6: Return $\hat{Q}^T_{k,m}$. $\forall (s_g, F_{s_\Delta}) \in \mathcal{S}_g \times \mu_k(\mathcal{S}_l)$, let $\hat{\pi}^{\text{est}}_{k,m}(s_g, F_{s_\Delta}) = \arg\max_{a_g \in \mathcal{A}_g} \hat{Q}^T_{k,m}(s_g, F_{s_\Delta}, a_g)$.

---

**Algorithm 3** SUBSAMPLE-Q: Execution

---

**Require:** A multi-agent system as described in Section 2. Parameter $T'$ for the number of rounds in the game. Hyperparameter $k \in [n]$. Discount parameter $\gamma$. Policy $\hat{\pi}^{\text{est}}_{k,m}(s_g, F_{s_\Delta})$.

1: If $|\mathcal{S}_l|^{k-1} \geq k^{|\mathcal{S}_l|}$, learn $\hat{\pi}^{\text{est}}_{k,m}$ from Algorithm 1.
2: If $|\mathcal{S}_l|^{k-1} < k^{|\mathcal{S}_l|}$, learn $\hat{\pi}^{\text{est}}_{k,m}$ from Algorithm 2.
3: Initialize $(s_g(0), s_{[n]}(0)) \sim s_0$, where $s_0$ is a distribution on the initial global state $(s_g, s_{[n]})$,
4: Initialize the total reward: $R_0 = 0$.
5: **Policy** $\pi^{\text{est}}_{k,m}(s)$ is defined as follows:
6: **for** $t = 0$ to $T'$ **do**
7:     Sample $\Delta$ uniformly at random from from $\binom{[n]}{k}$.
8:     Let $a_g(t) = \hat{\pi}^{\text{est}}_{k,m}(s_g(t), F_{s_{\Delta}(t)})$.
9:     Let $s_g(t+1) \sim P_g(\cdot|s_g(t), a_g(t))$ and $s_i(t+1) \sim P_l(\cdot|s_i(t), s_g(t))$, for all $i \in [n]$.
10:     $R_{t+1} = R_t + \gamma^t \cdot r(s, a_g)$

---

For all $(s_g, F_{s_\Delta}, a_g) \in \mathcal{S}_g \times \mu_k(\mathcal{S}_l) \times \mathcal{A}_g$, set $\hat{Q}^0_k(s_g, F_{s_\Delta}, a_g) = 0$. For $t \in \mathbb{N}$, let $\hat{Q}^{t+1}_k = \hat{\mathcal{T}}_k \hat{Q}^t_k$, where $\hat{\mathcal{T}}_k$ is defined for $k \leq n$ in Equation (15). Then, $\hat{\mathcal{T}}_k$ is also a $\gamma$-contraction (Lemma A.9) with fixed-point $\hat{Q}^*_k$. So, by the law of large numbers, $\lim_{m \to \infty} \hat{\mathcal{T}}_{k,m} = \hat{\mathcal{T}}_k$, and $\|\hat{Q}^{\text{est}}_{k,m} - \hat{Q}^*_k\|_\infty \to 0$ as $m \to \infty$. For finite $m$, $\|\hat{Q}^{\text{est}}_{k,m} - \hat{Q}^*_k\|_\infty =: \epsilon_{k,m}$ is the well-studied Bellman noise:

**Lemma 3.3** (Theorem 1 of Li et al. (2022)). *For $k \in [n]$ and $m \in \mathbb{N}$, where $m$ is the number of samples in Equation (12), there is a Bellman noise $\epsilon_{k,m}$ with $\|\hat{Q}^{\text{est}}_{k,m} - \hat{Q}^*_k\|_\infty \leq \epsilon_{k,m} \leq O(1/\sqrt{m})$.*

With the above preparations, we are now primed to present our main result: a bound on the optimality gap, for our learned policy $\pi^{\text{est}}_{k,m}$, that decays with $k$. Section 4 outlines the proof of Theorem 3.4.

**Theorem 3.4.** *For any state $s \in \mathcal{S}_g \times \mathcal{S}_l^n$,*

$$V^{\pi^*}(s) - V^{\pi^{\text{est}}_{k,m}}(s) \leq \frac{2\tilde{r}}{(1-\gamma)^2} \left( \sqrt{\frac{n-k+1}{2nk} \ln(2|\mathcal{S}_l||\mathcal{A}_g|\sqrt{k})} + \frac{1}{\sqrt{k}} \right) + \frac{2\epsilon_{k,m}}{1-\gamma}.$$

**Corollary 3.5.** *Theorem 3.4 implies an asymptotically decaying optimality gap for our learned policy $\tilde{\pi}^{\text{est}}_{k,m}$. Further, from Lemma 3.3, $\epsilon_{k,m} \leq O(1/\sqrt{m})$. Hence,*

$$V^{\pi^*}(s) - V^{\pi^{\text{est}}_{k,m}}(s) \leq \tilde{O}\left( 1/\sqrt{k} + 1/\sqrt{m} \right). \tag{16}$$

**Discussion 3.6.** Between Algorithms 1 and 2, the sample complexity to learn $\hat{\pi}_{k,m}$ for a fixed $k$ is $\min\{O(|\mathcal{S}_g||\mathcal{A}_g||\mathcal{S}_l|^k), O(|\mathcal{S}_g||\mathcal{A}_g||\mathcal{S}_l|k^{|\mathcal{S}_l|})\}$. By Theorem 3.4, as $k \to n$, the optimality gap decays, revealing a fundamental trade-off in the choice of $k$: increasing $k$ improves the policy, but increases the size of the $Q$-function. We explore this trade-off further in our experiments. For $k = O(\log n)$ and $m \to \infty$, the runtime is $\min\{O(|\mathcal{S}_g||\mathcal{A}_g|n^{\log |\mathcal{S}_l|}), O(|\mathcal{S}_g||\mathcal{A}_g||\mathcal{S}_l|(\log n)^{|\mathcal{S}_l|})\}$. This is an exponential speedup on the complexity from mean-field value iteration (from $\text{poly}(n)$ to $\text{poly}(\log n)$), as well as over traditional value-iteration (from $\exp(n)$ to $\text{poly}(n)$). Further, the optimality gap decays to 0 at the rate of $O(1/\sqrt{\log n})$.

**Discussion 3.7.** In the non-tabular setting with infinite state/action spaces, one could replace the $Q$-learning algorithm with an arbitrary value-based RL method that learns $\hat{Q}_k$ with function approximation (Sutton et al., 1999a) such as deep $Q$-networks (Silver et al., 2016). Doing so introduces a further error that factors into the bound in Theorem 3.5. We formalize this intuition in Appendix E.

# 4 PROOF OUTLINE

This section details an outline for the proof of Theorem 3.4, as well as some key ideas. At a high level, our SUBSAMPLE-Q framework recovers exact mean-field $Q$ learning and traditional value iteration when $k = n$ and as $m \to \infty$. Further, as $k \to n$, $\hat{Q}_k^*$ should intuitively get closer to $Q^*$ from which the optimal policy is derived. Thus, the proof is divided into three major steps: firstly, we prove a Lipschitz continuity bound between $\hat{Q}_k^*$ and $\hat{Q}_n^*$ in terms of the total variation (TV) distance between $F_{s_\Delta}$ and $F_{s_{[n]}}$. Next, we bound the TV distance between $F_{s_\Delta}$ and $F_{s_{[n]}}$. Finally, we bound the value differences between $\pi_{k,m}^{\text{est}}$ and $\pi^*$ by bounding $Q^*(s, \pi^*(s)) - Q^*(s, \pi_{k,m}^{\text{est}}(s))$ and then using the performance difference lemma from Kakade & Langford (2002).

**Step 1: Lipschitz Continuity Bound.** To compare $\hat{Q}_k^*(s_g, F_{s_\Delta}, a_g)$ with $Q^*(s, a_g)$, we prove a Lipschitz continuity bound between $\hat{Q}_k^*(s_g, F_{s_\Delta}, a_g)$ and $\hat{Q}_{k'}^*(s_g, F_{s_{\Delta'}}, a_g)$ with respect to the TV distance measure between $s_\Delta \in \binom{s_{[n]}}{k}$ and $s_{\Delta'} \in \binom{s_{[n]}}{k'}$:

**Theorem 4.1** (Lipschitz continuity in $\hat{Q}_k^*$)**.** *For all $(s, a_g) \in \mathcal{S} \times \mathcal{A}_g$, $\Delta \in \binom{[n]}{k}$ and $\Delta' \in \binom{[n]}{k'}$,*

$$|\hat{Q}_k^*(s_g, F_{s_\Delta}, a_g) - \hat{Q}_{k'}^*(s_g, F_{s_{\Delta'}}, a_g)| \le 2(1-\gamma)^{-1}\|r_l(\cdot, \cdot)\|_\infty \cdot \text{TV}\left(F_{s_\Delta}, F_{s_{\Delta'}}\right)$$

We defer the proof of Theorem 4.1 to Appendix C.6. See Figure 3 for a comparison between the $\hat{Q}_k^*$ learning and estimation process, and the exact $Q$-learning framework.

**Step 2: Bounding Total Variation (TV) Distance.** We bound the TV distance between $F_{s_\Delta}$ and $F_{s_{[n]}}$, where $\Delta \in \mathcal{U}\binom{[n]}{k}$. This task is equivalent to bounding the discrepancy between the empirical distribution and the distribution of the underlying finite population. Since each $i \in \Delta$ is uniformly sampled *without* replacement, standard concentration inequalities do not apply as they require the random variables to be i.i.d. Further, standard TV distance bounds using KL divergence produce a suboptimal decay as $|\Delta| \to n$ (Lemma C.7). Hence, we prove the following probabilistic result (which generalizes the Dvoretzky–Kiefer–Wolfowitz (DKW) concentration inequality (Dvoretzky et al., 1956) to the regime of sampling without replacement:

**Theorem 4.2.** *Given a finite population $\mathcal{X} = (x_1, \ldots, x_n)$ for $\mathcal{X} \in \mathcal{S}_l^n$, let $\Delta \subseteq [n]$ be a uniformly random sample from $\mathcal{X}$ of size $k$ chosen without replacement. Fix $\epsilon > 0$. Then, for all $x \in \mathcal{S}_l$:*

$$\Pr\left[\sup_{x \in \mathcal{S}_l}\left|\frac{1}{|\Delta|}\sum_{i \in \Delta}\mathbb{1}\{x_i = x\} - \frac{1}{n}\sum_{i \in [n]}\mathbb{1}\{x_i = x\}\right| \le \epsilon\right] \ge 1 - 2|\mathcal{S}_l|e^{-\frac{2kn\epsilon^2}{n-k+1}}.$$

Then, by Theorem 4.2 and the definition of TV distance from Section 2, we have that for $\delta \in (0, 1]$,

$$\Pr\left(\text{TV}(F_{s_\Delta}, F_{s_{[n]}}) \le \sqrt{\frac{n-k+1}{8nk}\ln\frac{2|\mathcal{S}_l|}{\delta}}\right) \ge 1 - \delta. \tag{17}$$

We then apply this result to our global decision-making problem by studying the rate of decay of the objective function between our learned policy $\pi_{k,m}^{\text{est}}$ and the optimal policy $\pi^*$ (Theorem 3.4).

**Step 3: Performance Difference Lemma to Complete the Proof.** As a consequence of the prior two steps and Lemma 3.3, $Q^*(s, a_g')$ and $\hat{Q}_{k,m}^{\text{est}}(s_g, F_{s_\Delta}, a_g')$ become similar as $k \to n$ (see Theorem C.6). We further prove that the value generated by their policies $\pi^*$ and $\pi_{k,m}^{\text{est}}$ must also be very close (where the residue shrinks as $k \to n$). We then use the well-known performance difference lemma (Kakade & Langford, 2002) which we restate in Appendix D.2. A crucial theorem needed to use the performance difference lemma is a bound on $Q^*(s', \pi^*(s')) - Q^*(s', \hat{\pi}_{k,m}^{\text{est}}(s_g', F_{s_{\Delta'}}))$. Therefore, we formulate and prove Theorem 4.3 which yields a probabilistic bound on this difference, where the randomness is over the choice of $\Delta \in \binom{[n]}{k}$:

**Theorem 4.3.** *For a fixed $s' \in \mathcal{S} := \mathcal{S}_g \times \mathcal{S}_l^n$ and for $\delta \in (0, 1]$, with probability atleast $1 - 2|\mathcal{A}_g|\delta$:*

$$Q^*(s', \pi^*(s')) - Q^*(s', \hat{\pi}_{k,m}^{\mathrm{est}}(s'_g, F_{s'_\Delta})) \leq \frac{2\|r_l(\cdot, \cdot)\|_\infty}{1 - \gamma} \sqrt{\frac{n - k + 1}{2nk} \ln\left(\frac{2|\mathcal{S}_l|}{\delta}\right)} + 2\epsilon_{k,m}.$$

We defer the proof of Theorem 4.3 and finding optimal value of $\delta$ to D.5 in the Appendix. Using Theorem 4.3 and the performance difference lemma leads to Theorem 3.4.

## 5    EXPERIMENTS

This section provides examples and numerical simulation results to validate our theoretical framework. All numerical experiments were run on a 3-core CPU server equipped with a 12GB RAM. We chose parameters with complexity sufficient to only validate the theory, such as the computational speedups, pseudo-heterogeneity of each local agent, and the decaying optimality gap.

**Example 5.1** (Demand-Response (DR)). DR is a pathway in the transformation towards a sustainable electricity grid where users (local agents) are compensated to lower their electricity consumption to a level set by a regulator (global agent). DR has applications ranging from pricing strategies for EV charging stations, regulating the supply of any product in a market with fluctuating demands, and maximizing the efficiency of allocating resources. We ran a small-scale simulation with $n = 8$ local agents, and a large-scale simulation with $n = 50$ local agents, where the goal was to learn an optimal policy for the global agent to moderate supply in the presence of fluctuating demand.

Let each local agent $i \in [n]$ have a state $s_i(t) = (\varepsilon_i, c_i(t), d_i(t)) \in \mathcal{S}_l := \mathcal{E} \times \mathcal{C} \times \mathcal{D} \subset \mathbb{Z}^3$. Here, $\varepsilon_i$ is the agent's type, $c_i(t)$ is its consumption, and $d_i(t)$ is its desired consumption level. Let $s_g(t) \in \mathcal{S}_g$ be the DR signal (target consumption set by the regulator). The global agent's transition is $s_g(t + 1) = \Pi^{\mathcal{S}_g}(s_g(t) + a_g(t))$, i.e., $a_g(t)$ changes the DR signal. Then, $s_i(t + 1) = (\varepsilon_i, c_i(t + 1), d_i(t + 1))$, where $d_i(t + 1)$ fluctuates based on the agent's type and prior demand:

$$c_i(t + 1) = \begin{cases} d_i(t), & d_i(t) \leq s_g(t) \\ \Pi^{\mathcal{C}}[d_i(t) + (s_g(t) - c_i(t))\mathcal{U}\{0, 1\}], & d_i(t) > s_g(t) \end{cases},$$

$$d_i(t + 1) = \begin{cases} d_i(t) + \mathcal{U}\{0, 1\}, & \varepsilon_i = 1 \\ \mathcal{U}[\mathcal{D}], & \varepsilon_i = 2 \end{cases},$$

where $\Pi^{\mathcal{C}}$ denotes a projection onto $\mathcal{C}$ in $\ell_1$-norm. Intuitively, the local agent either chases its desired consumption or reduces its consumption to match $s_g(t)$. The system's reward at each step is $r_g(s_g, a_g) = 15/s_g - \mathbf{1}\{a_g = -1\}$ and $r_l(s_i, s_g) = c_i - \frac{1}{2}\mathbf{1}\{c_i > s_g\}$. We set $\mathcal{C} = \mathcal{D} = [3], \mathcal{E} = \{1, 2\}, \gamma = 0.9, m = 10$, and the length of the decision game to be $T' = 300$. We use $T = 300$ iterations for the small-scale simulation, and $T = 50$ iterations for the large scale simulation.

For the small-scale simulation, Figure 1a illustrates the polynomial speedup of Algorithm 2 (note that $k = n$ exactly recovers mean-field value iteration (Yang et al., 2018), which we treat as our benchmark for comparison). Figure 1b plots the reward-optimality gap for varying $k$. Figure 1c plots the cumulative reward of the large-scale experiment. We observe that the rewards (on average) grow monotonically as they obey our worst-case guarantee in Theorem 3.4.

**Example 5.2** (Queueing). We model a system with $n$ queues, where $s_i(t) \in \mathcal{S}_l := \mathbb{N}$ at time $t$ denotes the number of jobs at time $t$ for queue $i \in [n]$. We model the job allocation mechanism as a global agent where $s_g(t) \in \mathcal{S}_g = \mathcal{A}_g = [n]$. Here, $s_g(t)$ denotes the queue to which the next job should be delivered. We choose the state transitions to capture the stochastic job arrival and departure: $s_g(t + 1) = a_g(t)$, and $s_i(t + 1) = \min\{c, \max\{0, s_i(t) + \mathbb{1}\{s_g(t) = i\} - \mathrm{Bern}(p)\}\}$. For the rewards, we set $r_g(s_g, a_g) = 0$ and $r_l(s_i, s_g) = -s_i - 10 \cdot \mathbf{1}\{s_i > c\}$, where $p = 0.8$ is the probability of finishing a job, $c = 30$ is the capacity of each queue, and $\gamma = 0.9$.

This simulation ran on a system of $n = 50$ local agents. The goal was to learn an optimal policy for a dispatcher to send incoming jobs to. We ran Algorithm 2 for $T = 300$ empirical adapted Bellman iterations with $m = 30$, and ran Algorithm 3 for $T' = 100$ iterations. Figure 2 illustrates the log-scale reward-optimality gap for varying $k$, showing that the gap decreases monotonically as $k \to n$ with a decay rate that is consistent with the $O(1/\sqrt{k})$ upper bound in Theorem 3.4.

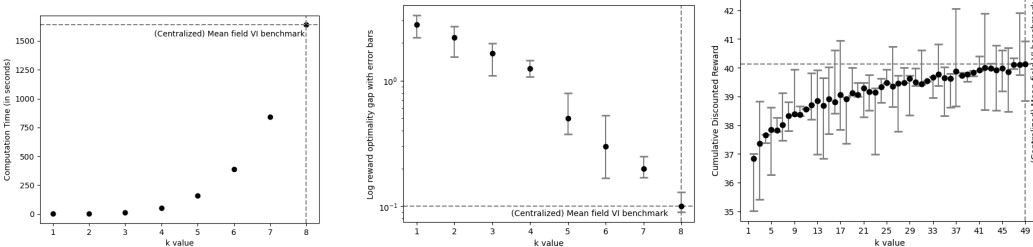

Figure 1: Demand-Response simulation. a) Computation time to learn $\hat{\pi}_{k,m}^{\text{est}}$ for $k \leq n = 8$. b) Reward optimality gap (log scale) with $\pi_{k,m}^{\text{est}}$ running 300 iterations for $k \leq n = 8$, c) Discounted cumulative rewards for $k \leq n = 50$. We note that $k = n$ recovers the mean-field RL iteration solution.

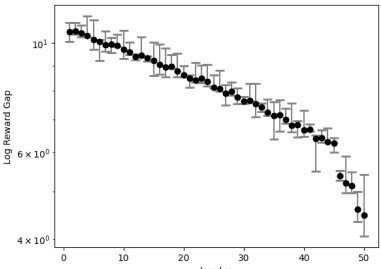

Figure 2: Reward optimality gap (log scale) with $\pi_{k,m}^{\text{est}}$ running 300 iterations.

## 6 CONCLUSION, LIMITATIONS, AND FUTURE WORK

**Conclusion.** This work considers a global decision-making agent in the presence of $n$ local homogeneous agents. We propose SUBSAMPLE-Q which derives a policy $\pi_{k,m}^{\text{est}}$ where $k \leq n$ and $m \in \mathbb{N}$ are tunable parameters, and show that $\pi_{k,m}^{\text{est}}$ converges to the optimal policy $\pi^*$ with a decay rate of $O(1/\sqrt{k} + \epsilon_{k,m})$, where $\epsilon_{k,m}$ is the Bellman noise. To establish the result, we develop an analytic framework which constructs an adapted Bellman operator $\hat{\mathcal{T}}_k$, shows a Lipschitz-continuity result for $\hat{Q}_k^*$, generalizes the DKW inequality, and proves a probabilistic bound on $Q$-functions with different actions. Further, we extend this result to the non-tabular setting with infinite state and action spaces. Finally, we validate our theoretical result through numerical experiments.

**Limitations and Future Work.** We recognize several future directions. Firstly, this model studies a 'star-network' setting to model a single source of density. It would be fascinating to extend this subsampling framework to general networks. We believe expander-graph decompositions (Anand & Umans, 2023; Reingold, 2008) are amenable for this. A second direction would be to find connections between our sub-sampling method to algorithms in federated learning, where the rewards can be stochastic, and to incorporate learning rates (Lin et al., 2021) to attain numerical stability. A third limitation of this work is that we have only partially resolved the problem for truly heterogeneous local agents by adding a 'type' property to each local agent to model some pseudoheterogeneity in the state space of each agent. Finally, it would be exciting to generalize this work to the online setting without a generative oracle. For this, we conjecture that tools from recent works on stochastic approximation (Chen & Theja Maguluri, 2022) and no-regret RL (Jin et al., 2021) might be valuable.

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

**Outline of the Appendices**.

- Appendix A presents additional definitions and remarks that support the main body.
- Appendix B-C contains a detailed proof of the Lipschitz continuity bound in Theorem 4.1 and total variation distance bound in Theorem 4.2.
- Appendix D contains a detailed proof of the main result in Theorem 3.4.

Table 1: Important notations in this paper.

| Notation | Meaning |
|---|---|
| $\|\cdot\|_1$ | $\ell_1$ (Manhattan) norm; |
| $\|\cdot\|_\infty$ | $\ell_\infty$ norm; |
| $\mathbb{R}^d$ | The set of $d$-dimensional reals; |
| $[n]$ | The set $\{1,\ldots,n\}$, where $n \in \mathbb{Z}_+$; |
| $\binom{[n]}{k}$ | The set of $k$-sized subsets of $\{1,\ldots,n\}$; |
| $a_g$ | $a_g \in \mathcal{A}_g$ is the action of the global agent; |
| $s_g$ | $s_g \in \mathcal{S}_g$ is the state of the global agent; |
| $s_1,\ldots,s_n$ | $s_1,\ldots,s_n \in \mathcal{S}_l^n$ are the states of the local agents $1,\ldots,n$; |
| $s$ | $s = (s_g, s_1, \ldots, s_n) \in \mathcal{S}_g \times \mathcal{S}_l^n$ is the tuple of states of all agents; |
| $s_\Delta$ | For $\Delta \subseteq [n]$, and a collection of variables $\{s_1,\ldots,s_n\}$, $s_\Delta := \{s_i : i \in \Delta\}$; |
| $\sigma(s_\Delta, s'_\Delta)$ | Product sigma-algebra generated by sequences $s_\Delta$ and $s'_\Delta$; |
| $\mu_k(\mathcal{S}_l)$ | $\mu_k(\mathcal{S}_l) := \{0, 1/k, 2/k, \ldots, 1\}^{|\mathcal{S}_l|}$; |
| $\mu(\mathcal{S}_l)$ | $\mu(\mathcal{S}_l) := \mu_n(\mathcal{S}_l) := \{0, 1/n, 2/n, \ldots, 1\}^{|\mathcal{S}_l|}$; |
| $\pi^*$ | $\pi^*$ is the optimal deterministic policy function such that $a = \pi^*(s)$; |
| $\hat{\pi}_{k,m}^{\text{est}}$ | $\hat{\pi}_{k,m}^{\text{est}}$ is the optimal deterministic policy function on a $k$ local agent system; |
| $\pi_{k,m}^{\text{est}}$ | $\pi_{k,m}^{\text{est}}$ is the stochastic policy mapping $a \sim \tilde{\pi}_{k,m}^{\text{est}}(s)$ learned with parameter $k$; |
| $P_g(\cdot\|s_g,a_g)$ | $P_g(\cdot\|s_g,a_g)$ is the stochastic transition kernel for the state of the global agent; |
| $P_l(\cdot\|s_i,s_g)$ | $P_l(\cdot\|s_i,s_g)$ is the stochastic transition kernel for the state of any local agent $i \in [n]$; |
| $r_g(s_g,a_g)$ | $r_g$ is the global agent's component of the reward; |
| $r_l(s_i,s_g)$ | $r_l$ is the component of the reward for local agent $i \in [n]$; |
| $r(s,a)$ | $r(s,a) := r_{[n]}(s,a) = r_g(s_g,a_g) + \frac{1}{n}\sum_{i\in[n]} r_l(s_i,s_g)$ is the reward of the system; |
| $r_\Delta(s,a)$ | $r_\Delta(s,a) = r_g(s_g,a_g) + \frac{1}{|\Delta|}\sum_{i\in\Delta} r_l(s_i,s_g)$ is the reward with $|\Delta| = k$ local agents; |
| $\mathcal{T}$ | $\mathcal{T}$ is the centralized Bellman operator; |
| $\hat{\mathcal{T}}_k$ | $\hat{\mathcal{T}}_k$ is the Bellman operator on a constrained system of $|\Delta| = k$ local agents; |
| $\Pi^\Theta(y)$ | $\ell_1$ projection of $y$ onto set $\Theta$. |

## A MATHEMATICAL BACKGROUND AND ADDITIONAL REMARKS

**Definition A.1** (Lipschitz continuity). Given two metric spaces $(\mathcal{X}, d_\mathcal{X})$ and $(\mathcal{Y}, d_\mathcal{Y})$ and a constant $L \in \mathbb{R}_+$, a mapping $f : \mathcal{X} \to \mathcal{Y}$ is $L$-Lipschitz continuous if for all $x, y \in \mathcal{X}$, $d_\mathcal{Y}(f(x), f(y)) \leq L \cdot d_\mathcal{X}(x,y)$.

**Theorem A.2** (Banach-Caccioppoli fixed point theorem Banach (1922)). Consider the metric space $(\mathcal{X}, d_\mathcal{X})$, and $T : \mathcal{X} \to \mathcal{X}$ such that $T$ is a $\gamma$-Lipschitz continuous mapping for $\gamma \in (0,1)$. Then, by the Banach-Caccioppoli fixed-point theorem, there exists a unique fixed point $x^* \in \mathcal{X}$ for which $T(x^*) = x^*$. Additionally, $x^* = \lim_{s\to\infty} T^s(x_0)$ for any $x_0 \in \mathcal{X}$.

For convenience, we restate below the various Bellman operators under consideration.

**Definition A.3** (Bellman Operator $\mathcal{T}$).
$$\mathcal{T}Q^t(s, a_g) := r_{[n]}(s, a_g) + \gamma \mathbb{E}_{\substack{s'_g \sim P_g(\cdot|s_g,a_g), \\ s'_i \sim P_l(\cdot|s_i,s_g), \forall i \in [n]}} \max_{a'_g \in \mathcal{A}_g} Q^t(s', a'_g) \tag{18}$$

**Definition A.4** (Adapted Bellman Operator $\hat{\mathcal{T}}_k$). The adapted Bellman operator updates a smaller $Q$ function (which we denote by $\hat{Q}_k$), for a surrogate system with the global agent and $k \in [n]$ local agents, using mean-field value iteration:
$$\hat{\mathcal{T}}_k\hat{Q}_k^t(s_g, F_{s_\Delta}, a_g) := r_\Delta(s, a_g) + \gamma \mathbb{E}_{\substack{s'_g \sim P_g(\cdot|s_g,a_g), \\ s'_i \sim P_l(\cdot|s_i,s_g), \forall i \in \Delta}} \max_{a'_g \in \mathcal{A}_g} \hat{Q}_k^t(s'_g, F_{s'_\Delta}, a'_g) \tag{19}$$

**Definition A.5** (Empirical Adapted Bellman Operator $\hat{\mathcal{T}}_{k,m}$). The empirical adapted Bellman operator $\hat{\mathcal{T}}_{k,m}$ empirically estimates the adapted Bellman operator update using mean-field value iteration by drawing $m$ random samples of $s_g \sim P_g(\cdot|s_g, a_g)$ and $s_i \sim P_l(\cdot|s_i, s_g)$ for $i \in \Delta$, where for $j \in [m]$, the $j$'th random sample is given by $s_g^j$ and $s_\Delta^j$:

$$\hat{\mathcal{T}}_{k,m}\hat{Q}_{k,m}^t(s_g, F_{s_\Delta}, a_g) := r_\Delta(s, a_g) + \frac{\gamma}{m}\sum_{j\in[m]}\max_{a_g'\in\mathcal{A}_g}\hat{Q}_{k,m}^t(s_g^j, F_{s_\Delta^j}, a_g') \tag{20}$$

**Remark A.6.** We remark on the following relationships between the variants of the Bellman operators from Theorems A.3 to A.5. First, by the law of large numbers, we have $\lim_{m\to\infty}\hat{\mathcal{T}}_{k,m} = \hat{\mathcal{T}}_k$, where the error decays in $O(1/\sqrt{m})$ by the Chernoff bound. Secondly, by comparing Theorem A.4 and Theorem A.3, we have $\mathcal{T}_n = \mathcal{T}$.

**Lemma A.7.** For any $\Delta \subseteq [n]$ such that $|\Delta| = k$, suppose $0 \le r_\Delta(s, a_g) \le \tilde{r}$. Then, $\hat{Q}_k^t \le \frac{\tilde{r}}{1-\gamma}$.

*Proof.* We prove this by induction on $t \in \mathbb{N}$. The base case is satisfied as $\hat{Q}_k^0 = 0$. Assume that $\|\hat{Q}_k^{t-1}\|_\infty \le \frac{\tilde{r}}{1-\gamma}$. We bound $\hat{Q}_k^{t+1}$ from the Bellman update at each time step as follows, for all $s_g \in \mathcal{S}_g, F_{s_\Delta} \in \mu_k(\mathcal{S}_l|), a_g \in \mathcal{A}_g$:

$$\hat{Q}_k^{t+1}(s_g, F_{s_\Delta}, a_g) = r_\Delta(s, a_g) + \gamma\mathbb{E}_{\substack{s_g'\sim P_g(\cdot|s_g,a_g),\\ s_i'\sim P_l(\cdot|s_i,s_g),\forall i\in\Delta}}\max_{a_g'\in\mathcal{A}_g}\hat{Q}_k^t(s_g', F_{s_\Delta'}, a_g')$$

$$\le \tilde{r} + \gamma\max_{a_g'\in\mathcal{A}_g, s_g'\in\mathcal{S}_g, F_{s_\Delta'}\in\mu_k(\mathcal{S}_l)}\hat{Q}_k^t(s_g', F_{s_\Delta'}, a_g') \le \frac{\tilde{r}}{1-\gamma}$$

Here, the first inequality follows by noting that the maximum value of a random variable is at least as large as its expectation. The second inequality follows from the inductive hypothesis. $\square$

**Remark A.8.** Theorem A.7 is independent of the choice of $k$. Therefore, for $k = n$, this implies an identical bound on $Q^t$. A similar argument as Theorem A.7 implies an identical bound on $\hat{Q}_{k,m}^t$.

Recall that the original Bellman operator $\mathcal{T}$ satisfies a $\gamma$-contractive property under the infinity norm. We similarly show that $\hat{\mathcal{T}}_k$ and $\hat{\mathcal{T}}_{k,m}$ satisfy a $\gamma$-contractive property under infinity norm in Theorem A.9 and Theorem A.10.

**Lemma A.9.** $\hat{\mathcal{T}}_k$ satisfies the $\gamma$-contractive property under infinity norm:

$$\|\hat{\mathcal{T}}_k\hat{Q}_k' - \hat{\mathcal{T}}_k\hat{Q}_k\|_\infty \le \gamma\|\hat{Q}_k' - \hat{Q}_k\|_\infty$$

*Proof.* Suppose we apply $\hat{\mathcal{T}}_k$ to $\hat{Q}_k(s_g, F_{s_\Delta}, a_g)$ and $\hat{Q}_k'(s_g, F_{s_\Delta}, a_g)$ for $|\Delta| = k$. Then:

$$\|\hat{\mathcal{T}}_k\hat{Q}_k' - \hat{\mathcal{T}}_k\hat{Q}_k\|_\infty$$

$$= \gamma\max_{\substack{s_g\in\mathcal{S}_g,\\ a_g\in\mathcal{A}_g,\\ F_{s_\Delta}\in\mu_k(\mathcal{S}_l)}}\left|\mathbb{E}_{\substack{s_g'\sim P_g(\cdot|s_g,a_g),\\ s_i'\sim P_l(\cdot|s_i,s_g),\\ \forall s_i'\in s_\Delta',}}\max_{a_g'\in\mathcal{A}_g}\hat{Q}_k'(s_g', F_{s_\Delta'}, a_g') - \mathbb{E}_{\substack{s_g'\sim P_g(\cdot|s_g,a_g),\\ s_i'\sim P_l(\cdot|s_i,s_g),\\ \forall s_i'\in s_\Delta'}}\max_{a_g'\in\mathcal{A}_g}\hat{Q}_k(s_g', F_{s_\Delta'}, a_g')\right|$$

$$\le \gamma\max_{s_g'\in\mathcal{S}_g, F_{s_\Delta'}\in\mu_k(\mathcal{S}_l), a_g'\in\mathcal{A}_g}\left|\hat{Q}_k'(s_g', F_{s_\Delta'}, a_g') - \hat{Q}_k(s_g', F_{s_\Delta'}, a_g')\right|$$

$$= \gamma\|\hat{Q}_k' - \hat{Q}_k\|_\infty$$

The equality implicitly cancels the common $r_\Delta(s, a_g)$ terms from each application of the adapted-Bellman operator. The inequality follows from Jensen's inequality, maximizing over the actions, and bounding the expected value with the maximizers of the random variables. The last line recovers the definition of infinity norm. $\square$

**Lemma A.10.** $\hat{\mathcal{T}}_{k,m}$ satisfies the $\gamma$-contractive property under infinity norm.

*Proof.* Similarly to Theorem A.9, suppose we apply $\hat{\mathcal{T}}_{k,m}$ to $\hat{Q}_{k,m}(s_g, F_{s_\Delta}, a_g)$ and $\hat{Q}'_{k,m}(s_g, F_{s_\Delta}, a_g)$. Then:

$$\|\hat{\mathcal{T}}_{k,m}\hat{Q}_k - \hat{\mathcal{T}}_{k,m}\hat{Q}'_k\|_\infty = \frac{\gamma}{m} \left\| \sum_{j \in [m]} ( \max_{a'_g \in \mathcal{A}_g} \hat{Q}_k(s^j_g, F_{s^j_\Delta}, a'_g) - \max_{a'_g \in \mathcal{A}_g} \hat{Q}'_k(s^j_g, F_{s^j_\Delta}, a'_g)) \right\|_\infty$$

$$\leq \gamma \max_{a'_g \in \mathcal{A}_g, s'_g \in \mathcal{S}_g, s_\Delta \in \mathcal{S}^k_l} |\hat{Q}_k(s'_g, F_{s'_\Delta}, a'_g) - \hat{Q}'_k(s'_g, F_{s'_\Delta}, a'_g)|$$

$$\leq \gamma \|\hat{Q}_k - \hat{Q}'_k\|_\infty$$

The first inequality uses the triangle inequality and the general property $|\max_{a \in A} f(a) - \max_{b \in A} f(b)| \leq \max_{c \in A} |f(a) - f(b)|$. In the last line, we recover the definition of infinity norm. $\qquad\square$

**Remark A.11.** Intuitively, the $\gamma$-contractive property of $\hat{\mathcal{T}}_k$ and $\hat{\mathcal{T}}_{k,m}$ causes the trajectory of two $\hat{Q}_k$ and $\hat{Q}_{k,m}$ functions on the same state-action tuple to decay by $\gamma$ at each time step such that repeated applications of their corresponding Bellman operators produce a unique fixed-point from the Banach-Cacciopoli fixed-point theorem which we introduce in Theorems A.12 and A.13.

**Definition A.12** ($\hat{Q}^*_k$). *Suppose $\hat{Q}^0_k := 0$ and let $\hat{Q}^{t+1}_k(s_g, F_{s_\Delta}, a_g) = \hat{\mathcal{T}}_k \hat{Q}^t_k(s_g, F_{s_\Delta}, a_g)$ for $t \in \mathbb{N}$. Denote the fixed-point of $\hat{\mathcal{T}}_k$ by $\hat{Q}^*_k$ such that $\hat{\mathcal{T}}_k \hat{Q}^*_k(s_g, F_{s_\Delta}, a_g) = \hat{Q}^*_k(s_g, F_{s_\Delta}, a_g)$.*

**Definition A.13** ($\hat{Q}^{\text{est}}_{k,m}$). *Suppose $\hat{Q}^0_{k,m} := 0$ and let $\hat{Q}^{t+1}_{k,m}(s_g, F_{s_\Delta}, a_g) = \hat{\mathcal{T}}_{k,m} \hat{Q}^t_{k,m}(s_g, F_{s_\Delta}, a_g)$ for $t \in \mathbb{N}$. Denote the fixed-point of $\hat{\mathcal{T}}_{k,m}$ by $\hat{Q}^{\text{est}}_{k,m}$ such that $\hat{\mathcal{T}}_{k,m} \hat{Q}^{\text{est}}_{k,m}(s_g, F_{s_\Delta}, a_g) = \hat{Q}^{\text{est}}_{k,m}(s_g, F_{s_\Delta}, a_g)$.*

Furthermore, recall the assumption on our empirical approximation of $\hat{Q}^*_k$:

**Theorem 3.3.** For all $k \in [n]$ and $m \in \mathbb{N}$, we assume that:

$$\|\hat{Q}^{\text{est}}_{k,m} - \hat{Q}^*_k\|_\infty \leq \epsilon_{k,m} \tag{21}$$

**Corollary A.14.** *Observe that by backpropagating results of the $\gamma$-contractive property for $T$ steps:*

$$\|\hat{Q}^*_k - \hat{Q}^T_k\|_\infty \leq \gamma^T \cdot \|\hat{Q}^*_k - \hat{Q}^0_k\|_\infty \tag{22}$$

$$\|\hat{Q}^{\text{est}}_{k,m} - \hat{Q}^T_{k,m}\|_\infty \leq \gamma^T \cdot \|\hat{Q}^{\text{est}}_{k,m} - \hat{Q}^0_{k,m}\|_\infty \tag{23}$$

*Further, noting that $\hat{Q}^0_k = \hat{Q}^0_{k,m} := 0$, $\|\hat{Q}^*_k\|_\infty \leq \frac{\tilde{r}}{1-\gamma}$, and $\|\hat{Q}^{\text{est}}_{k,m}\|_\infty \leq \frac{\tilde{r}}{1-\gamma}$ from Theorem A.7:*

$$\|\hat{Q}^*_k - \hat{Q}^T_k\|_\infty \leq \gamma^T \frac{\tilde{r}}{1-\gamma} \tag{24}$$

$$\|\hat{Q}^{\text{est}}_{k,m} - \hat{Q}^T_{k,m}\|_\infty \leq \gamma^T \frac{\tilde{r}}{1-\gamma} \tag{25}$$

**Remark A.15.** Theorem A.14 characterizes the error decay between $\hat{Q}^T_k$ and $\hat{Q}^*_k$ as well as between $\hat{Q}^T_{k,m}$ and $\hat{Q}^{\text{est}}_{k,m}$ and shows that it decays exponentially in the number of corresponding Bellman iterations with the $\gamma^T$ multiplicative factor.

Furthermore, we characterize the maximal policies greedy policies obtained from $Q^*, \hat{Q}^*_k$, and $\hat{Q}^{\text{est}}_{k,m}$.

**Definition A.16** ($\pi^*$). *The greedy policy derived from $Q^*$ is*

$$\pi^*(s) := \arg \max_{a_g \in \mathcal{A}_g} Q^*(s, a_g).$$

**Definition A.17** ($\hat{\pi}_k^*$). *The greedy policy from $\hat{Q}_k^*$ is*

$$\hat{\pi}_k^*(s_g, F_{s_\Delta}) := \arg \max_{a_g \in \mathcal{A}_g} \hat{Q}_k^*(s_g, F_{s_\Delta}, a_g).$$

**Definition A.18** ($\hat{\pi}_{k,m}^{\text{est}}$). *The greedy policy from $\hat{Q}_{k,m}^{\text{est}}$ is given by*

$$\hat{\pi}_{k,m}^{\text{est}}(s_g, F_{s_\Delta}) := \arg \max_{a_g \in \mathcal{A}_g} \hat{Q}_{k,m}^{\text{est}}(s_g, F_{s_\Delta}, a_g).$$

Figure 3 details the analytic flow on how we use the empirical adapted Bellman operator to perform value iteration on $\hat{Q}_{k,m}$ to get $\hat{Q}_{k,m}^{\text{est}}$ which approximates $Q^*$.

$$\hat{Q}_{k,m}^0(s_g, F_{s_\Delta}, a_g)$$

$$\Big\downarrow^{(1)}$$

$$\hat{Q}_{k,m}^{\text{est}}(s_g, F_{s_\Delta}, a_g) \xrightarrow{\overset{(2)}{=}} \hat{Q}_k^*(s_g, F_{s_\Delta}, a_g) \xrightarrow{\overset{(3)}{\approx}} \hat{Q}_n^*(s_g, F_{s_{[n]}}, a_g)$$

$$\Big\downarrow^{(4)}_{=}$$

$$Q^*(s_g, s_{[n]}, a_g)$$

Figure 3: Flow of the algorithm and relevant analyses in learning $Q^*$. Here, (1) follows by performing Algorithm 2 (SUBSAMPLE-Q: Learning) on $\hat{Q}_{k,m}^0$. (2) follows from Theorem 3.3. (3) follows from the Lipschitz continuity and total variation distance bounds in Theorems 4.1 and 4.2. Finally, (4) follows from noting that $\hat{Q}_n^* = Q^*$.

Algorithm 4 provides a stable implementation of Algorithm 2: SUBSAMPLE-Q: Learning, where we incorporate a sequence of learning rates $\{\eta_t\}_{t \in [T]}$ into the framework (Watkins & Dayan, 1992). Algorithm 4 is also provably numerical stable under fixed-point arithmetic (Anand et al., 2024).

---

**Algorithm 4** Stable (Practical) Implementation of Algorithm 2: SUBSAMPLE-Q: Learning

---

**Require:** A multi-agent system as described in Section 2. Parameter $T$ for the number of iterations in the initial value iteration step. Hyperparameter $k \in [n]$. Discount parameter $\gamma \in (0, 1)$. Oracle $\mathcal{O}$ to sample $s_g' \sim P_g(\cdot | s_g, a_g)$ and $s_i \sim P_l(\cdot | s_i, s_g)$ for all $i \in [n]$. Sequence of learning rates $\{\eta_t\}_{t \in [T]}$ where $\eta_t \in (0, 1]$.
1: Choose any $\Delta \subseteq [n]$ such that $|\Delta| = k$.
2: Set $\hat{Q}_{k,m}^0(s_g, F_{s_\Delta}, a_g) = 0$ for $(s_g, F_{s_\Delta}, a_g) \in \mathcal{S}_g \times \mu_k(\mathcal{S}_l) \times \mathcal{A}_g$.
3: **for** $t = 1$ to $T$ **do**
4:    **for** $(s_g, F_{s_\Delta}) \in \mathcal{S}_g \times \mu_k(\mathcal{S}_l)$ **do**
5:       **for** $a_g \in \mathcal{A}_g$ **do**
6:          $\hat{Q}_{k,m}^{t+1}(s_g, F_{s_\Delta}, a_g) \leftarrow (1 - \eta_t)\hat{Q}_{k,m}^t(s_g, F_{s_\Delta}, a_g) + \eta_t \hat{\mathcal{T}}_{k,m} \hat{Q}_{k,m}^t(s_g, F_{s_\Delta}, a_g)$
7: For all $(s_g, F_{s_\Delta}) \in \mathcal{S}_g \times \mu_k(\mathcal{S}_l)$, let the approximate policy be

$$\hat{\pi}_{k,m}^T(s_g, F_{s_\Delta}) = \arg \max_{a_g \in \mathcal{A}_g} \hat{Q}_{k,m}^T(s_g, F_{s_\Delta}, a_g).$$

---

Notably, $\hat{Q}_{k,m}^t$ in Algorithm 4 due to a similar $\gamma$-contractive property as in Theorem A.9, given an appropriately conditioned sequence of learning rates $\eta_t$:

**Theorem A.19.** *As $T \to \infty$, if $\sum_{t=1}^T \eta_t = \infty$, and $\sum_{t=1}^T \eta_t^2 < \infty$, then Q-learning converges to the optimal Q function asymptotically with probability 1.*

Furthermore, finite-time guarantees with the learning rate and sample complexity have been shown recently in Chen & Theja Maguluri (2022), which when adapted to our $\hat{Q}_{k,m}$ framework in Algorithm 4 yields:

**Theorem A.20** (Chen & Theja Maguluri (2022)). *For all $t \in [T]$ and $\epsilon > 0$, if $\eta_t = (1-\gamma)^4 \epsilon^2$ and $T = k^{|\mathcal{S}_l|} |\mathcal{S}_g| |\mathcal{A}_g| |\mathcal{S}_l| / (1-\gamma)^5 \epsilon^2$,*

$$\|\hat{Q}_{k,m}^T - \hat{Q}_{k,m}^{\text{est}}\| \leq \epsilon.$$

This global decision-making problem can be viewed as a generalization of the network setting to a specific type of dense graph: the star graph (Figure 4). We briefly elaborate more on this connection below.

**Definition A.21** (Star Graph $S_n$). *For $n \in \mathbb{N}$, the star graph $S_n$ is the complete bipartite graph $K_{1,n}$.*

$S_n$ captures the graph density notion by saturating the set of neighbors for the central node. Furthermore, it models interactions between agents identically to our setting, where the central node is a global agent and the peripheral nodes are local agents. The cardinality of the search space simplex for the optimal policy is $|\mathcal{S}_g| |\mathcal{S}_l|^n |\mathcal{A}_g|$, which is exponential in $n$. Hence, this problem cannot be naively modeled by an MDP: we need to exploit the symmetry of the local agents. This intuition allows our subsampling algorithm to run in polylogarithmic time (in $n$). Further, works that leverage the exponential decaying property that truncates the search space for policies over immediate neighborhoods of agents still rely on the assumption that the graph neighborhood for the agent is sparse Lin et al. (2021); Qu et al. (2020a;b); Lin et al. (2020); however, the graph $S_n$ violates this local sparsity condition; hence, previous methods do not apply to this problem instance.

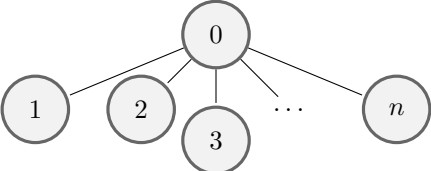

Figure 4: Star graph $S_n$

## B   PROOF OF LIPSCHITZ-CONTINUITY BOUND

This section proves the Lipschitz-continuity bound Theorem 4.1 between $\hat{Q}_k^*$ and $Q^*$ in Theorem B.2 and includes a framework to compare $\frac{1}{\binom{n}{k}} \sum_{\Delta \in \binom{[n]}{k}} \hat{Q}_k^*(s_g, F_{s_\Delta}, a_g)$ and $Q^*(s, a_g)$ in Theorem B.12. The following definition will be relevant to the proof of Theorem 4.1.

**Definition B.1.** *[Joint Stochastic Kernels]*The joint stochastic kernel on $(s_g, s_\Delta)$ for $\Delta \subseteq [n]$ where $|\Delta| = k$ is defined as $\mathcal{J}_k : \mathcal{S}_g \times \mathcal{S}_l^k \times \mathcal{S}_g \times \mathcal{A}_g \times \mathcal{S}_l^k \to [0,1]$, where

$$\mathcal{J}_k(s_g', s_\Delta' | s_g, a_g, s_\Delta) := \Pr[(s_g', s_\Delta') | s_g, a_g, s_\Delta] \tag{26}$$

**Theorem B.2** ($\hat{Q}_k^T$ is $(\sum_{t=0}^{T-1} 2\gamma^t) \|r_l(\cdot, \cdot)\|_\infty$-Lipschitz continuous with respect to $F_{s_\Delta}$ in total variation distance). *Suppose $\Delta, \Delta' \subseteq [n]$ such that $|\Delta| = k$ and $|\Delta'| = k'$. Then:*

$$\left| \hat{Q}_k^T(s_g, F_{s_\Delta}, a_g) - \hat{Q}_{k'}^T(s_g, F_{s_{\Delta'}}, a_g) \right| \leq \left( \sum_{t=0}^{T-1} 2\gamma^t \right) \|r_l(\cdot, \cdot)\|_\infty \cdot \text{TV}\left( F_{s_\Delta}, F_{s_{\Delta'}} \right)$$

*Proof.* We prove this inductively. Note that $\hat{Q}_k^0(\cdot, \cdot, \cdot) = \hat{Q}_{k'}^0(\cdot, \cdot, \cdot) = 0$ from the initialization step in Algorithm 2, which proves the lemma for $T = 0$ since $\text{TV}(\cdot, \cdot) \geq 0$. For the remainder of this proof, we adopt the shorthand $\mathbb{E}_{s_g', s_\Delta'}$ to refer to $\mathbb{E}_{s_g' \sim P_g(\cdot | s_g, a_g), s_i' \sim P_l(\cdot | s_i, s_g), \forall i \in \Delta}$.

Then, at $T = 1$:

$$|\hat{Q}_k^1(s_g, F_{s_\Delta}, a_g) - \hat{Q}_{k'}^1(s_g, F_{s_{\Delta'}}, a_g)|$$

$$= \left|\hat{\mathcal{T}}_k \hat{Q}_k^0(s_g, F_{s_\Delta}, a_g) - \hat{\mathcal{T}}_{k'} \hat{Q}_{k'}^0(s_g, F_{s_{\Delta'}}, a_g)\right|$$

$$= |r(s_g, F_{s_\Delta}, a_g) + \gamma \mathbb{E}_{s_g', s_\Delta'} \max_{a_g' \in \mathcal{A}_g} \hat{Q}_k^0(s_g', F_{s_\Delta'}, a_g')$$

$$- r(s_g, F_{s_{\Delta'}}, a_g) - \gamma \mathbb{E}_{s_g', s_{\Delta'}'} \max_{a_g' \in \mathcal{A}_g} \hat{Q}_{k'}^0(s_g', F_{s_{\Delta'}'}, a_g')|$$

$$= |r(s_g, F_{s_\Delta}, a_g) - r(s_g, F_{s_{\Delta'}}, a_g)|$$

$$= \left|\frac{1}{k} \sum_{i \in \Delta} r_l(s_g, s_i) - \frac{1}{k'} \sum_{i \in \Delta'} r_l(s_g, s_i)\right|$$

$$= |\mathbb{E}_{s_l \sim F_{s_\Delta}} r_l(s_g, s_l) - \mathbb{E}_{s_l' \sim F_{s_{\Delta'}}} r_l(s_g, s_l')|$$

In the first and second equalities, we use the time evolution property of $\hat{Q}_k^1$ and $\hat{Q}_{k'}^1$ by applying the adapted Bellman operators $\hat{\mathcal{T}}_k$ and $\hat{\mathcal{T}}_{k'}$ to $\hat{Q}_k^0$ and $\hat{Q}_{k'}^0$, respectively, and expanding. In the third and fourth equalities, we note that $\hat{Q}_k^0(\cdot, \cdot, \cdot) = \hat{Q}_{k'}^0(\cdot, \cdot, \cdot) = 0$, and subtract the common 'global component' of the reward function.

Then, noting the general property that for any function $f : \mathcal{X} \to \mathcal{Y}$ for $|\mathcal{X}| < \infty$ we can write $f(x) = \sum_{y \in \mathcal{X}} f(y) \mathbb{1}\{y = x\}$, we have:

$$|\hat{Q}_k^1(s_g, F_{s_\Delta}, a_g) - \hat{Q}_{k'}^1(s_g, F_{s_{\Delta'}}, a_g)|$$

$$= \left|\mathbb{E}_{s_l \sim F_{s_\Delta}} \left[\sum_{z \in \mathcal{S}_l} r_l(s_g, z) \mathbb{1}\{s_l = z\}\right] - \mathbb{E}_{s_l' \sim F_{s_{\Delta'}}} \left[\sum_{z \in \mathcal{S}_l} r_l(s_g, z) \mathbb{1}\{s_l' = z\}\right]\right|$$

$$= |\sum_{z \in \mathcal{S}_l} r_l(s_g, z) \cdot (\mathbb{E}_{s_l \sim F_{s_\Delta}} \mathbb{1}\{s_l = z\} - \mathbb{E}_{s_l' \sim F_{s_{\Delta'}}} \mathbb{1}\{s_l' = z\})|$$

$$= |\sum_{z \in \mathcal{S}_l} r_l(s_g, z) \cdot (F_{s_\Delta}(z) - F_{s_{\Delta'}}(z))|$$

$$\leq |\max_{z \in \mathcal{S}_l} r_l(s_g, z)| \cdot \sum_{z \in \mathcal{S}_l} |F_{s_\Delta}(z) - F_{s_{\Delta'}}(z)|$$

$$\leq 2\|r_l(\cdot, \cdot)\|_\infty \cdot \mathrm{TV}(F_{s_\Delta}, F_{s_{\Delta'}})$$

The second equality follows from the linearity of expectations, and the third equality follows by noting that for any random variable $X \sim \mathcal{X}$, $\mathbb{E}_X \mathbb{1}[X = x] = \Pr[X = x]$. Then, the first inequality follows from an application of the triangle inequality and the Cauchy-Schwarz inequality, and the second inequality follows by the definition of total variation distance. Thus, when $T = 1$, $\hat{Q}$ is $(2\|r_l(\cdot, \cdot)\|_\infty)$-Lipschitz continuous with respect to total variation distance, proving the base case.

We now assume that for $T \leq t' \in \mathbb{N}$:

$$\left|\hat{Q}_k^T(s_g, F_{s_\Delta}, a_g) - \hat{Q}_{k'}^T(s_g, F_{s_{\Delta'}}, a_g)\right| \leq \left(\sum_{t=0}^{T-1} 2\gamma^t\right) \|r_l(\cdot, \cdot)\|_\infty \cdot \mathrm{TV}\left(F_{s_\Delta}, F_{s_{\Delta'}}\right)$$

Then, inductively we have:

$$|\hat{Q}_k^{T+1}(s_g, F_{s_\Delta}, a_g) - \hat{Q}_{k'}^{T+1}(s_g, F_{s_{\Delta'}}, a_g)|$$

$$\leq \left|\frac{1}{|\Delta|} \sum_{i \in \Delta} r_l(s_g, s_i) - \frac{1}{|\Delta'|} \sum_{i \in \Delta'} r_l(s_g, s_i)\right|$$

$$+ \gamma \left|\mathbb{E}_{s_g', s_\Delta'} \max_{a_g' \in \mathcal{A}_g} \hat{Q}_k^T(s_g', F_{s_\Delta'}, a_g') - \mathbb{E}_{s_g', s_{\Delta'}'} \max_{a_g' \in \mathcal{A}_g} \hat{Q}_{k'}^T(s_g', F_{s_{\Delta'}'}, a_g')\right|$$

$$\leq 2\|r_l(\cdot, \cdot)\|_\infty \cdot \mathrm{TV}\left(F_{s_\Delta}, F_{s_{\Delta'}}\right)$$

$$+ \gamma \left|\mathbb{E}_{s_g', s_\Delta'} \max_{a_g' \in \mathcal{A}_g} \hat{Q}_k^T(s_g', F_{s_\Delta'}, a_g') - \mathbb{E}_{s_g', s_{\Delta'}'} \max_{a_g' \in \mathcal{A}_g} \hat{Q}_{k'}^T(s_g', F_{s_{\Delta'}'}, a_g')\right|$$

In the first equality, we use the time evolution property of $\hat{Q}_k^{T+1}$ and $\hat{Q}_{k'}^{T+1}$ by applying the adapted-Bellman operators $\hat{\mathcal{T}}_k$ and $\hat{\mathcal{T}}_{k'}$ to $\hat{Q}_k^T$ and $\hat{Q}_{k'}^T$, respectively. We then expand and use the triangle inequality. In the first term of the second inequality, we use our Lipschitz bound from the base case. For the second term, we now rewrite the expectation over the states $s_g', s_\Delta', s_{\Delta'}'$ into an expectation over the joint transition probabilities $\mathcal{J}_k$ and $\mathcal{J}_{k'}$ from Theorem B.1.

Therefore, using the shorthand $\mathbb{E}_{(s_g', s_\Delta') \sim \mathcal{J}_k}$ to denote $\mathbb{E}_{(s_g', s_\Delta') \sim \mathcal{J}_k(\cdot, \cdot | s_g, a_g, s_\Delta)}$, we have:

$$
\begin{aligned}
&|\hat{Q}_k^{T+1}(s_g, F_{s_\Delta}, a_g) - \hat{Q}_{k'}^{T+1}(s_g, F_{s_{\Delta'}}, a_g)| \\
&\leq 2\|r_l(\cdot, \cdot)\|_\infty \cdot \mathrm{TV}(F_{s_\Delta}, F_{s_{\Delta'}}) \\
&\qquad + \gamma |\mathbb{E}_{(s_g', s_\Delta') \sim \mathcal{J}_k} \max_{a_g' \in \mathcal{A}_g} \hat{Q}_k^T(s_g', F_{s_\Delta'}, a_g') - \mathbb{E}_{(s_g', s_{\Delta'}') \sim \mathcal{J}_{k'}} \max_{a_g' \in \mathcal{A}_g} \hat{Q}_{k'}^T(s_g', F_{s_{\Delta'}'}, a_g')| \\
&\leq 2\|r_l(\cdot, \cdot)\|_\infty \cdot \mathrm{TV}(F_{s_\Delta}, F_{s_{\Delta'}}) \\
&\qquad + \gamma \max_{a_g' \in \mathcal{A}_g} |\mathbb{E}_{(s_g', s_\Delta') \sim \mathcal{J}_k} \hat{Q}_k^T(s_g', F_{s_\Delta'}, a_g') - \mathbb{E}_{(s_g', s_{\Delta'}') \sim \mathcal{J}_{k'}} \hat{Q}_{k'}^T(s_g', F_{s_{\Delta'}'}, a_g')| \\
&\leq 2\|r_l(\cdot, \cdot)\|_\infty \cdot \mathrm{TV}(F_{s_\Delta}, F_{s_{\Delta'}}) + \gamma \left( \sum_{\tau=0}^{T-1} 2\gamma^\tau \right) \|r_l(\cdot, \cdot)\|_\infty \cdot \mathrm{TV}(F_{s_\Delta}, F_{s_{\Delta'}}) \\
&= \left( \sum_{\tau=0}^{T} 2\gamma^\tau \right) \|r_l(\cdot, \cdot)\|_\infty \cdot \mathrm{TV}(F_{s_\Delta}, F_{s_{\Delta'}})
\end{aligned}
$$

In the first inequality, we rewrite the expectations over the states as the expectation over the joint transition probabilities. The second inequality then follows from Theorem B.9.

To apply it to Theorem B.9, we conflate the joint expectation over $(s_g, s_{\Delta \cup \Delta'})$ and reduce it back to the original form of its expectation. Finally, the third inequality follows from Theorem B.3.

Then, by the inductive hypothesis, the claim is proven. $\qquad\square$

**Lemma B.3.** For all $T \in \mathbb{N}$, for any $a_g, a_g' \in \mathcal{A}_g, s_g \in \mathcal{S}_g, s_\Delta \in \mathcal{S}_l^k$, and for all joint stochastic kernels $\mathcal{J}_k$ as defined in Theorem B.1, we have that $\mathbb{E}_{(s_g', s_\Delta') \sim \mathcal{J}_k(\cdot, \cdot | s_g, a_g, s_\Delta)} \hat{Q}_k^T(s_g', F_{s_\Delta'}, a_g')$ is $(\sum_{t=0}^{t-1} 2\gamma^t)\|r_l(\cdot, \cdot)\|_\infty$-Lipschitz continuous with respect to $F_{s_\Delta}$ in total variation distance:

$$
\begin{aligned}
&|\mathbb{E}_{(s_g', s_\Delta') \sim \mathcal{J}_k(\cdot, \cdot | s_g, a_g, s_\Delta)} \hat{Q}_k^T(s_g', F_{s_\Delta'}, a_g') - \mathbb{E}_{(s_g', s_{\Delta'}') \sim \mathcal{J}_{k'}(\cdot, \cdot | s_g, a_g, s_{\Delta'})} \hat{Q}_{k'}^T(s_g', F_{s_{\Delta'}'}, a_g')| \\
&\qquad\qquad\qquad \leq \left( \sum_{\tau=0}^{T-1} 2\gamma^\tau \right) \|r_l(\cdot, \cdot)\|_\infty \cdot \mathrm{TV}\left(F_{s_\Delta}, F_{s_{\Delta'}}\right)
\end{aligned}
$$

*Proof.* We prove this inductively. At $T = 0$, the statement is true since $\hat{Q}_k^0(\cdot, \cdot, \cdot) = \hat{Q}_{k'}^0(\cdot, \cdot, \cdot) = 0$ and $\mathrm{TV}(\cdot, \cdot) \geq 0$. For $T = 1$, applying the adapted Bellman operator yields:

$$
\begin{aligned}
&|\mathbb{E}_{(s_g', s_\Delta') \sim \mathcal{J}_k(\cdot, \cdot | s_g, a_g, s_\Delta)} \hat{Q}_k^1(s_g', F_{s_\Delta'}, a_g') - \mathbb{E}_{(s_g', s_{\Delta'}') \sim \mathcal{J}_{k'}(\cdot, \cdot | s_g, a_g, s_{\Delta'})} \hat{Q}_{k'}^1(s_g', F_{s_{\Delta'}'}, a_g')| \\
&= \left| \mathbb{E}_{(s_g', s_{\Delta \cup \Delta'}') \sim \mathcal{J}_{|\Delta \cup \Delta'|}(\cdot, \cdot | s_g, a_g, s_{\Delta \cup \Delta'})} \left[ \frac{1}{|\Delta|} \sum_{i \in \Delta} r_l(s_g', s_i') - \frac{1}{|\Delta'|} \sum_{i \in \Delta'} r_l(s_g', s_i') \right] \right| \\
&= \left| \mathbb{E}_{(s_g', s_{\Delta \cup \Delta'}') \sim \mathcal{J}_{|\Delta \cup \Delta'|}(\cdot, \cdot | s_g, a_g, s_{\Delta \cup \Delta'})} \left[ \sum_{z \in \mathcal{S}_l} r_l(s_g', z) \cdot (F_{s_\Delta'}(z) - F_{s_{\Delta'}'}(z)) \right] \right|
\end{aligned}
$$

Similarly to Theorem B.2, we implicitly write the result as an expectation over the reward functions and use the general property that for any function $f : \mathcal{X} \to \mathcal{Y}$ for $|\mathcal{X}| < \infty$, we can write $f(x) = \sum_{y \in \mathcal{X}} f(y) \mathbb{1}\{y = x\}$. Then, taking the expectation over the indicator variable yields the second equality. As a shorthand, let $\mathfrak{D}$ denote the distribution of $s_g' \sim$

$\sum_{s'_{\Delta\cup\Delta'}\in\mathcal{S}_l^{|\Delta\cup\Delta'|}}\mathcal{J}_{|\Delta\cup\Delta|}(\cdot,s'_{\Delta\cup\Delta'}|s_g,a_g,s_{\Delta\cup\Delta'})$. Then, by the law of total expectation,

$$|\mathbb{E}_{(s'_g,s'_\Delta)\sim\mathcal{J}_k(\cdot,\cdot|s_g,a_g,s_\Delta)}\hat{Q}_k^1(s'_g,F_{s'_\Delta},a'_g) - \mathbb{E}_{(s'_g,s'_{\Delta'})\sim\mathcal{J}_{k'}(\cdot,\cdot|s_g,a_g,s_{\Delta'})}\hat{Q}_{k'}^1(s'_g,F_{s'_{\Delta'}},a'_g)|$$

$$= |\mathbb{E}_{s'_g\sim\mathfrak{D}}\sum_{z\in\mathcal{S}_l}r_l(s'_g,z)\mathbb{E}_{s'_{\Delta\cup\Delta'}\sim\mathcal{J}_{|\Delta\cup\Delta'|}}(\cdot|s'_g,s_g,a_g,s_{\Delta\cup\Delta'})(F_{s'_\Delta}(z) - F_{s'_{\Delta'}}(z))|$$

$$\leq \|r_l(\cdot,\cdot)\|_\infty\cdot\mathbb{E}_{s'_g\sim\mathfrak{D}}\sum_{z\in\mathcal{S}_l}|\mathbb{E}_{s'_{\Delta\cup\Delta'}\sim\mathcal{J}_{|\Delta\cup\Delta'|}}(\cdot|s'_g,s_g,a_g,s_{\Delta\cup\Delta'})(F_{s'_\Delta}(z) - F_{s'_{\Delta'}}(z))|$$

$$\leq 2\|r_l(\cdot,\cdot)\|_\infty\cdot\mathbb{E}_{s'_g\sim\mathfrak{D}}\,\mathrm{TV}(\mathbb{E}_{s'_{\Delta\cup\Delta'}|s'_g}F_{s'_\Delta},\mathbb{E}_{s'_{\Delta\cup\Delta'}|s'_g}F_{s'_{\Delta'}})$$

$$\leq 2\|r_l(\cdot,\cdot)\|_\infty\cdot\mathrm{TV}(F_{s_\Delta},F_{s_{\Delta'}})$$

In the ensuing inequalities, we first use Jensen's inequality and the triangle inequality to pull out $\mathbb{E}_{s'_g}\sum_{z\in\mathcal{S}_l}$ from the absolute value, and then use Cauchy-Schwarz to further factor $\|r_l(\cdot,\cdot)\|_\infty$. The second inequality follows from Theorem B.5 and does not have a dependence on $s'_g$ thus eliminating $\mathbb{E}_{s'_g}$ and proving the base case.

We now assume that for $T\leq t'\in\mathbb{N}$, for all joint stochastic kernels $\mathcal{J}_k$ and $\mathcal{J}_{k'}$, and for all $a'_g\in\mathcal{A}_g$:

$$|\mathbb{E}_{(s'_g,s'_\Delta)\sim\mathcal{J}_k(\cdot,\cdot|s_g,a_g,s_\Delta)}\hat{Q}_k^T(s'_g,F_{s'_\Delta},a'_g) - \mathbb{E}_{(s'_g,s'_{\Delta'})\sim\mathcal{J}_{k'}(\cdot,\cdot|s_g,a_g,s_{\Delta'})}\hat{Q}_{k'}^T(s'_g,F_{s'_{\Delta'}},a'_g)|$$

$$\leq \left(\sum_{t=0}^{T-1}2\gamma^t\right)\|r_l(\cdot,\cdot)\|_\infty\cdot\mathrm{TV}(F_{s_\Delta},F_{s_{\Delta'}})$$

For the remainder of the proof, we adopt the shorthand $\mathbb{E}_{(s'_g,s'_\Delta)\sim\mathcal{J}}$ to denote $\mathbb{E}_{(s'_g,s'_\Delta)\sim\mathcal{J}_{|\Delta|}(\cdot,\cdot|s_g,a_g,s_\Delta)}$, and $\mathbb{E}_{(s''_g,s''_\Delta)\sim\mathcal{J}}$ to denote $\mathbb{E}_{(s''_g,s''_\Delta)\sim\mathcal{J}_{|\Delta|}(\cdot,\cdot|s'_g,a'_g,s'_\Delta)}$.

Then, inductively, we have:

$$|\mathbb{E}_{(s'_g,s'_\Delta)\sim\mathcal{J}}\hat{Q}_k^{T+1}(s'_g,F_{s'_\Delta},a'_g) - \mathbb{E}_{(s'_g,s'_{\Delta'})\sim\mathcal{J}}\hat{Q}_{k'}^{T+1}(s'_g,F_{s'_{\Delta'}},a'_g)|$$

$$= |\mathbb{E}_{(s'_g,s'_{\Delta\cup\Delta'})\sim\mathcal{J}}[r(s'_g,s'_\Delta,a'_g) - r(s'_g,s'_{\Delta'},a'_g)$$

$$+ \gamma\mathbb{E}_{(s''_g,s''_{\Delta\cup\Delta'})\sim\mathcal{J}}[\max_{a''_g\in\mathcal{A}_g}\hat{Q}_k^T(s''_g,F_{s''_\Delta},a''_g) - \max_{a''_g\in\mathcal{A}_g}\hat{Q}_{k'}^T(s''_g,F_{s''_{\Delta'}},a''_g)]]|$$

$$\leq 2\|r_l(\cdot,\cdot)\|_\infty\cdot\mathrm{TV}(F_{s_\Delta},F_{s_{\Delta'}})$$

$$+ \gamma|\mathbb{E}_{(s'_g,s'_{\Delta\cup\Delta'})\sim\mathcal{J}}[\mathbb{E}_{(s''_g,s''_{\Delta\cup\Delta'})\sim\mathcal{J}}[\max_{a''_g\in\mathcal{A}_g}\hat{Q}_k^T(s''_g,F_{s''_\Delta},a''_g) - \max_{a''_g\in\mathcal{A}_g}\hat{Q}_{k'}^T(s''_g,F_{s''_{\Delta'}},a''_g)]]|$$

Here, we expand out $\hat{Q}_k^{T+1}$ and $\hat{Q}_{k'}^{T+1}$ using the adapted Bellman operator. In the ensuing inequality, we apply the triangle inequality and bound the first term using the base case. Then, note that

$$\mathbb{E}_{(s'_g,s'_{\Delta\cup\Delta'})\sim\mathcal{J}(\cdot,\cdot|s_g,a_g,s_{\Delta\cup\Delta'})}\mathbb{E}_{(s''_g,s''_{\Delta\cup\Delta'})\sim\mathcal{J}(\cdot,\cdot|s'_g,a'_g,s'_{\Delta\cup\Delta'})}\max_{a''_g\in\mathcal{A}_g}\hat{Q}_k^T(s''_g,F_{s''_\Delta},a''_g)$$

is, for some stochastic function $\mathcal{J}'_{|\Delta\cup\Delta'|}$, equal to

$$\mathbb{E}_{(s''_g,s''_{\Delta\cup\Delta'})\sim\mathcal{J}'_{|\Delta\cup\Delta'|}(\cdot,\cdot|s_g,a_g,s_{\Delta\cup\Delta'})}\max_{a''_g\in\mathcal{A}_g}\hat{Q}_k^T(s''_g,F_{s''_\Delta},a''_g),$$

where $\mathcal{J}'$ is implicitly a function of $a'_g$ which is fixed from the beginning.

In the special case where $a_g = a'_g$, we can derive an explicit form of $\mathcal{J}'$ which we show in Theorem B.11. As a shorthand, we denote $\mathbb{E}_{(s''_g,s''_{\Delta\cup\Delta'})\sim\mathcal{J}'_{|\Delta\cup\Delta'|}(\cdot,\cdot|s_g,a_g,s_{\Delta\cup\Delta'})}$ by $\mathbb{E}_{(s''_g,s''_{\Delta\cup\Delta'})\sim\mathcal{J}'}$.

Therefore,

$$|\mathbb{E}_{(s'_g, s'_\Delta) \sim \mathcal{J}} \hat{Q}_k^{T+1}(s'_g, F_{s'_\Delta}, a'_g) - \mathbb{E}_{(s'_g, s'_{\Delta'}) \sim \mathcal{J}} \hat{Q}_{k'}^{T+1}(s'_g, F_{s'_{\Delta'}}, a'_g)|$$

$$\leq 2\|r_l(\cdot, \cdot)\|_\infty \cdot \mathrm{TV}(F_{s_\Delta}, F_{s_{\Delta'}}) + \gamma |\mathbb{E}_{(s''_g, s''_{\Delta \cup \Delta'}) \sim \mathcal{J}'} \max_{a''_g \in \mathcal{A}_g} \hat{Q}_k^T(s''_g, F_{s''_\Delta}, a''_g)$$

$$- \mathbb{E}_{(s''_g, s''_{\Delta \cup \Delta'}) \sim \mathcal{J}'} \max_{a''_g \in \mathcal{A}_g} \hat{Q}_{k'}^T(s''_g, F_{s''_{\Delta'}}, a''_g)|$$

$$\leq 2\|r_l(\cdot, \cdot)\|_\infty \cdot \mathrm{TV}(F_{s_\Delta}, F_{s_{\Delta'}}) + \gamma \max_{a''_g \in \mathcal{A}_g} |\mathbb{E}_{(s''_g, s''_{\Delta \cup \Delta'}) \sim \mathcal{J}'} \hat{Q}_k^T(s''_g, F_{s''_\Delta}, a''_g)$$

$$- \mathbb{E}_{(s''_g, s''_{\Delta \cup \Delta'}) \sim \mathcal{J}'} \hat{Q}_{k'}^T(s''_g, F_{s''_{\Delta'}}, a''_g)|$$

$$\leq 2\|r_l(\cdot, \cdot)\|_\infty \cdot \mathrm{TV}(F_{s_\Delta}, F_{s_{\Delta'}}) + \gamma \left( \sum_{t=0}^{T-1} 2\gamma^t \right) \|r_l(\cdot, \cdot)\|_\infty \cdot \mathrm{TV}(F_{s_\Delta}, F_{s_{\Delta'}})$$

$$= \left( \sum_{t=0}^{T} 2\gamma^t \right) \|r_l(\cdot, \cdot)\|_\infty \cdot \mathrm{TV}(F_{s_\Delta}, F_{s_{\Delta'}})$$

The second inequality follows from Theorem B.9 where we set the joint stochastic kernel to be $\mathcal{J}'_{|\Delta \cup \Delta'|}$. In the ensuing lines, we concentrate the expectation towards the relevant terms and use the induction assumption for the transition probability functions $\mathcal{J}'_k$ and $\mathcal{J}'_{k'}$. This proves the lemma. $\square$

**Remark B.4.** Given a joint transition probability function $\mathcal{J}_{|\Delta \cup \Delta'|}$ as defined in Theorem B.1, we can recover the transition function for a single agent $i \in \Delta \cup \Delta'$ given by $\mathcal{J}_1$ using the law of total probability and the conditional independence between $s_i$ and $s_g \cup s_{[n] \setminus i}$ in Equation (27). This characterization is crucial in Theorem B.5 and Theorem B.6.

$$\mathcal{J}_1(\cdot | s'_g, s_g, a_g, s_i) = \sum_{s'_{\Delta \cup \Delta' \setminus i} \sim \mathcal{S}_l^{|\Delta \cup \Delta'| - 1}} \mathcal{J}_{|\Delta \cup \Delta'|}(s'_{\Delta \cup \Delta' \setminus i}, s'_i | s'_g, s_g, a_g, s_{\Delta \cup \Delta'}) \quad (27)$$

**Lemma B.5.** *Given a joint transition probability $\mathcal{J}_{|\Delta \cup \Delta'|}$ as defined in Theorem B.1,*

$$\mathrm{TV}(\mathbb{E}_{s'_{\Delta \cup \Delta'} \sim \mathcal{J}_{|\Delta \cup \Delta'|}(\cdot | s'_g, s_g, a_g, s_{\Delta \cup \Delta'})} F_{s'_\Delta}, \mathbb{E}_{s'_{\Delta \cup \Delta'} \sim \mathcal{J}_{|\Delta \cup \Delta'|}(\cdot | s'_g, s_g, a_g, s_{\Delta \cup \Delta'})} F_{s'_{\Delta'}}) \leq \mathrm{TV}(F_{s_\Delta}, F_{s_{\Delta'}})$$

*Proof.* Note that from Theorem B.6:

$$\mathbb{E}_{s'_{\Delta \cup \Delta'} \sim \mathcal{J}_{|\Delta \cup \Delta'|}(\cdot, \cdot | s'_g, s_g, a_g, s_{\Delta \cup \Delta'})} F_{s'_\Delta} = \mathbb{E}_{s'_\Delta \sim \mathcal{J}_{|\Delta|}(\cdot, \cdot | s'_g, s_g, a_g, s_\Delta)} F_{s'_\Delta}$$

$$= \mathcal{J}_1(\cdot | s_g(t+1), s_g(t), a_g(t), \cdot) F_{s_\Delta}$$

Then, by expanding the TV distance in $\ell_1$-norm:

$$\mathrm{TV}(\mathbb{E}_{s'_{\Delta \cup \Delta'} \sim \mathcal{J}_{|\Delta \cup \Delta'|}(\cdot | s'_g, s_g, a_g, s_{\Delta \cup \Delta'})} F_{s'_\Delta}, \mathbb{E}_{s'_{\Delta \cup \Delta'} \sim \mathcal{J}_{|\Delta \cup \Delta'|}(\cdot | s'_g, s_g, a_g, s_{\Delta \cup \Delta'})} F_{s'_{\Delta'}})$$

$$= \frac{1}{2} \|\mathcal{J}_1(\cdot | s_g(t+1), s_g(t), a_g(t), \cdot) F_{s_\Delta} - \mathcal{J}_1(\cdot | s_g(t+1), s_g(t), a_g(t), \cdot) F_{s_{\Delta'}}\|_1$$

$$\leq \|\mathcal{J}_1(\cdot | s_g(t+1), s_g(t), a_g(t), \cdot)\|_1 \cdot \frac{1}{2} \|F_{s_\Delta} - F_{s_{\Delta'}}\|_1$$

$$\leq \frac{1}{2} \|F_{s_\Delta} - F_{s_{\Delta'}}\|_1$$

$$= \mathrm{TV}(F_{s_\Delta}, F_{s_{\Delta'}})$$

In the first inequality, we factorize $\|\mathcal{J}_1(\cdot | s_g(t+1), s_g(t), a_g(t))\|_1$ from the $\ell_1$-normed expression by the sub-multiplicativity of the matrix norm. Finally, since $\mathcal{J}_1$ is a column-stochastic matrix, we bound its norm by 1 to recover the total variation distance between $F_{s_\Delta}$ and $F_{s_{\Delta'}}$. $\square$

**Lemma B.6.** *Given the joint transition probability $\mathcal{J}_k$ from Theorem B.1:*

$$\mathbb{E}_{s_{\Delta\cup\Delta'}(t+1)\sim\mathcal{J}_{|\Delta\cup\Delta'|}(\cdot|s_g(t+1),s_g(t),a_g(t),s_{\Delta\cup\Delta'}(t))}F_{s_\Delta(t+1)} := \mathcal{J}_1(\cdot|s_g(t+1),s_g(t),a_g(t),\cdot)F_{s_\Delta}(t)$$

*Proof.* First, observe that for all $x \in \mathcal{S}_l$:

$$\mathbb{E}_{s_{\Delta\cup\Delta'}(t+1)\sim\mathcal{J}_{|\Delta\cup\Delta'|}(\cdot|s_g(t+1),s_g(t),a_g(t),s_{\Delta\cup\Delta'}(t))}F_{s_\Delta(t+1)}(x)$$

$$= \frac{1}{|\Delta|}\sum_{i\in\Delta}\mathbb{E}_{s_{\Delta\cup\Delta'}(t+1)\sim\mathcal{J}_{|\Delta\cup\Delta'|}(\cdot|s_g(t+1),s_g(t),a_g(t),s_{\Delta\cup\Delta'}(t))}\mathbb{1}(s_i(t+1)=x)$$

$$= \frac{1}{|\Delta|}\sum_{i\in\Delta}\Pr[s_i(t+1)=x|s_g(t+1),s_g(t),a_g(t),s_{\Delta\cup\Delta'}(t))]$$

$$= \frac{1}{|\Delta|}\sum_{i\in\Delta}\Pr[s_i(t+1)=x|s_g(t+1),s_g(t),a_g(t),s_i(t))]$$

$$= \frac{1}{|\Delta|}\sum_{i\in\Delta}\mathcal{J}_1(x|s_g(t+1),s_g(t),a_g(t),s_i(t))$$

In the first line, we expand on the definition of $F_{s_\Delta(t+1)}(x)$. Finally, we note that $s_i(t+1)$ is conditionally independent to $s_{\Delta\cup\Delta'\setminus i}$, which yields the equality above. Then, aggregating across every entry $x \in \mathcal{S}_l$,

$$\mathbb{E}_{s_{\Delta\cup\Delta'}(t+1)\sim\mathcal{J}_{|\Delta\cup\Delta'|}(\cdot|s_g(t+1),s_g(t),a_g(t),s_{\Delta\cup\Delta'}(t))}F_{s_\Delta(t+1)}$$

$$= \frac{1}{|\Delta|}\sum_{i\in\Delta}\mathcal{J}_1(\cdot|s_g(t+1),s_g(t),a_g(t),\cdot)\vec{\mathbb{1}}_{s_i(t)}$$

$$= \mathcal{J}_1(\cdot|s_g(t+1),s_g(t),a_g(t),\cdot)F_{s_\Delta}$$

Notably, every $x$ corresponds to a choice of rows in $\mathcal{J}_1(\cdot|s_g(t+1),s_g(t),a_g(t),\cdot)$ and every choice of $s_i(t)$ corresponds to a choice of columns in $\mathcal{J}_1(\cdot|s_g(t+1),s_g(t),a_g(t),\cdot)$, making $\mathcal{J}_1(\cdot|s_g(t+1),s_g(t),a_g(t),\cdot)$ column-stochastic. This yields the claim. $\square$

**Lemma B.7.** The total variation distance between the expected empirical distribution of $s_\Delta(t+1)$ and $s_{\Delta'}(t+1)$ is linearly bounded by the total variation distance of the empirical distributions of $s_\Delta(t)$ and $s_{\Delta'}(t)$, for $\Delta, \Delta' \subseteq [n]$:

$$\mathrm{TV}\left(\mathbb{E}_{\substack{s_i(t+1)\sim P_l(\cdot|s_i(t),s_g(t)),\\\forall i\in\Delta}}F_{s_\Delta(t+1)}, \mathbb{E}_{\substack{s_i(t+1)\sim P_l(\cdot|s_i(t),s_g(t)),\\\forall i\in\Delta'}}F_{s_{\Delta'}(t+1)}\right) \le \mathrm{TV}\left(F_{s_\Delta(t)}, F_{s_{\Delta'}(t)}\right)$$

*Proof.* We expand the total variation distance measure in $\ell_1$-norm and utilize the result from Theorem B.10 that $\mathbb{E}_{\substack{s_i(t+1)\sim P_l(\cdot|s_i(t),s_g(t))\\\forall i\in\Delta}}F_{s_\Delta(t+1)} = P_l(\cdot|s_g(t))F_{s_\Delta(t)}$ as follows:

$$\mathrm{TV}\left(\mathbb{E}_{\substack{s_i(t+1)\sim P_l(\cdot|s_i(t),s_g(t))\\\forall i\in\Delta}}F_{s_\Delta(t+1)}, \mathbb{E}_{\substack{s_i(t+1)\sim P_l(\cdot|s_i(t),s_g(t))\\\forall i\in\Delta'}}F_{s_{\Delta'}(t+1)}\right)$$

$$= \frac{1}{2}\left\|\mathbb{E}_{\substack{s_i(t+1)\sim P_l(\cdot|s_i(t),s_g(t))\\\forall i\in\Delta}}F_{s_\Delta(t+1)} - \mathbb{E}_{\substack{s_i(t+1)\sim P_l(\cdot|s_i(t),s_g(t))\\\forall i\in\Delta'}}F_{s_{\Delta'}(t+1)}\right\|_1$$

$$= \frac{1}{2}\left\|P_l(\cdot|\cdot,s_g(t))F_{s_\Delta(t)} - P_l(\cdot|\cdot,s_g(t))F_{s_{\Delta'}(t)}\right\|_1$$

$$\le \|P_l(\cdot|\cdot,s_g(t))\|_1 \cdot \frac{1}{2}|F_{s_\Delta(t)} - F_{s_{\Delta'}(t)}|_1$$

$$= \|P_l(\cdot|\cdot,s_g(t))\|_1 \cdot \mathrm{TV}(F_{s_\Delta(t)}, F_{s_{\Delta'}(t)})$$

In the last line, we recover the total variation distance from the $\ell_1$ norm. Finally, by the column stochasticity of $P_l(\cdot|\cdot,s_g)$, we have that $\|P_l(\cdot|\cdot,s_g)\|_1 \le 1$, which then implies

$$\mathrm{TV}\left(\mathbb{E}_{\substack{s_i(t+1)\sim P_l(\cdot|s_i(t),s_g(t))\\\forall i\in\Delta}}F_{s_\Delta(t+1)}, \mathbb{E}_{\substack{s_i(t+1)\sim P_l(\cdot|s_i(t),s_g(t))\\\forall i\in\Delta'}}F_{s_{\Delta'}(t+1)}\right) \le \mathrm{TV}(F_{s_\Delta(t)}, F_{s_{\Delta'}(t)})$$

This proves the lemma. $\square$

**Remark B.8.** Theorem B.7 can be viewed as an irreducibility and aperiodicity result on the finite-state Markov chain whose state space is given by $\mathcal{S} = \mathcal{S}_g \times \mathcal{S}_l^n$. Let $\{s_t\}_{t \in \mathbb{N}}$ denote the sequence of states visited by this Markov chain where the transitions are induced by the transition functions $P_g, P_l$. Through this, Theorem B.7 describes an ergodic behavior of the Markov chain.

**Lemma B.9.** The absolute difference between the expected maximums between $\hat{Q}_k$ and $\hat{Q}_{k'}$ is atmost the maximum of the absolute difference between $\hat{Q}_k$ and $\hat{Q}_{k'}$, where the expectations are taken over any joint distributions of states $\mathcal{J}$, and the maximums are taken over the actions.

$$|\mathbb{E}_{(s'_g, s'_{\Delta \cup \Delta'}) \sim \mathcal{J}_{|\Delta \cup \Delta'|}(\cdot, \cdot|s_g, a_g, s_{\Delta \cup \Delta'})}[\max_{a'_g \in \mathcal{A}_g} \hat{Q}_k^T(s'_g, F_{s'_\Delta}, a'_g) - \max_{a'_g \in \mathcal{A}_g} \hat{Q}_{k'}^T(s'_g, F_{s'_{\Delta'}}, a'_g)]|$$

$$\leq \max_{a'_g \in \mathcal{A}_g} |\mathbb{E}_{(s'_g, s'_{\Delta \cup \Delta'}) \sim \mathcal{J}_{|\Delta \cup \Delta'|}(\cdot, \cdot|s_g, a_g, s_{\Delta \cup \Delta'})}[\hat{Q}_k^T(s'_g, F_{s'_\Delta}, a'_g) - \hat{Q}_{k'}^T(s'_g, F_{s'_{\Delta'}}, a'_g)]|$$

*Proof.*
$$a_g^* := \arg \max_{a'_g \in \mathcal{A}_g} \hat{Q}_k^T(s'_g, F_{s'_\Delta}, a'_g), \quad \tilde{a}_g^* := \arg \max_{a'_g \in \mathcal{A}_g} \hat{Q}_{k'}^T(s'_g, F_{s'_{\Delta'}}, a'_g)$$

For the remainder of this proof, we adopt the shorthand $\mathbb{E}_{s'_g, s'_{\Delta \cup \Delta'}}$ to refer to $\mathbb{E}_{(s'_g, s'_{\Delta \cup \Delta'}) \sim \mathcal{J}_{|\Delta \cup \Delta'|}(\cdot, \cdot|s_g, a_g, s_{\Delta \cup \Delta'})}$.

Then, if $\mathbb{E}_{s'_g, s'_{\Delta \cup \Delta'}} \max_{a'_g \in \mathcal{A}_g} \hat{Q}_k^T(s'_g, F_{s'_\Delta}, a'_g) - \mathbb{E}_{s'_g, s'_{\Delta \cup \Delta'}} \max_{a'_g \in \mathcal{A}_g} \hat{Q}_{k'}^T(s'_g, F_{s'_{\Delta'}}, a'_g) > 0$, we have:

$$|\mathbb{E}_{s'_g, s'_{\Delta \cup \Delta'}} \max_{a'_g \in \mathcal{A}_g} \hat{Q}_k^T(s'_g, F_{s'_\Delta}, a'_g) - \mathbb{E}_{s'_g, s'_{\Delta \cup \Delta'}} \max_{a'_g \in \mathcal{A}_g} \hat{Q}_{k'}^T(s'_g, F_{s'_{\Delta'}}, a'_g)|$$

$$= \mathbb{E}_{s'_g, s'_{\Delta \cup \Delta'}} \hat{Q}_k^T(s'_g, F_{s'_\Delta}, a_g^*) - \mathbb{E}_{s'_g, s'_{\Delta \cup \Delta'}} \hat{Q}_{k'}^T(s'_g, F_{s'_{\Delta'}}, \tilde{a}_g^*)$$

$$\leq \mathbb{E}_{s'_g, s'_{\Delta \cup \Delta'}} \hat{Q}_k^T(s'_g, F_{s'_\Delta}, a_g^*) - \mathbb{E}_{s'_g, s'_{\Delta \cup \Delta'}} \hat{Q}_{k'}^T(s'_g, F_{s'_{\Delta'}}, a_g^*)$$

$$\leq \max_{a'_g \in \mathcal{A}_g} |\mathbb{E}_{s'_g, s'_{\Delta \cup \Delta'}} \hat{Q}_k^T(s'_g, F_{s'_\Delta}, a'_g) - \mathbb{E}_{s'_g, s'_{\Delta \cup \Delta'}} \hat{Q}_{k'}^T(s'_g, F_{s'_{\Delta'}}, a'_g)|$$

Similarly, if $\mathbb{E}_{s'_g, s'_{\Delta \cup \Delta'}} \max_{a'_g \in \mathcal{A}_g} \hat{Q}_k^T(s'_g, F_{s'_\Delta}, a'_g) - \mathbb{E}_{s'_g, s'_{\Delta \cup \Delta'}} \max_{a'_g \in \mathcal{A}_g} \hat{Q}_{k'}^T(s'_g, F_{s'_{\Delta'}}, a'_g) < 0$, an analogous argument by replacing $a_g^*$ with $\tilde{a}_g^*$ yields an identical bound. $\qquad \square$

**Lemma B.10.** *For all $t \in \mathbb{N}$ and $\Delta \subseteq [n]$,*
$$\mathbb{E}_{\substack{s_i(t+1) \sim P_l(\cdot|s_i(t), s_g(t)) \\ \forall i \in \Delta}}[F_{s_\Delta(t+1)}] = P_l(\cdot|\cdot, s_g(t)) F_{s_\Delta(t)}$$

*Proof.* For all $x \in \mathcal{S}_l$:

$$\mathbb{E}_{\substack{s_i(t+1) \sim P_l(\cdot|s_i(t), s_g(t)) \\ \forall i \in \Delta}}[F_{s_\Delta(t+1)}(x)] := \frac{1}{|\Delta|} \sum_{i \in \Delta} \mathbb{E}_{s_i(t+1) \sim P_l(s_i(t), s_g(t))}[\mathbb{1}(s_i(t+1) = x)]$$

$$= \frac{1}{|\Delta|} \sum_{i \in \Delta} \Pr[s_i(t+1) = x | s_i(t+1) \sim P_l(\cdot|s_i(t), s_g(t))]$$

$$= \frac{1}{|\Delta|} \sum_{i \in \Delta} P_l(x|s_i(t), s_g(t))$$

In the first line, we are writing out the definition of $F_{s_\Delta(t+1)}(x)$ and using the conditional independence in the evolutions of $\Delta \setminus i$ and $i$. In the second line, we use the fact that for any random variable $X \in \mathcal{X}$, $\mathbb{E}_X \mathbb{1}[X = x] = \Pr[X = x]$. In line 3, we observe that the above probability can be written as an entry of the local transition matrix $P_l$. Then, aggregating across every entry $x \in \mathcal{S}_l$, we have that:

$$\mathbb{E}_{\substack{s_i(t+1) \sim P_l(\cdot|s_i(t), s_g(t)) \\ \forall i \in \Delta}}[F_{s_\Delta(t+1)}] = \frac{1}{|\Delta|} \sum_{i \in \Delta} P_l(\cdot|s_i(t), s_g(t))$$

$$= \frac{1}{|\Delta|} \sum_{i \in \Delta} P_l(\cdot|\cdot, s_g(t)) \vec{\mathbb{1}}_{s_i(t)} =: P_l(\cdot|\cdot, s_g(t)) F_{s_\Delta(t)}$$

Here, $\vec{\mathbb{1}}_{s_i(t)} \in \{0,1\}^{|\mathcal{S}_l|}$ such that $\vec{\mathbb{1}}_{s_i(t)}$ is 1 at the index corresponding to $s_i(t)$, and is 0 everywhere else. The last equality follows since $P_l(\cdot|\cdot, s_g(t))$ is a column-stochastic matrix which yields that $P_l(\cdot|\cdot, s_g(t))\vec{\mathbb{1}}_{s_i(t)} = P_l(\cdot|s_i(t), s_g(t))$, thus proving the lemma. $\qquad\square$

**Lemma B.11.** *For any joint transition probability function on $s_g, s_\Delta$, where $|\Delta| = k$, given by $\mathcal{J}_k : \mathcal{S}_g \times \mathcal{S}_l^{|\Delta|} \times \mathcal{S}_g \times \mathcal{A}_g \times \mathcal{S}_l^{|\Delta|} \to [0,1]$, we have:*

$$\mathbb{E}_{(s_g', s_\Delta') \sim \mathcal{J}_k(\cdot,\cdot|s_g, a_g, s_\Delta)} \left[ \mathbb{E}_{(s_g'', s_\Delta'') \sim \mathcal{J}_k(\cdot,\cdot|s_g', a_g, s_\Delta')} \max_{a_g'' \in \mathcal{A}_g} \hat{Q}_k^T(s_g'', F_{s_\Delta''}, a_g'') \right]$$

$$= \mathbb{E}_{(s_g'', s_\Delta'') \sim \mathcal{J}_k^2(\cdot,\cdot|s_g, a_g, s_\Delta)} \max_{a_g'' \in \mathcal{A}_g} \hat{Q}_k^T(s_g'', F_{s_\Delta''}, a_g'')$$

*Proof.* We start by expanding the expectations:

$$\mathbb{E}_{(s_g', s_\Delta') \sim \mathcal{J}_k(\cdot,\cdot|s_g, a_g, s_\Delta)} \left[ \mathbb{E}_{(s_g'', s_\Delta'') \sim \mathcal{J}_k(\cdot,\cdot|s_g', a_g, s_\Delta')} \max_{a_g' \in \mathcal{A}_g} \hat{Q}_k^T(s_g'', F_{s_\Delta''}, a_g') \right]$$

$$= \sum_{(s_g', s_\Delta') \in \mathcal{S}_g \times \mathcal{S}_l^{|\Delta|}} \sum_{(s_g'', s_\Delta'') \in \mathcal{S}_g \times \mathcal{S}_l^{|\Delta|}} \mathcal{J}_k[s_g', s_\Delta', s_g, a_g, s_\Delta] \mathcal{J}_k[s_g'', s_\Delta'', s_g', a_g, s_\Delta'] \max_{a_g' \in \mathcal{A}_g} \hat{Q}_k^T(s_g'', F_{s_\Delta''}, a_g')$$

$$= \sum_{(s_g'', s_\Delta'') \in \mathcal{S}_g \times \mathcal{S}_l^{|\Delta|}} \mathcal{J}_k^2[s_g'', s_\Delta'', s_g, a_g, s_\Delta] \max_{a_g' \in \mathcal{A}_g} \hat{Q}_k^T(s_g'', F_{s_\Delta''}, a_g')$$

$$= \mathbb{E}_{(s_g'', s_\Delta'') \sim \mathcal{J}_k^2(\cdot,\cdot|s_g, a_g, s_\Delta)} \max_{a_g' \in \mathcal{A}_g} \hat{Q}_k^T(s_g'', F_{s_\Delta''}, a_g')$$

The right-stochasticity of $\mathcal{J}_k$ implies the right-stochasticity of $\mathcal{J}_k^2$. Further, observe that $\mathcal{J}_k[s_g', s_\Delta', s_g, a_g, s_\Delta]\mathcal{J}_k[s_g'', s_\Delta'', s_g', a_g, s_\Delta']$ denotes the probability of the transitions $(s_g, s_\Delta) \to (s_g', s_\Delta') \to (s_g'', s_\Delta'')$ with actions $a_g$ at each step, where the joint state evolution is governed by $\mathcal{J}_k$.

Thus, $\sum_{(s_g', s_\Delta') \in \mathcal{S}_g \times \mathcal{S}_l^{|\Delta|}} \mathcal{J}_k[s_g', s_\Delta', s_g, a_g, s_\Delta]\mathcal{J}_k[s_g'', s_\Delta'', s_g', a_g, s_g']$ is the stochastic probability function corresponding to the two-step evolution of the joint states from $(s_g, s_\Delta)$ to $(s_g'', s_\Delta'')$ under the action $a_g$, which is equivalent to $\mathcal{J}_k^2[s_g'', s_\Delta'', s_g, a_g, s_\Delta]$. In the third equality, we recover the definition of the expectation, where the joint probabilities are taken over $\mathcal{J}_k^2$. $\qquad\square$

The following lemma bounds the average difference between $\hat{Q}_k^T$ (across every choice of $\Delta \in \binom{[n]}{k}$) and $Q^*$ and shows that the difference decays to 0 as $T \to \infty$.

**Lemma B.12.** *For all $s \in \mathcal{S}_g \times \mathcal{S}_{[n]}$, and for all $a_g \in \mathcal{A}_g$, we have:*

$$Q^*(s, a_g) - \frac{1}{\binom{n}{k}} \sum_{\Delta \in \binom{[n]}{k}} \hat{Q}_k^T(s_g, F_{s_\Delta}, a_g) \le \gamma^T \frac{\tilde{r}}{1 - \gamma}$$

*Proof.* We bound the differences between $\hat{Q}_k^T$ at each Bellman iteration of our approximation to $Q^*$.

$$Q^*(s, a_g) - \frac{1}{\binom{n}{k}} \sum_{\Delta \in \binom{[n]}{k}} \hat{Q}_k^T(s_g, F_{s_\Delta}, a_g)$$

$$= \mathcal{T}Q^*(s, a_g) - \frac{1}{\binom{n}{k}} \sum_{\Delta \in \binom{[n]}{k}} \hat{\mathcal{T}}_k \hat{Q}_k^{T-1}(s_g, F_{s_\Delta}, a_g)$$

$$= r_{[n]}(s_g, s_{[n]}, a_g) + \gamma \mathbb{E}_{\substack{s_g' \sim P_g(\cdot|s_g, a_g), \\ s_i' \sim P_l(\cdot|s_i, s_g), \forall i \in [n]}} \max_{a_g' \in \mathcal{A}_g} Q^*(s', a_g')$$

$$\qquad - \frac{1}{\binom{n}{k}} \sum_{\Delta \in \binom{[n]}{k}} [r_{[\Delta]}(s_g, s_\Delta, a_g) + \gamma \mathbb{E}_{\substack{s_g' \sim P_g(\cdot|s_g, a_g) \\ s_i' \sim P_l(\cdot|s_i, s_g), \forall i \in \Delta}} \max_{a_g' \in \mathcal{A}_g} Q_k^T(s_g', F_{s_\Delta'}, a_g')]$$

Next, observe that $r_{[n]}(s_g, s_{[n]}, a_g) = \frac{1}{\binom{n}{k}} \sum_{\Delta \in \binom{[n]}{k}} r_{[\Delta]}(s_g, s_\Delta, a_g)$. To prove this, we write:

$$\frac{1}{\binom{n}{k}} \sum_{\Delta \in \binom{[n]}{k}} r_{[\Delta]}(s_g, s_\Delta, a_g) = \frac{1}{\binom{n}{k}} \sum_{\Delta \in \binom{[n]}{k}} \left( r_g(s_g, a_g) + \frac{1}{k} \sum_{i \in \Delta} r_l(s_i, s_g) \right)$$

$$= r_g(s_g, a_g) + \frac{\binom{n-1}{k-1}}{k\binom{n}{k}} \sum_{i \in [n]} r_l(s_i, s_g)$$

$$= r_g(s_g, a_g) + \frac{1}{n} \sum_{i \in [n]} r_l(s_i, s_g) := r_{[n]}(s_g, s_{[n]}, a_g)$$

In the second equality, we reparameterized the sum to count the number of times each $r_l(s_i, s_g)$ was added for each $i \in \Delta$, and in the last equality, we expanded and simplified the binomial coefficients.

Therefore:

$$\sup_{(s,a_g) \in \mathcal{S} \times \mathcal{A}_g} [Q^*(s, a_g) - \frac{1}{\binom{n}{k}} \sum_{\Delta \in \binom{[n]}{k}} \hat{Q}_k^T(s_g, F_{s_{[n]}}, a_g)]$$

$$= \sup_{(s,a_g) \in \mathcal{S} \times \mathcal{A}_g} [\mathcal{T}Q^*(s, a_g) - \frac{1}{\binom{n}{k}} \sum_{\Delta \in \binom{[n]}{k}} \hat{\mathcal{T}}_k \hat{Q}_k^{T-1}(s_g, F_{s_{[n]}}, a_g)]$$

$$= \gamma \sup_{(s,a_g) \in \mathcal{S} \times \mathcal{A}_g} [\mathbb{E}_{\substack{s_g' \sim P(\cdot|s_g,a_g) \\ s_i' \sim P_l(\cdot|s_i,s_g) \\ \forall i \in [n]}} \max_{a_g' \in \mathcal{A}_g} Q^*(s', a_g') - \frac{1}{\binom{n}{k}} \sum_{\Delta \in \binom{[n]}{k}} \mathbb{E}_{\substack{s_g' \sim P_g(\cdot|s_g,a_g) \\ s_i' \sim P_l(\cdot|s_i,s_g) \\ \forall i \in \Delta}} \max_{a_g' \in \mathcal{A}_g} \hat{Q}_k^{T-1}(s_g', F_{s_\Delta'}, a_g')]$$

$$= \gamma \sup_{(s,a_g) \in \mathcal{S} \times \mathcal{A}_g} \mathbb{E}_{\substack{s_g' \sim P_g(\cdot|s_g,a_g), \\ s_i' \sim P_l(\cdot|s_i,s_g), \forall i \in [n]}} [\max_{a_g' \in \mathcal{A}_g} Q^*(s', a_g') - \frac{1}{\binom{n}{k}} \sum_{\Delta \in \binom{[n]}{k}} \max_{a_g' \in \mathcal{A}_g} \hat{Q}_k^{T-1}(s_g', F_{s_\Delta'}, a_g')]$$

$$\leq \gamma \sup_{(s,a_g) \in \mathcal{S} \times \mathcal{A}_g} \mathbb{E}_{\substack{s_g' \sim P_g(\cdot|s_g,a_g), \\ s_i' \sim P_l(\cdot|s_i,s_g), \forall i \in [n]}} \max_{a_g' \in \mathcal{A}_g} [Q^*(s', a_g') - \frac{1}{\binom{n}{k}} \sum_{\Delta \in \binom{[n]}{k}} \hat{Q}_k^{T-1}(s_g', F_{s_\Delta'}, a_g')]$$

$$\leq \gamma \sup_{(s',a_g') \in \mathcal{S} \times \mathcal{A}_g} [Q^*(s', a_g') - \frac{1}{\binom{n}{k}} \sum_{\Delta \in \binom{[n]}{k}} \hat{Q}_k^{T-1}(s_g', F_{s_\Delta'}, a_g')]$$

We justify the first inequality by noting the general property that for positive vectors $v, v'$ for which $v \succeq v'$ which follows from the triangle inequality:

$$\|v - \frac{1}{\binom{n}{k}} \sum_{\Delta \in \binom{[n]}{k}} v'\|_\infty \geq |\|v\|_\infty - \|\frac{1}{\binom{n}{k}} \sum_{\Delta \in \binom{[n]}{k}} v'\|_\infty|$$

$$= \|v\|_\infty - \|\frac{1}{\binom{n}{k}} \sum_{\Delta \in \binom{[n]}{k}} v'\|_\infty$$

$$\geq \|v\|_\infty - \frac{1}{\binom{n}{k}} \sum_{\Delta \in \binom{[n]}{k}} \|v'\|_\infty$$

Therefore:

$$Q^*(s, a_g) - \frac{1}{\binom{n}{k}} \sum_{\Delta \in \binom{[n]}{k}} \hat{Q}_k^T(s_g, F_{s_\Delta}, a_g)$$

$$\leq \gamma^T \sup_{(s',a_g) \in \mathcal{S} \times \mathcal{A}_g} [Q^*(s', a_g') - \frac{1}{\binom{n}{k}} \sum_{\Delta \in \binom{[n]}{k}} \hat{Q}_k^0(s_g', F_{s_\Delta'}, a_g')]$$

$$= \frac{\gamma^T \tilde{r}}{1 - \gamma}$$

The first inequality follows from the $\gamma$-contraction property of the update procedure, and the ensuing equality follows from our bound on the maximum possible value of $Q$ from Theorem A.7 and noting that $\hat{Q}_k^0 := 0$.

Therefore, as $T \to \infty$,

$$Q^*(s, a_g) - \frac{1}{\binom{n}{k}} \sum_{\Delta \in \binom{[n]}{k}} \hat{Q}^T(s_g, F_{s_\Delta}, a_g) \to 0,$$

which proves the lemma. $\qquad\square$

## C   BOUNDING TOTAL VARIATION DISTANCE

As $|\Delta| \to n$, the total variation (TV) distance between the empirical distribution of $s_{[n]}$ and $s_\Delta$ goes to 0. We formalize this notion and prove this statement by obtaining tight bounds on the difference and showing that this error decays quickly.

**Remark C.1.** First, observe that if $\Delta$ is an independent random variable uniformly supported on $\binom{[n]}{k}$, then $s_\Delta$ is also an independent random variable uniformly supported on the global state $\binom{s_{[n]}}{k}$. To see this, let $\psi_1 : [n] \to \mathcal{S}_l$ where $\psi(i) = s_i$. This naturally extends to $\psi_k : [n]^k \to \mathcal{S}_l^k$ given by $\psi_k(i_1, \ldots, i_k) = (s_{i_1}, \ldots, s_{i_k})$, for all $k \in [n]$. Then, the independence of $\Delta$ implies the independence of the generated $\sigma$-algebra. Further, $\psi_k$ (which is a Lebesgue measurable function of a $\sigma$-algebra) is a sub-algebra, implying that $s_\Delta$ must also be an independent random variable.

For reference, we present the multidimensional Dvoretzky-Kiefer-Wolfowitz (DKW) inequality Dvoretzky et al. (1956); Massart (1990); Naaman (2021) which bounds the difference between an empirical distribution function for $s_\Delta$ and $s_{[n]}$ when each element of $\Delta$ for $|\Delta| = k$ is sampled uniformly randomly from $[n]$ *with* replacement.

**Theorem C.2** (Dvoretzky-Kiefer-Wolfowitz (DFW) inequality Dvoretzky et al. (1956)). *By the multi-dimensional version of the DKW inequality Naaman (2021), assume that $\mathcal{S}_l \subset \mathbb{R}^d$. Then, for any $\epsilon > 0$, the following statement holds for when $\Delta \subseteq [n]$ is sampled uniformly* with *replacement.*

$$\Pr\left[\sup_{x \in \mathcal{S}_l} \left| \frac{1}{|\Delta|} \sum_{i \in \Delta} \mathbb{1}\{s_i = x\} - \frac{1}{n} \sum_{i=1}^n \mathbb{1}\{s_i = x\} \right| < \epsilon \right] \geq 1 - d(n+1)e^{-2|\Delta|\epsilon^2}.$$

We give an analogous bound for the case when $\Delta$ is sampled uniformly from $[n]$ without replacement. However, our bound does *not* have a dependency on $d$, the dimension of $\mathcal{S}_l$ which allows us to consider non-numerical state-spaces.

Before giving the proof, we add a remark on this problem. Intuitively, when samples are chosen without replacement from a finite population, the marginal distribution, when conditioned on the random variable chosen, takes the running empirical distribution closer to the true distribution with high probability. However, we need a uniform probabilistic bound on the error that adapts to *worst-case marginal distributions* and decays with $k$.

Recall the landmark results of Hoeffding and Serfling in Hoeffding (1963) and Serfling (1974) which we restate below.

**Lemma C.3** (Lemma 4, Hoeffding). *Given a finite population, note that for any convex and continuous function $f : \mathbb{R} \to \mathbb{R}$, if $X = \{x_1, \ldots, x_k\}$ denotes a sample with replacement and $Y = \{y_1, \ldots, y_k\}$ denotes a sample without replacement, then:*

$$\mathbb{E}f\left(\sum_{i \in X} i\right) \leq \mathbb{E}f\left(\sum_{i \in Y} i\right)$$

**Lemma C.4** (Corollary 1.1, Serfling). *Suppose the finite subset $\mathcal{X} \subset \mathbb{R}$ such that $|\mathcal{X}| = n$ is bounded between $[a, b]$. Then, let $X = (x_1, \ldots, x_k)$ be a random sample of $\mathcal{X}$ of size $k$ chosen uniformly and without replacement. Denote $\mu := \frac{1}{n} \sum_{i=1}^n x_i$. Then:*

$$\Pr\left[\left| \frac{1}{k} \sum_{i=1}^k x_i - \mu \right| > \epsilon \right] < 2e^{-\frac{2k\epsilon^2}{(b-a)^2(1-\frac{k-1}{n})}}$$

We now present a sampling *without* replacement analog of the DKW inequality.

**Theorem C.5** (Sampling without replacement analogue of the DKW inequality). *Consider a finite population $\mathcal{X} = (x_1, \ldots, x_n) \in \mathcal{S}_l^n$. Let $\Delta \subseteq [n]$ be a random sample of size $k$ chosen uniformly and without replacement.*

*Then, for all $x \in \mathcal{S}_l$:*

$$\Pr\left[\sup_{x \in \mathcal{S}_l}\left|\frac{1}{|\Delta|}\sum_{i \in \Delta}\mathbb{1}\{x_i = x\} - \frac{1}{n}\sum_{i \in [n]}\mathbb{1}\{x_i = x\}\right| < \epsilon\right] \geq 1 - 2|\mathcal{S}_l|e^{-\frac{2|\Delta|n\epsilon^2}{n-|\Delta|+1}}$$

*Proof.* For each $x \in \mathcal{S}_l$, define the "$x$-surrogate population" of indicator variables as

$$\bar{\mathcal{X}}_x = (\mathbb{1}_{\{x_1=x\}}, \ldots, \mathbb{1}_{\{x_n=x\}}) \in \{0,1\}^n \tag{28}$$

Since the maximal difference between each element in this surrogate population is 1, we set $b - a = 1$ in Theorem C.4 when applied to $\bar{\mathcal{X}}_x$ to get:

$$\Pr\left[\left|\frac{1}{|\Delta|}\sum_{i \in \Delta}\mathbb{1}\{x_i = x\} - \frac{1}{n}\sum_{i \in [n]}\mathbb{1}\{x_i = x\}\right| < \epsilon\right] \geq 1 - 2e^{-\frac{2|\Delta|n\epsilon^2}{n-|\Delta|+1}}$$

In the above equation, the probability is over $\Delta \subseteq \binom{[n]}{k}$ and it holds for each $x \in \mathcal{S}_l$. Therefore, the randomness is only over $\Delta$.

Then, by a union bounding argument, we have:

$$\Pr\left[\sup_{x \in \mathcal{S}_l}\left|\frac{1}{|\Delta|}\sum_{i \in \Delta}\mathbb{1}\{x_i = x\} - \frac{1}{n}\sum_{i \in [n]}\mathbb{1}\{x_i = x\}\right| < \epsilon\right]$$

$$= \Pr\left[\bigcap_{x \in \mathcal{S}_l}\left\{\left|\frac{1}{|\Delta|}\sum_{i \in \Delta}\mathbb{1}\{x_i = x\} - \frac{1}{n}\sum_{i \in [n]}\mathbb{1}\{x_i = x\}\right| < \epsilon\right\}\right]$$

$$= 1 - \sum_{x \in \mathcal{S}_l}\Pr\left[\left|\frac{1}{|\Delta|}\sum_{i \in \Delta}\mathbb{1}\{x_i = x\} - \frac{1}{n}\sum_{i \in [n]}\mathbb{1}\{x_i = x\}\right| \geq \epsilon\right]$$

$$\geq 1 - 2|\mathcal{S}_l|e^{-\frac{2|\Delta|n\epsilon^2}{n-|\Delta|+1}}$$

This proves the claim. $\qquad\square$

Then, combining the Lipschitz continuity bound from Theorem 4.1 and the total variation distance bound from Theorem 4.2 yields Theorem C.6.

**Theorem C.6.** *For all $s_g \in \mathcal{S}_g, s_1, \ldots, s_n \in \mathcal{S}_l^n, a_g \in \mathcal{A}_g$, we have that with probability atleast $1 - \delta$:*

$$|\hat{Q}_k^T(s_g, F_{s_\Delta}, a_g) - \hat{Q}_n^T(s_g, F_{s_{[n]}}, a_g)| \leq \frac{2\|r_l(\cdot, \cdot)\|_\infty}{1 - \gamma}\sqrt{\frac{n - |\Delta| + 1}{8n|\Delta|}\ln(2|\mathcal{S}_l|/\delta)}$$

*Proof.* By the definition of total variation distance, observe that

$$\mathrm{TV}(F_{s_\Delta}, F_{s_{[n]}}) \leq \epsilon \iff \sup_{x \in \mathcal{S}_l}|F_{s_\Delta} - F_{s_{[n]}}| < 2\epsilon \tag{29}$$

Then, let $\mathcal{X} = \mathcal{S}_l$ be the finite population in Theorem C.5 and recall the Lipschitz-continuity of $\hat{Q}_k^T$ from Theorem B.2:

$$\left|\hat{Q}_k^T(s_g, F_{s_\Delta}, a_g) - \hat{Q}_n^T(s_g, F_{s_{[n]}}, a_g)\right| \leq \left(\sum_{t=0}^{T-1}2\gamma^t\right)\|r_l(\cdot, \cdot)\|_\infty \cdot \mathrm{TV}(F_{s_\Delta}, F_{s_{[n]}})$$

$$\leq \frac{2}{1 - \gamma}\|r_l(\cdot, \cdot)\|_\infty \cdot \epsilon$$

By setting the error parameter in Theorem C.5 to $2\epsilon$, we find that Equation (29) occurs with probability at least $1 - 2|\mathcal{S}_l|e^{-2|\Delta|n\epsilon^2/(n-|\Delta|+1)}$.

$$\Pr\left[\left|\hat{Q}_k^T(s_g, F_{s_\Delta}, a_g) - \hat{Q}_n^T(s_g, F_{s_{[n]}}, a_g)\right| \leq \frac{2\epsilon}{1-\gamma}\|r_l(\cdot, \cdot)\|_\infty\right] \geq 1 - 2|\mathcal{S}_l|e^{-\frac{8n|\Delta|\epsilon^2}{n-|\Delta|+1}}$$

Finally, we parameterize the probability to $1 - \delta$ to solve for $\epsilon$, which yields

$$\epsilon = \sqrt{\frac{n - |\Delta| + 1}{8n|\Delta|}\ln(2|\mathcal{S}_l|/\delta)}.$$

This proves the theorem. □

The following lemma is not used in the main result; however, we include it to demonstrate why popular TV-distance bounding methods using the Kullback-Liebler (KL) divergence and the Bretagnolle-Huber inequality (Tsybakov, 2008) only yield results with a suboptimal subtractive decay of $\sqrt{|\Delta|/n}$. In comparison, Theorem 4.2 achieves a stronger multiplicative decay of $1/\sqrt{|\Delta|}$.

**Lemma C.7.**
$$\mathrm{TV}(F_{s_\Delta}, F_{s_{[n]}}) \leq \sqrt{1 - |\Delta|/n}$$

*Proof.* By the symmetry of the total variation distance, we have $\mathrm{TV}(F_{s_{[n]}}, F_{s_\Delta}) = \mathrm{TV}(F_{s_\Delta}, F_{s_{[n]}})$.

From the Bretagnolle-Huber inequality Tsybakov (2008) we have that $\mathrm{TV}(f, g) = \sqrt{1 - e^{-D_{\mathrm{KL}}(f\|g)}}$. Here, $D_{\mathrm{KL}}(f\|g)$ is the Kullback-Leibler (KL) divergence metric between probability distributions $f$ and $g$ over the sample space, which we denote by $\mathcal{X}$ and is given by

$$D_{\mathrm{KL}}(f\|g) := \sum_{x \in \mathcal{X}} f(x)\ln\frac{f(x)}{g(x)} \tag{30}$$

Thus, from Equation (30):

$$D_{\mathrm{KL}}(F_{s_\Delta}\|F_{s_{[n]}}) = \sum_{x \in \mathcal{S}_l}\left(\frac{1}{|\Delta|}\sum_{i \in \Delta}\mathbb{1}\{s_i = x\}\right)\ln\frac{n\sum_{i \in \Delta}\mathbb{1}\{s_i = x\}}{|\Delta|\sum_{i \in [n]}\mathbb{1}\{s_i = x\}}$$

$$= \frac{1}{|\Delta|}\sum_{x \in \mathcal{S}_l}\left(\sum_{i \in \Delta}\mathbb{1}\{s_i = x\}\right)\ln\frac{n}{|\Delta|}$$

$$+ \frac{1}{|\Delta|}\sum_{x \in \mathcal{S}_l}\left(\sum_{i \in \Delta}\mathbb{1}\{s_i = x\}\right)\ln\frac{\sum_{i \in \Delta}\mathbb{1}\{s_i = x\}}{\sum_{i \in [n]}\mathbb{1}\{s_i = x\}}$$

$$= \ln\frac{n}{|\Delta|} + \frac{1}{|\Delta|}\sum_{x \in \mathcal{S}_l}\left(\sum_{i \in \Delta}\mathbb{1}\{s_i = x\}\right)\ln\frac{\sum_{i \in \Delta}\mathbb{1}\{s_i = x\}}{\sum_{i \in [n]}\mathbb{1}\{s_i = x\}}$$

$$\leq \ln(n/|\Delta|)$$

In the third line, we note that $\sum_{x \in \mathcal{S}_l}\sum_{i \in \Delta}\mathbb{1}\{s_i = x\} = |\Delta|$ since each local agent contained in $\Delta$ must have some state contained in $\mathcal{S}_l$. In the last line, we note that $\sum_{i \in \Delta}\mathbb{1}\{s_i = x\} \leq \sum_{i \in [n]}\mathbb{1}\{s_i = x\}$, for each $x \in \mathcal{S}_l$, and hence the summation of logarithmic terms in the third line is negative.

Finally, using this bound in the Bretagnolle-Huber inequality yields the lemma. □

# D USING THE PERFORMANCE DIFFERENCE LEMMA TO BOUND THE OPTIMALITY GAP

Recall from Theorem A.13 that the fixed-point of the empirical adapted Bellman operator $\hat{\mathcal{T}}_{k,m}$ is $\hat{Q}_{k,m}^{\text{est}}$. Further, recall from Theorem 3.3 that $\|\hat{Q}_k^* - \hat{Q}_{k,m}^{\text{est}}\|_\infty \le \epsilon_{k,m}$.

**Lemma D.1.** Fix $s \in \mathcal{S} := \mathcal{S}_g \times \mathcal{S}_l^n$. Suppose we are given a $T$-length sequence of i.i.d. random variables $\Delta_1, \ldots, \Delta_T$, distributed uniformly over the support $\binom{[n]}{k}$. Further, suppose we are given a fixed sequence $\delta_1, \ldots, \delta_T \in (0, 1)$. Then, for each action $a_g \in \mathcal{A}_g$ and for $i \in [T]$, define events $B_i^{a_g}$ such that:

$$B_i^{a_g} := \left\{ \left| Q^*(s_g, s_{[n]}, a_g) - \hat{Q}_{k,m}^{\text{est}}(s_g, F_{s_{\Delta_i}}, a_g) \right| > \sqrt{\frac{n-k+1}{8kn} \ln \frac{2|\mathcal{S}_l|}{\delta_i}} \cdot \frac{2}{1-\gamma} \|r_l(\cdot, \cdot)\|_\infty + \epsilon_{k,m} \right\}$$

Next, for $i \in [M]$, we define "bad-events" $B_i$ such that $B_i = \bigcup_{a_g \in \mathcal{A}_g} B_i^{a_g}$. Next, denote $B = \cup_{i=1}^T B_i$. Then, the probability that no "bad event" occurs is:

$$\Pr\left[\bar{B}\right] \ge 1 - |\mathcal{A}_g| \sum_{i=1}^T \delta_i$$

*Proof.*

$$\left| Q^*(s_g, s_{[n]}, a_g) - \hat{Q}_{k,m}^{\text{est}}(s_g, F_{s_\Delta}, a_g) \right| \le \left| Q^*(s_g, s_{[n]}, a_g) - \hat{Q}_k^*(s_g, F_{s_\Delta}, a_g) \right|$$
$$+ \left| \hat{Q}_k^*(s_g, F_{s_\Delta}, a_g) - \hat{Q}_{k,m}^{\text{est}}(s_g, F_{s_\Delta}, a_g) \right|$$
$$\le \left| Q^*(s_g, s_{[n]}, a_g) - \hat{Q}_k^*(s_g, F_{s_\Delta}, a_g) \right| + \epsilon_{k,m}$$

The first inequality above follows from the triangle inequality, and the second inequality uses $|Q^*(s_g, s_{[n]}, a_g) - \hat{Q}_k^*(s_g, F_{s_\Delta}, a_g)| \le \|Q^*(s_g, s_{[n]}, a_g) - \hat{Q}_k^*(s_g, F_{s_\Delta}, a_g)\|_\infty \le \epsilon_{k,m}$, where $\epsilon_{k,m}$ is defined in Theorem 3.3. Then, from Theorem C.6, we have that with probability at least $1 - \delta_i$,

$$\left| Q^*(s_g, s_{[n]}, a_g) - \hat{Q}_k^*(s_g, F_{s_\Delta}, a_g) \right| \le \sqrt{\frac{n-k+1}{8nk} \ln \frac{2|\mathcal{S}_l|}{\delta_i}} \cdot \frac{2}{1-\gamma} \|r_l(\cdot, \cdot)\|_\infty$$

So, event $B_i$ occurs with probability almost $\delta_i$. Thus, by repeated applications of the union bound, we get:

$$\Pr[\bar{B}] \ge 1 - \sum_{i=1}^T \sum_{a_g \in \mathcal{A}_g} \Pr[B_i^{a_g}]$$
$$\ge 1 - |\mathcal{A}_g| \sum_{i=1}^T \Pr[B_i^{a_g}]$$

Finally, substituting $\Pr[\bar{B}_i^{a_g}] \le \delta_i$ yields the lemma. $\qquad\square$

Recall that for any $s \in \mathcal{S} := \mathcal{S}_g \times \mathcal{S}_l^n \cong \mathcal{S}_g$, the policy function $\pi_{k,m}^{\text{est}}(s)$ is defined as a uniformly random element in the maximal set of $\hat{\pi}_{k,m}^{\text{est}}$ evaluated on all possible choices of $\Delta$. Formally:

$$\pi_{k,m}^{\text{est}}(s) \sim \mathcal{U}\left\{ \hat{\pi}_{k,m}^{\text{est}}(s_g, F_{s_\Delta}) : \Delta \in \binom{[n]}{k} \right\} \tag{31}$$

We now use the celebrated performance difference lemma from Kakade & Langford (2002), restated below for convenience in Theorem D.2, to bound the value functions generated between $\pi_{k,m}^{\text{est}}$ and $\pi^*$.

**Theorem D.2** (Performance Difference Lemma). Given policies $\pi_1, \pi_2$, with corresponding value functions $V^{\pi_1}, V^{\pi_2}$:

$$V^{\pi_1}(s) - V^{\pi_2}(s) = \frac{1}{1-\gamma} \mathbb{E}_{\substack{s' \sim d_s^{\pi_1} \\ a_g' \sim \pi_1(\cdot | s')}} [A^{\pi_2}(s', a_g')]$$

Here, $A^{\pi_2}(s', a_g') := Q^{\pi_2}(s', a_g') - V^{\pi_2}(s')$ and $d_s^{\pi_1}(s') = (1-\gamma) \sum_{h=0}^{\infty} \gamma^h \Pr_h^{\pi_1}[s', s]$ where $\Pr_h^{\pi_1}[s', s]$ is the probability of $\pi_1$ reaching state $s'$ at time step $h$ starting from state $s$.

**Theorem D.3** (Bounding value difference). *For any $s \in \mathcal{S} := \mathcal{S}_g \times \mathcal{S}_l^n$ and $(\delta_1, \delta_2) \in (0, 1]^2$, we have:*

$$V^{\pi^*}(s) - V^{\pi_{k,m}^{\text{est}}}(s) \leq \frac{2\|r_l(\cdot, \cdot)\|_\infty}{(1-\gamma)^2} \sqrt{\frac{n-k+1}{2nk}} \sqrt{\ln \frac{2|\mathcal{S}_l|}{\delta_1}} + \frac{2\tilde{r}}{(1-\gamma)^2} |\mathcal{A}_g| \delta_1 + \frac{2\epsilon_{k,m}}{1-\gamma}$$

*Proof.* Note that by definition of the advantage function,

$$\mathbb{E}_{a_g' \sim \pi_{k,m}^{\text{est}}(\cdot | s')} A^{\pi^*}(s', a_g') = \mathbb{E}_{a_g' \sim \pi_{k,m}^{\text{est}}(\cdot | s')} [Q^{\pi^*}(s', a_g') - V^{\pi^*}(s')]$$

$$= \mathbb{E}_{a_g' \sim \pi_{k,m}^{\text{est}}(\cdot | s')} [Q^{\pi^*}(s', a_g') - \mathbb{E}_{a \sim \pi^*(\cdot | s')} Q^{\pi^*}(s', a_g)]$$

$$= \mathbb{E}_{a_g' \sim \pi_{k,m}^{\text{est}}(\cdot | s')} \mathbb{E}_{a_g \sim \pi^*(\cdot | s')} [Q^{\pi^*}(s', a_g') - Q^{\pi^*}(s', a_g)].$$

Since $\pi^*$ is a deterministic policy, we can write:

$$\mathbb{E}_{a_g' \sim \pi_{k,m}^{\text{est}}(\cdot | s')} \mathbb{E}_{a_g \sim \pi^*(\cdot | s')} A^{\pi^*}(s', a_g') = \mathbb{E}_{a_g' \sim \pi_{k,m}^{\text{est}}(\cdot | s')} [Q^{\pi^*}(s', a_g') - Q^{\pi^*}(s', \pi^*(s'))]$$

$$= \frac{1}{\binom{n}{k}} \sum_{\Delta \in \binom{[n]}{k}} [Q^{\pi^*}(s', \hat{\pi}_{k,m}^{\text{est}}(s_g', F_{s_\Delta'})) - Q^{\pi^*}(s', \pi^*(s'))]$$

Then, by the linearity of expectations and the performance difference lemma (while noting that $Q^{\pi^*}(\cdot, \cdot) = Q^*(\cdot, \cdot)$):

$$V^{\pi^*}(s) - V^{\pi_{k,m}^{\text{est}}}(s) = \frac{1}{1-\gamma} \sum_{\Delta \in \binom{[n]}{k}} \frac{1}{\binom{n}{k}} \mathbb{E}_{s' \sim d_s^{\pi_{k,m}^{\text{est}}}} \left[ Q^{\pi^*}(s', \pi^*(s')) - Q^{\pi^*}(s', \hat{\pi}_{k,m}^{\text{est}}(s_g', F_{s_\Delta'})) \right]$$

$$= \frac{1}{1-\gamma} \sum_{\Delta \in \binom{[n]}{k}} \frac{1}{\binom{n}{k}} \mathbb{E}_{s' \sim d_s^{\pi_{k,m}^{\text{est}}}} \left[ Q^*(s', \pi^*(s')) - Q^*(s', \hat{\pi}_{k,m}^{\text{est}}(s_g', F_{s_\Delta'})) \right]$$

Next, we use Theorem D.4 to bound this difference (where the probability distribution function of $\mathcal{D}$ is set as $d_s^{\pi_{k,m}^{\text{est}}}$ as defined in Theorem D.2) while letting $\delta_1 = \delta_2$:

$$V^{\pi^*}(s) - V^{\pi_{k,m}^{\text{est}}}(s)$$

$$\leq \frac{1}{1-\gamma} \sum_{\Delta \in \binom{[n]}{k}} \frac{1}{\binom{n}{k}} \left[ \frac{2\|r_l(\cdot, \cdot)\|_\infty}{1-\gamma} \sqrt{\frac{n-k+1}{2nk}} \left( \sqrt{\ln \frac{2|\mathcal{S}_l|}{\delta_1}} \right) + \frac{2\tilde{r}}{1-\gamma} |\mathcal{A}_g| \delta_1 + 2\epsilon_{k,m} \right]$$

$$\leq \frac{2\|r_l(\cdot, \cdot)\|_\infty}{(1-\gamma)^2} \sqrt{\frac{n-k+1}{2nk}} \left( \sqrt{\ln \frac{2|\mathcal{S}_l|}{\delta_1}} \right) + \frac{2\tilde{r}}{(1-\gamma)^2} |\mathcal{A}_g| \delta_1 + \frac{2\epsilon_{k,m}}{1-\gamma}$$

This proves the theorem. $\qquad \square$

**Lemma D.4.** *For any arbitrary distribution $\mathcal{D}$ of states $\mathcal{S} := \mathcal{S}_g \times \mathcal{S}_l^n$, for any $\Delta \in \binom{[n]}{k}$ and for $\delta_1, \delta_2 \in (0, 1]$, we have:*

$$\mathbb{E}_{s' \sim \mathcal{D}}[Q^*(s', \pi^*(s')) - Q^*(s', \hat{\pi}_{k,m}^{\text{est}}(s_g', F_{s_\Delta'}))]$$

$$\leq \frac{2\|r_l(\cdot, \cdot)\|_\infty}{1-\gamma} \sqrt{\frac{n-k+1}{8nk}} \left( \sqrt{\ln \frac{2|\mathcal{S}_l|}{\delta_1}} + \sqrt{\ln \frac{2|\mathcal{S}_l|}{\delta_2}} \right) + \frac{\tilde{r}}{1-\gamma} |\mathcal{A}_g| (\delta_1 + \delta_2) + 2\epsilon_{k,m}$$

*Proof.* Denote $\zeta_{k,m}^{s,\Delta} := Q^*(s, \pi^*(s)) - Q^*(s, \hat{\pi}_{k,m}^{\text{est}}(s_g, F_{s_\Delta}))$. We define the indicator function $\mathcal{I} : \mathcal{S} \times \mathbb{N} \times (0,1] \times (0,1]$ by:

$$\mathcal{I}(s, k, \delta_1, \delta_2) = \mathbb{1}\left\{\zeta_{k,m}^{s,\Delta} \leq \frac{2\|r_l(\cdot,\cdot)\|_\infty}{1-\gamma}\sqrt{\frac{n-k+1}{8nk}}\left(\sqrt{\ln\frac{2|\mathcal{S}_l|}{\delta_1}} + \sqrt{\ln\frac{2|\mathcal{S}_l|}{\delta_2}}\right) + 2\epsilon_{k,m}\right\}$$

We then study the expected difference between $Q^*(s', \pi^*(s'))$ and $Q^*(s', \hat{\pi}_{k,m}^{\text{est}}(s'_g, F_{s'_\Delta}))$. Observe that:

$$\mathbb{E}_{s'\sim\mathcal{D}}[\zeta_{k,m}^{s,\Delta}] = \mathbb{E}_{s'\sim\mathcal{D}}[Q^*(s', \pi^*(s')) - Q^*(s', \hat{\pi}_{k,m}^{\text{est}}(s'_g, F_{s'_\Delta}))]$$

$$= \mathbb{E}_{s'\sim\mathcal{D}}\left[\mathcal{I}(s', k, \delta_1, \delta_2)(Q^*(s', \pi^*(s')) - Q^*(s', \hat{\pi}_{k,m}^{\text{est}}(s'_g, F_{s'_\Delta})))\right]$$

$$+ \mathbb{E}_{s'\sim\mathcal{D}}[(1 - \mathcal{I}(s', k, \delta_1, \delta_2))(Q^*(s', \pi^*(s')) - Q^*(s', \hat{\pi}_{k,m}^{\text{est}}(s'_g, F_{s'_\Delta})))]$$

Here, we have used the general property for a random variable $X$ and constant $c$ that $\mathbb{E}[X] = \mathbb{E}[X\mathbb{1}\{X \leq c\}] + \mathbb{E}[(1 - \mathbb{1}\{X \leq c\})X]$. Then,

$$\mathbb{E}_{s'\sim\mathcal{D}}[Q^*(s', \pi^*(s')) - Q^*(s', \hat{\pi}_{k,m}^{\text{est}}(s'_g, F_{s'_\Delta}))]$$

$$\leq \frac{2\|r_l(\cdot,\cdot)\|_\infty}{1-\gamma}\sqrt{\frac{n-k+1}{8nk}}\left(\sqrt{\ln\frac{2|\mathcal{S}_l|}{\delta_1}} + \sqrt{\ln\frac{2|\mathcal{S}_l|}{\delta_2)}}\right) + 2\epsilon_{k,m}$$

$$+ \frac{\tilde{r}}{1-\gamma}\left(1 - \mathbb{E}_{s'\sim\mathcal{D}}\mathcal{I}(s', k, \delta_1, \delta_2))\right)$$

$$\leq \frac{2\|r_l(\cdot,\cdot)\|_\infty}{1-\gamma}\sqrt{\frac{n-k+1}{8nk}}\left(\sqrt{\ln\frac{2|\mathcal{S}_l|}{\delta_1}} + \sqrt{\ln\frac{2|\mathcal{S}_l|}{\delta_2)}}\right) + 2\epsilon_{k,m}$$

$$+ \frac{\tilde{r}}{1-\gamma}|\mathcal{A}_g|(\delta_1 + \delta_2)$$

For the first term in the first inequality, we use $\mathbb{E}[X\mathbb{1}\{X \leq c\}] \leq c$. For the second term, we trivially bound $Q^*(s', \pi^*(s')) - Q^*(s', \hat{\pi}_{k,m}^{\text{est}}(s'_g, F_{s'_\Delta}))$ by the maximum value $Q^*$ can take, which is $\frac{\tilde{r}}{1-\gamma}$ by Theorem A.7.

In the second inequality, we use the fact that the expectation of an indicator function is the conditional probability of the underlying event. The second inequality follows from Theorem D.5 which yields the claim. $\qquad\square$

**Lemma D.5.** *For a fixed $s' \in \mathcal{S} := \mathcal{S}_g \times \mathcal{S}_l^n$, for any $\Delta \in \binom{[n]}{k}$, and for $\delta_1, \delta_2 \in (0,1]$, we have that with probability at least $1 - |\mathcal{A}_g|(\delta_1 + \delta_2)$:*

$$Q^*(s', \pi^*(s')) - Q^*(s', \hat{\pi}_{k,m}^{\text{est}}(s'_g, F_{s'_\Delta})) \leq \frac{2\|r_l(\cdot,\cdot)\|_\infty}{1-\gamma}\sqrt{\frac{n-k+1}{8nk}}\left(\sqrt{\ln\frac{2|\mathcal{S}_l|}{\delta_1}} + \sqrt{\ln\frac{2|\mathcal{S}_l|}{\delta_2}}\right) + 2\epsilon_{k,m}$$

*Proof.*

$$Q^*(s', \pi^*(s')) - Q^*(s', \hat{\pi}_{k,m}^{\text{est}}(s'_g, F_{s'_\Delta}))$$

$$= Q^*(s', \pi^*(s')) - Q^*(s', \hat{\pi}_{k,m}^{\text{est}}(s'_g, F_{s'_\Delta})) + \hat{Q}_{k,m}^{\text{est}}(s'_g, s'_\Delta, \pi^*(s'))$$

$$- \hat{Q}_{k,m}^{\text{est}}(s'_g, s'_\Delta, \pi^*(s')) + \hat{Q}_{k,m}^{\text{est}}(s'_g, s'_\Delta, \hat{\pi}_{k,m}^{\text{est}}(s'_g, F_{s'_\Delta}))$$

$$- \hat{Q}_{k,m}^{\text{est}}(s'_g, F_{s'_\Delta}, \hat{\pi}_{k,m}^{\text{est}}(s'_g, F_{s'_\Delta}))$$

By the monotonicity of the absolute value and by the triangle inequality,

$$Q^*(s', \pi^*(s')) - Q^*(s', \hat{\pi}_{k,m}^{\text{est}}(s'_g, F_{s'_\Delta}))$$

$$\leq |Q^*(s', \pi^*(s')) - \hat{Q}_{k,m}^{\text{est}}(s'_g, F_{s'_\Delta}, \pi^*(s'))|$$

$$+ |\hat{Q}_{k,m}^{\text{est}}(s'_g, F_{s'_\Delta}, \hat{\pi}_{k,m}^{\text{est}}(s'_g, F_{s'_\Delta})) - Q^*(s', \hat{\pi}_{k,m}^{\text{est}}(s'_g, F_{s'_\Delta}))|$$

The above inequality crucially uses the fact that the residual term $\hat{Q}_{k,m}^{\text{est}}(s_g', F_{s_\Delta'}, \pi^*(s')) - \hat{Q}_{k,m}^{\text{est}}(s_g', F_{s_\Delta'}, \hat{\pi}_{k,m}^{\text{est}}(s_g', F_{s_\Delta'})) \leq 0$, since $\hat{\pi}_{k,m}^{\text{est}}$ is the optimal greedy policy for $\hat{Q}_{k,m}^{\text{est}}$.

Finally, applying the error bound derived in Theorem D.1 for two timesteps completes the proof. $\square$

**Corollary D.6.** Optimizing parameters in Theorem D.3 yields:

$$V^{\pi^*}(s) - V^{\pi_{k,m}^{\text{est}}}(s) \leq \frac{2\tilde{r}}{(1-\gamma)^2} \left( \sqrt{\frac{n-k+1}{2nk} \ln(2|\mathcal{S}_l||\mathcal{A}_g|\sqrt{k})} + \frac{1}{\sqrt{k}} \right) + \frac{2\epsilon_{k,m}}{1-\gamma}$$

*Proof.* Recall from Theorem D.3 that:

$$V^{\pi^*}(s) - V^{\pi_{k,m}^{\text{est}}}(s) \leq \frac{2\|r_l(\cdot,\cdot)\|_\infty}{(1-\gamma)^2} \sqrt{\frac{n-k+1}{2nk}} \left( \sqrt{\ln \frac{2|\mathcal{S}_l|}{\delta_1}} \right) + \frac{2\|r_l(\cdot,\cdot)\|_\infty}{(1-\gamma)^2}|\mathcal{A}_g|\delta_1 + \frac{2\epsilon_{k,m}}{1-\gamma}$$

Note $\|r_l(\cdot,\cdot)\|_\infty \leq \tilde{r}$ from Assumption 2.2. Then,

$$V^{\pi^*}(s) - V^{\pi_{k,m}^{\text{est}}}(s) \leq \frac{2\tilde{r}}{(1-\gamma)^2} \left( \sqrt{\frac{n-k+1}{2nk} \ln \frac{2|\mathcal{S}_l|}{\delta_1}} + |\mathcal{A}_g|\delta_1 \right) + \frac{2\epsilon_{k,m}}{1-\gamma}$$

Finally, setting $\delta_1 = \frac{1}{k^{1/2}|\mathcal{A}_g|}$ yields the claim. $\square$

**Corollary D.7.** Therefore, from Theorem D.6, we have:

$$V^{\pi^*}(s) - V^{\pi_{k,m}^{\text{est}}}(s) \leq O\left( \frac{\tilde{r}}{\sqrt{k}(1-\gamma)^2} \sqrt{\ln(2|\mathcal{S}_l||\mathcal{A}_g|\sqrt{k})} + \frac{\epsilon_{k,m}}{1-\gamma} \right)$$

$$= \widetilde{O}\left( \frac{\tilde{r}(1-\gamma)^{-2}}{\sqrt{k}} + \frac{\epsilon_{k,m}}{1-\gamma} \right)$$

This yields the bound from Theorem 3.4.

# E BEYOND THE TABULAR SETTING (IN THE LINEAR BELLMAN COMPLETE SETTING)

This section extends our result to non-tabular settings where the global agent's state space $\mathcal{S}_g$ can be a compact infinite set, and the global agent's action space $\mathcal{A}_g$ and each local agent's state space $\mathcal{S}_l$ is a finite set. In order to solve this problem, we make assumptions on the underlying MDP. A common assumption made is the linearity of value functions with respect to some known features (Sutton et al., 1999b; Chen & Theja Maguluri, 2022; Min et al., 2023).

At a high-level, this section learns the non-tabular function $\hat{Q}_{k,m}^{\text{est}}$ using function approximation methods from Golowich & Moitra (2024) under assumptions of Linear Bellman completeness, and using the triangle inequality to bound the performance between the optimal policy and the subsampled policy learned via sampling and linear function approximation.

Typically, existing works in the literature assume the existence of a map $\phi$ such that $\phi : \mathcal{S} \times \mathcal{A} \to \mathbb{R}^d$, where $d$ is the dimension of the embedding $\phi$. The weakest assumption made on the value function is that $Q$ is linear: for some $w \in \mathbb{R}^d$, $Q(s,a) = \langle w, \phi(s,a) \rangle$ for all $(s,a) \in \mathcal{S} \times \mathcal{A}$. However, it is conjectured that it is impossible to computationally learn a near-optimal policy under this assumption. Therefore, in accordance with Golowich & Moitra (2024), we make the stronger assumption that the underlying Markov decision process on the subsampled $Q$-function, $\hat{Q}_{k,m}^{\text{est}}$, satisfies Linear Bellman completeness. This class of Linear Bellman completeness captures a variety of function classes: for instance, it subsumes the set of linear MDPs and MDPs with low Bellman-Eluder (BE) dimension, which in turn contains rich subclasses such as functions with low Eluder dimension or low Bellman rank.

**Definition E.1** (Linear Bellman Completeness). Firstly, for $t \in \mathbb{N}$ and $k \leq n$, let $\mathcal{B}_{k,t}$ denote the set of coefficient vectors bounding linear functions on $\mathcal{S}_g \times \mathcal{S}_l^k \times \mathcal{A}_g$ such that

$$\mathcal{B}_{k,t} = \{\theta_k \in \mathbb{R}^d : |\langle \phi_{k,t}(s_g, s_\Delta, a_g), \theta_k \rangle| \leq 1, \forall (s_g, s_\Delta, a_t) \in \mathcal{S} \times \mathcal{S}_l^k \times \mathcal{A}_g)\}$$

Then, a Markov decision process is said to be *linear Bellman complete* with respect to the feature mapping $\{\phi_{k,t}\}_{t \in [T]}$ if for each $t \in [T]$ and $k \leq n$, there is a mapping $\mathcal{M}_{k,t} : \mathcal{B}_{k,t+1} \to \mathcal{B}_{k,t}$ such that for all $\theta_k \in \mathcal{B}_{k,t}$ and all $(s_g, s_\Delta, a_g) \in \mathcal{S}_g \times \mathcal{S}_l^k \times \mathcal{A}_g$,

$$\langle \phi_{k,t}(s_g, s_\Delta, a_g), \mathcal{M}_{k,t}\theta_k \rangle = \mathbb{E}_{s'_g, s'_\Delta \sim \mathbb{P}(\cdot|s_g, s_\Delta, a_g)} \left[ \max_{a'_g \in \mathcal{A}_g} \langle \phi_{k,t+1}(s'_g, s'_\Delta, a'_g), \theta_k \rangle \right], \quad (32)$$

and such that the reward $r_\Delta(s, a_g)$ is given by $r_\Delta(s, a_g) = \langle \phi_t(s_g, s_\Delta, a_g), \theta_{k,t} \rangle$, for $\theta_{k,t} \in \mathcal{B}_{k,t}$.

Therefore, we make the following assumptions:

**Assumption E.1.** For all $k \leq n$, the corresponding MDPs underlying the dynamics of $\hat{Q}_k^*$ is Linear Bellman complete.

**Assumption E.2.** $r_g$ and $r_l$ have a linear form, such that the structured reward function $r_\Delta(s, a_g) = r_g(s_g, a_g) + \frac{1}{k} \sum_{i \in \Delta} r_l(s_i, s_g)$ can be linearly decomposed to satisfy linear Bellman completeness.

Under the above assumptions of Linear Bellman completeness, the problem of learning the subsampled $\hat{Q}_{k,m}^{\text{est}}$ in the non-tabular setting is amenable to Algorithm 1 from Golowich & Moitra (2024), which provides the following theoretical guarantee:

**Lemma E.2** (Adapting theorem 5.10 from Golowich & Moitra (2024)). Suppose Algorithm 1 has $\tau$ samples and produces policy $\hat{\sigma}_{k,m}^{\text{est}}$ which is used to derive a subsampling policy $\sigma_{k,m}^{\text{est}}$. Then, if $\sigma_{k,m}^{\text{est}}$ is used $T'$ times, we have:

$$|V^{\pi_{k,m}^{\text{est}}}(s) - V^{\sigma_{k,m}^{\text{est}}}(s)| \leq 64 \frac{T'd|\mathcal{A}_g|}{\tau^{1/|\mathcal{A}_g|}}. \quad (33)$$

**Remark E.3.** We refer the interested reader to Algorithm 1 of Golowich & Moitra (2024). At a high-level, their algorithm designs exploration bonuses for which $\mathcal{B}_{k,t}$ is linear, and uses policy search through dynamic programming to design the bonus. This idea can be viewed as a variant of optimistic exploration. The result then follows by applying a variant of least-squares value iteration (LSVI) on these locally optimistic rewards.

**Corollary E.4.** Applying the triangle inequality, we see that:

$$V^{\pi^*}(s) - V^{\sigma_{k,m}^{\text{est}}}(s) = V^{\pi^*}(s) - V^{\pi_{k,m}^{\text{est}}}(s) + V^{\pi_{k,m}^{\text{est}}}(s) - V^{\sigma_{k,m}^{\text{est}}}(s)$$

$$\leq |V^{\pi^*}(s) - V^{\pi_{k,m}^{\text{est}}}(s)| + |V^{\pi_{k,m}^{\text{est}}}(s) - V^{\sigma_{k,m}^{\text{est}}}(s)|$$

$$\leq \frac{2\tilde{r}}{(1-\gamma)^2} \left( \sqrt{\frac{n-k+1}{2nk} \ln(2|\mathcal{S}_l||\mathcal{A}_g|\sqrt{k})} + \frac{1}{\sqrt{k}} \right) + \frac{2\epsilon_{k,m}}{1-\gamma} + \frac{64T'd|\mathcal{A}_g|}{\tau^{1/|\mathcal{A}_g|}}$$

Therefore, as the number of samples $\tau$ goes to infinity, we recover an optimality gap that decays with $k$ as $k \to n$.

# F ADDITIONAL DISCUSSIONS

**Discussion F.1** (Tighter Endpoint Analysis). Our theoretical result shows that $V^{\pi^*}(s) - V^{\pi_{k,m}^{\text{est}}}$ decays on the order of $O(1/\sqrt{k} + \epsilon_{k,m})$. For $k = n$, this bound is actually suboptimal since $\hat{Q}_k^*$ becomes $Q^*$. However, placing $|\Delta| = n$ in our weaker TV bound in Lemma C.7, we recovers a total variation distance of 0 when $k = n$, recovering the optimal endpoint bound.

**Discussion F.2** (Choice of $k$). Discussion 3.6 previously discussed the tradeoff in $k$ between the polynomial in $k$ complexity of learning the $\hat{Q}_k$ function and the decay in the optimality gap of $O(1/\sqrt{k})$. This discussion promoted $k = O(\log n)$ as a means to balance the tradeoff. However, the "correct" choice of $k$ truly depends on the amount of compute available, as well as the accuracy desired from the method. If the former is available, we recommend setting $k = \Omega(n)$ as it will yield a more optimal policy. Conversely, setting $k = O(\log n)$, when $n$ is large, would be the minimum $k$ recommended to realize any asymptotic decay of the optimality gap.

