# OpenReview forum: "Efficient Reinforcement Learning for Global Decision Making in the Presence of Local Agents at Scale"
_ICLR.cc/2025/Conference — Submitted to ICLR 2025_

### Official Review · Reviewer_KC7o · 2024-10-26

**Soundness:** 3
**Presentation:** 2
**Contribution:** 2
**Rating:** 5
**Confidence:** 2

**Summary:**

This paper considers a Multi-agent Reinforcement Learning problem where the global agent will make decisions for all local agents. This work proposes SUBSAMPLE-Q algorithm and prove its convergence. The authors also provide simulations to support their result.

**Strengths:**

This paper proposes SUBSAMPLE-Q algorithm to sample $k$ out of $n$ agents to update policy. The authors show that the time is polynomial in $k$. They also provide numerical result to support the finding.

**Weaknesses:**

My primary concern lies with the novelty of this work. The runtime of the approach in the paper is polynomial in $k$, but this comes at the expense of exponential dependence on $|S_l|$. In most of the case, $|S_l|$ is expected to be significantly larger than $k$ and $n$. Therefore, $k^{|S_l|}$ can perform worse than $|S_l|^k$, which suggests the proposed algorithm may not offer any advantages over standard Q-learning.

Additionally, the presentation of this work is poor and leaves several aspects unclear. For instance, the goal of Algorithm 2 is not clear to me. I would suggest the authors clarify the goal in the paragraph before those two algorithms.

Besides, there are lots of typos in this paper, including:

1. In line 110, I think it should be $s^0 \sim d_0$.

2. In line 204, I think the $\hat{Q}_k$ should be $\hat{Q}^t$.

3. The third line in algorithm 2 is unfinished.

4. The notation $\pi_{k,m}$ is introduced in line 220 but the definition of $m$ is not explained until line 249.

I strongly recommend that the authors carefully review and revise the paper to address these issues.

**Questions:**

Please see my comments in Weakness section.

---

> ### Author Response · Authors · 2024-11-18
> **Response to Reviewer KC7o**
>
> We thank the reviewer for their constructive comments. We have provided detailed responses to all the questions raised. In addition, we have corrected the typos, added a notation table in the Appendix, and written a reparameterized algorithm to handle the case when $|\mathcal{S}_l|\gg k$ which was brought up in the reviewer’s comments, and provided further exposition on the algorithms. We hope the reviewer can examine them and re-evaluate our paper. We are looking forward to addressing any further questions in the author-reviewer discussion period.
>
> > My primary concern lies with the novelty of this work. The runtime of the approach in the paper is polynomial in k, but this comes at the expense of exponential dependence on |Sl|. In most of the case, |Sl| is expected to be significantly larger than k and n. Therefore, k^|Sl| can perform worse than |Sl|^k, which suggests the proposed algorithm may not offer any advantages over standard Q-learning.
>
> We thank the reviewer for pointing this out and agree that if $|\mathcal{S}_l|\gg n$, then $n^{|\mathcal{S}_l|}$ (from mean-field iteration) would be worse than $|Sl|^n$ (what standard Q-learning offers), but it really depends on what the values of $n$ and $|\mathcal{S}_l|$ are. For a system of $n=200$ agents that only have binary 0/1 states, $2^{200}\gg 200^2$. However, in the scalable agent setting where $Sl$ is a fixed static set, but $n$ grows arbitrarily large, $n^{|\mathcal{S}_l|}$ grows slower than $|\mathcal{S}_l|^n$. In practical applications, $n^{|\mathcal{S}_l|}$ might still be too large; so, our work reduces this dependency to $k^{|\mathcal{S}_l|}$ for $k\leq n$ using our subsampling framework.
>
> However, by incorporating our subsampling approach into traditional value iteration (as opposed to only on mean-field value iteration), our algorithm can be reparameterized to also run in time $O(|\mathcal{S}_g||\mathcal{A}_g||\mathcal{S}_l|^k)$. This dependency on $k$ does not immediately follow from standard Q-learning, which only offers a $O((|\mathcal{S}_g||\mathcal{A}_g||\mathcal{S}_l|^n)$ runtime. For $k\ll n$, this is still an exponential improvement in runtime from $n$ to $k$. Since each of these learned $\hat{Q}_k$ functions encode the same value, the performance of the learned policies will be the same. Therefore, the runtime of our algorithm can easily be extended to be $\min\{O(|\mathcal{S}_l|^k|\mathcal{A}_g||\mathcal{S}_g|), O(k^{|\mathcal{S}_l|}|\mathcal{S}_l||\mathcal{S}_g||\mathcal{A}_g|)\}$, which (when $k\ll n$) is superior to the runtimes offered by _both_ traditional value iteration and mean-field value iteration of $\min\{O(|\mathcal{S}_l|^n|\mathcal{A}_g||\mathcal{S}_g|), O(n^{|\mathcal{S}_l|}|\mathcal{S}_l||\mathcal{S}_g||\mathcal{A}_g|)\}$.
> When we set $k\leq O(\log n)$, our runtimes become $\min\{O(n^{\log |\mathcal{S}_l|} |\mathcal{A}_g| |\mathcal{S}_g|), O((\log n)^{|\mathcal{S}_l|} |\mathcal{S}_l||\mathcal{S}_g| |\mathcal{A}_g|)\}$, which overcomes the curse of dimensionality with an error gap of $O(1/\sqrt{k})$ that decays to $0$ as $n\to\infty$. We have added this algorithm and some detailed discussions for this in the manuscript, and deeply thank the reviewer for raising this point.
>
> >Additionally, the presentation of this work is poor and leaves several aspects unclear. For instance, the goal of Algorithm 2 is not clear to me. I would suggest the authors clarify the goal in the paragraph before those two algorithms. For instance, the goal of Algorithm 2 is not clear to me.
>
> We thank the reviewer for their comment. To clarify the purpose of Algorithm 2 (now Algorithm 3 in the updated version), we believe that this subsampling approach is critical for converting our k-local-agent optimality guarantee to an n-local-agent approximate optimality. Specifically, in this multi-agent setting, learning $\hat{Q}_k$ alone is insufficient, as it only provides the optimal action for the global agent in a subsystem of $k$ agents. However, when we have $n$ agents (where $n$ can be much larger than $k$), we cannot ignore the remaining $n-k$ other agents, as one could design a reward function that penalizes them (intuitively, weakening the optimality of the global agent’s action). In this case, we have shown that it is better for the global agent to (uniformly at random) choose $k$ _different_ agents at each iteration, and use the policy derived in Algorithm 1 to make an action. This intuition is provided in the paragraphs preceding the SUBSAMPLE: execution algorithm. We have now elaborated and provided additional explanations in the revised manuscript.

---

> > ### Comment · Reviewer_KC7o · 2024-11-21
> > **Response to authors**
> >
> > Thank you for your response. Before engaging in further discussion, could you please specify where the changes have been made in your updated draft? It would be helpful if you could highlight the revisions using a different color or provide a summary indicating which lines or paragraphs correspond to the specific concerns raised in my initial review. This will make it easier for me to follow.

---

> ### Author Response · Authors · 2024-11-21
> **Reply to Reviewer KC7o**
>
> Dear Reviewer,
>
> Please find a revision in the uploaded manuscript, where the substantial changes to the draft have been highlighted in blue. Specifically, we draw your attention to:
>
> - Lines 205-225, where we made deep changes to the notations,
> - Lines 227-281, 288-291, and 340-341, where we incorporated the re-parameterized algorithm for the case where the state space is much larger than the number of local agents,
> - Lines 248-252, which elaborate on the purpose of Algorithm 2, and
> - Lines 371-377, where we discuss the novelty and theoretical significance of the result.
>
> Additionally, we have revised the notation and wording in many other areas throughout the document to improve the clarity of the exposition. Thank you.
>
> Best,
>
> Authors

---

> > ### Comment · Reviewer_KC7o · 2024-11-22
> > **Response to authors**
> >
> > Thank you for the effort you have put into addressing my concerns.
> >
> > However, my concern regarding the rate still persists. In many cases, $|S_l|$ is expected to approach infinity and $k$, even if very large, will remain finite. Consequently, $k^{|S_l|}$ would still be worse than $|S_l|^k$. The issue of $|S_l|$ approaching infinity was also raised by Reviewer kSFM. In your response to Reviewer kSFM, the authors added Appendix E to address this issue. However, Corollary E.4 still includes $|S_l|$ in the bound. This suggests that the proposed method may perform worse as $|S_l|$ grows unbounded.
> >
> > However, I acknowledge there is improvement in some cases, such as when $n=200$ and agents have binary 0/1 states, as mentioned by the authors. Based on this, I will increase my score to 5 at this moment.
> >
> > In addition, I hope the authors can carefully check for typos in the manuscript. For example, in line 262, I believe it should be $b \in 1, \ldots, k$ instead of $b \in 1, \ldots, n$.

---

> > > ### Author Response · Authors · 2024-11-30
> > > **Reply to Reviewer KC7o**
> > >
> > > We thank the reviewer again for their previous comments.
> > >
> > > After studying the dependence of $|\mathcal{S}\_l|$ on the optimality gap further, we have realized that we can indeed prove a stronger result in the non-tabular setting that replaces $|\mathcal{S}\_l|$ with $d$ in the bound of Corollary E.4 (page 35), without significant additions to the proofs. Here, $d$ is the dimension of the coefficient vector $\theta\_{k,t}$ used in Linear function approximation. Importantly, under the Linear Bellman completeness assumption, $d$ is much less than $\mathcal{S}\_l$ (and is expected to be a constant). Under this result, the right-hand side of Corollary E.4 decays to $0$ as $k\to n$, without any dependence on $|\mathcal{S}\_l|$.
> > >
> > > To elaborate on the proof method (as the period to upload new revisions has elapsed): we previously derived the first term in Corollary E.4 by simply plugging in the tabular-case bound on the optimality gap from Corollary D.7 (page 34). Indeed, as the reviewer previously pointed out, this term can already be large when $|\mathcal{S}\_l|$ is infinity, rendering this bound quite weak. This weakening is also expected, as that bound was derived for the tabular setting. However, the  $\mathcal{S}\_l$ dependence in the $\sqrt{\log |\mathcal{S}\_l|}$ term in the bound originates from Theorem C.5 (page 29). We can actually prove an alternate version of Theorem C.5 for the non-tabular case, under assumptions of Linear Bellman completeness, where the $|\mathcal{S}_l|$ is replaced by $d$. We can do this since the empirical distribution term is now observed through the coefficient vector $\theta\_{k,t}\in\mathbb{R}^d$; therefore, when we repeat the proof, we only accrue a constant multiplicative factor of dimension-$d$ in the term. Therefore, as $|\mathcal{S}_l|$ and $|\mathcal{S}_g|$ become infinite, Corollary E.4 still decays to $0$ when $k\to n$, without any dependence on the state spaces.
> > >
> > > Furthermore, even in the tabular setting: when $|\mathcal{S}_g|$ and $|\mathcal{S}_l|$ are small, but where the number of agents $n$ is extremely large, the popular mean-field techniques have a worse dependency on $|\mathcal{S}_l|$ in the runtime. For instance, in Reference 1, proposition 2.1 has an exponential dependence, and theorem 5.6 has a polynomial dependence. Similarly, Theorem 4.9 in Reference 2 shows that (in the tabular setting, in the worst case) there _must exist_ a dependence on $|\mathcal{S}_l|$ (via a reduction to the PPAD complexity class, which is generally believed to require exponential time to solve exactly). This issue also persists in the convergence guarantees in mean-field MARL (for instance, see the $\sqrt{X}=\sqrt{|\mathcal{S}_l|}$ dependence in Lemma 1 of Reference 3). While the previous best-works in the mean-field literature offer a runtime of $O(n^{\mathcal{S}_l})$, our subsampling work reduces this runtime to $O(k^{\mathcal{S}_l})$. When $k \leq O(\log n)$, this provides an exponential improvement in the run-time.
> > >
> > > Beyond the mean-field techniques, when $|\mathcal{S}_g|$ and $|\mathcal{S}_l|$ are large (but finite), relative to the number of agents $n$, the traditional $Q$-learning techniques have an exponential dependence on the number of agents, which can also make this setting intractable. However, _even_ in this setting, our algorithm provides an improvement in the runtime from $O(|\mathcal{S}_l|^n)$ to $O(|\mathcal{S}_l|^k)$. When $k \leq O(\log n)$, this again provides an exponential improvement in the run-time. Furthermore, our results _exactly_ recover these two previous-best algorithms when $k=n$: in this sense, our work can be viewed as a generalization of mean-field MARL and traditional $Q$-learning.
> > >
> > > We hope that this addresses some of the reviewer's earlier concerns. Again, we extend our thanks for the time devoted to reviewing our manuscript and for the insightful comments and feedback provided and are happy to answer any further questions the reviewer might have.
> > >
> > >
> > >
> > > References:
> > >
> > > [1] Mean-Field Controls with Q-Learning for Cooperative MARL: Convergence and Complexity Analysis (Gu et. al. SIAM 2021)
> > >
> > > [2] When is Mean-Field Reinforcement Learning Tractable and Relevant? (Yardim et. al. AAMAS 2024)
> > >
> > > [3] Mean-Field Approximation of Cooperative Constrained Multi-Agent Reinforcement Learning (CMARL) (Mondal et. al., JMLR 2024)

---

> > > > ### Comment · Reviewer_KC7o · 2024-12-01
> > > > **Respond to authors**
> > > >
> > > > Thank you for sharing your recent result on $|S_l|$ with me.
> > > >
> > > > I am glad to see that $|S_l|$ in the bound can be replaced by $d$, which appears to be an improvement in this work. Additionally, I agree with the authors that this work reduces the runtime from $O(n^{|S_l|})$ to $O(k^{|S_l|})$. However, since I am not very familiar with many related works, I am unable to tell the level of difficulty involved in achieving this improvement. Therefore, at this time, I am inclined not to further increase my score.

---

> ### Author Response · Authors · 2024-11-24
> **Reply to Reviewer KC7o**
>
> We sincerely appreciate the reviewer's insightful comments and suggestions. We have carefully checked and fixed the typos in the manuscript, and have uploaded a revised draft.
>
> We would like to clarify the following: in the tabular setting, when $|\mathcal{S}_l|$ is finite, the run-time of our (revised) algorithm scales as $\min(O(k^{|\mathcal{S}_l|}), O(|\mathcal{S}_l|^k))$ due to our sub-sampling algorithm. When we set $k = \log n$, this is exponentially faster than the run-time of $\min(O(n^{|\mathcal{S}_l|}), O(|\mathcal{S}_l|^n))$ offered by the previous-best methods (mean-field value iteration and traditional value-iteration, respectively). Additionally, our policy that sub-samples $k$ local agents is flexible for global agents who have a limited capacity to observe the local agents. Therefore, in this tabular setting, our proposed method outperforms existing methods in this decision-making setting.
>
> >In many cases, $|\mathcal{S}_l|$ is expected to approach infinity and $k$, even if very large, will remain finite.
>
> In our extension of the algorithm to the non-tabular setting in Appendix E, we look at the case where the global agent's state-space $|\mathcal{S}_g|$ is infinite but where the local agents' state space $|\mathcal{S}_l|$ remains finite. We showed in Corollary 3.4 that even if $\mathcal{S}_g$ is an infinite space, $|\mathcal{S}_g|$ does not directly impact the performance of the learned policy.
>
> As the reviewer pointed out, Corollary 3.4 contains a $\sqrt{\log |\mathcal{S}_l|}$ term, where if $|\mathcal{S}_l|$ goes to infinity, our bound on the performance of this algorithm gets worse. Here, when $|\mathcal{S}_l|$ is _also_ an infinite space, we believe that obtaining a bound on the optimality gap of the learned policy (where the bound does not depend on $|\mathcal{S}_l|$) is possible, and we leave it as a challenging (but exciting) problem for future works. Specifically, we conjecture that one could weaken the $|\mathcal{S}_l|$ dependence in Corollary 3.4 to a dependence on the dimension $d$ of the mapping $\phi:\mathcal{S}\times\mathcal{A}_g\to \mathbb{R}^d$. The existing results in the multi-agent RL with function approximation literature hold for competitive decentralized learning algorithms [1], and sparse centralized learning algorithms [2].
>
> We agree that the problem of integrating function approximation into our subsampling methodology is an interesting direction of future research, but emphasize that the goal of the paper is to provide performance guarantees and a more efficient policy-learning algorithm (than mean-field RL and value iteration) in the more fundamental _tabular_ setting where the number of local agents $n$ is large. In this case, extending the result to the case where $|\mathcal{S}_g|$ is an infinite space is still a significant improvement.
>
> References:
>
> 1) The Power of Exploiter: Provable Multi-Agent RL in Large State Spaces (Jin et. al., ICML 2022)
>
> 2) Scalable spectral representations for multi-agent reinforcement learning in network MDPs (Ren et. al., 2024)
>
> Thank you again for your effort in providing us with valuable and helpful suggestions. We are happy to provide further clarifications if you have any questions.

---

### Official Review · Reviewer_7eBU · 2024-11-04

**Soundness:** 3
**Presentation:** 2
**Contribution:** 2
**Rating:** 5
**Confidence:** 2

**Summary:**

The authors consider the setting of reinforcement learning in the context of distributed control. That is, a bunch of local agents are governed by a single global agent. The global agent is the only one in power to enact a policy. The state transitions of the local agents are determined by their previous state and the previous state of the global agent. The state transition of the global agent depends on the current state of the global agent and its enacted action. The reward obtained at each iteration is a sum of a global agent reward which depends on the action enacted and the average of the local agent rewards which do not rely on the action. The objective to optimize is the infinite horizon discounted reward. The main bottleneck of these problems is the exponential growth of the state space with the number of local agents. The paper proposes a technique of initially sampling k local agents, to learn their corresponding empirical deterministic optimal policy, and later using random sampling of k agents at each iteration to learn a random optimal policy.

**Strengths:**

1. The problem considered has wide applications across multiple domains such as power grid control, EV logistics planning, queuing system control, etc.

2. The suboptimality scales as $O(\frac{1}{\sqrt{k}})+O(\frac{1}{\sqrt{m}})$, where $k$ is the number of local agents sampled at each iteration and $m$ is the number of samples obtained at each iteration to solve for the deterministic optimal policy as a function of $k$.

3. Although they model the agents to be homogeneous for the most part, some heterogeniety is introduced by attaching a type to each agent which is transition invariant.

**Weaknesses:**

1. The total number of samples required for $T$ iterations of the algorithm is $Tm$, where $m$ is required to be large for lower suboptimalities. Moreover, they assume access to a generative model which is quite often an unrealistic assumption.

2. The heterogeniety modelled in the local agents is incredibly mild and the functional aspect of the problem largely treats them as homogeneous.

3. For applications such as queuing control, etc, the average reward is a more meaningful metric, since discounted reward objective doesn't capture stability issues.

4. It is unclear as to what the role of $T$ is in the final bounds (ie Theorem 3.4).

**Questions:**

Please refer above.

---

> ### Author Response · Authors · 2024-11-18
> **Response to Reviewer 7eBU**
>
> We thank the reviewer for their comments. Please find our detailed responses to all the questions raised. We look forward to addressing any further questions during the author-reviewer discussion period.
>
> >The total number of samples required for T iterations of the algorithm is Tm, where m is required to be large for lower suboptimalities.
>
> Indeed, at each iteration, for each state/action pair, the number of samples required in $m$, which is the same sample complexity that one incurs in value iteration. Having $Tm$ samples is common for many RL algorithms, and is usually a standard accepted bound, as one needs to have large $m$ so that the average of the samples converges to the expectation in the Bellman operator. Finding the smallest values for $m$ is also an area of ongoing work in the field [1,2]. Furthermore, for our setting where we solve a combinatorial problem of the curse of multiagency that emerges from the size of the joint state space of the agents, the bottleneck is the $O(|\mathcal{S}_l|^n)$ samples, which we reduce to $O(k^{|\mathcal{S}_l|})$ for $k\ll n$, rather than the multiplicative factor of $m$. As we show in Theorem 4.3, we can reduce $m$ to decrease the runtime but would pay a price in the optimality of the learned policy.
>
> >Moreover, they assume access to a generative model which is quite often an unrealistic assumption.
>
> In the offline-RL setting, a generative oracle is a standard assumption in the modern literature [3,4]. As we mention in Remark 3.2, it might be possible to convert this to an online problem by replacing the generative oracle with online samples. This might require tools from the stochastic approximation and no-regret RL literature [5]; however, we believe this is a highly non-trivial task, and leave this conversion for future works.
>
> >The heterogeneity modeled in the local agents is incredibly mild and the functional aspect of the problem largely treats them as homogeneous.
>
> Indeed, the heterogeneity in the local agents arises from assigning each agent a ‘type’ via a state factorization. This, in turn, allows pseudoheterogeneous local-agent reward functions. We agree that this heterogeneity is mild, and that the functional aspect of the problem treats them as homogeneous. However, modeling true heterogeneity in these settings has been known to be challenging: historically, it has come at the tradeoff of weak interactions between the agents where each agent’s action can only depend on its own state (which leads to a loss in optimality). Further, our paper models _two_ kinds of _truly_ heterogeneous agents, as the global agent has a different state space from the local agents. We leave it to future works to integrate our subsampling algorithm and analytic framework toward more strongly heterogeneous agent settings.
>
> >For applications such as queuing control, etc, the average reward is a more meaningful metric, since discounted reward objective doesn't capture stability issues.
>
> We thank the reviewer for their comment. Unlike the single-agent RL setting, the average reward version ($\gamma=1$) of the problem is provably worst-case NP-hard (see A.2 of [7]), and obtaining provable guarantees in this setting will require strong structural assumptions. Typically, the discounted metric is sufficient for learning stationary MDP policies, whereas the average reward setting is more meaningful for non-stationary MDPs. Since our model has stochastic time-invariant transition functions, the discounted metric is more meaningful. However, we leave it as an exciting topic of future study to see if our subsampling algorithm could be applied to the more challenging average reward setting.
>
> >It is unclear as to what the role of T is in the final bounds (ie Theorem 3.4).
>
> Algorithm 1 learns $\hat{\pi}\_{k,m}^T$, which we denote as $\hat{\pi}\_{k,m}^{est}$ when $T$ goes to infinity. Then, Algorithm 2 converts this to a distribution policy $\pi_{k,m}^{est}$. The bound in Theorem 3.4 takes $T$ to be large, where the difference is $\frac{1}{1-\gamma} - \sum_{t=0}^T \gamma^t$, which goes to $0$ as $T$ becomes large (by the convergence of the geometric sum, as $\gamma \in (0,1)$). We have clarified this in the manuscript.
>
> References:
>
> [1] Finite-Time Analysis of Asynchronous Stochastic Approximation and Q-learning [Qu, Wierman, PMLR 2020]
>
> [2] Sample complexity of asynchronous Q-learning: Sharper analysis and variance reduction [Li et. al. NeurIPS 2021]
>
> [3] Near-Optimal Sample Complexity Bounds for Constrained MDPs (Vaswani, Yang, Czepesvári, NeurIPS 2022)
>
> [4] Breaking the Curse of Multiagency in Robust Multi-Agent Reinforcement Learning (Shi et. al., 2024)
>
> [5] Concentration of Contractive Stochastic Approximation: Additive and Multiplicative Noise [Chen et. al. 2023]
>
> [6] Graphon Mean-Field Control for Cooperative Multi-Agent Reinforcement Learning (Hu et. al. 2024)
>
> [7] Scalable Multi-Agent Reinforcement Learning for Networked Systems with Average Reward [Qu et. al., NeurIPS 2020]

---

> ### Author Response · Authors · 2024-11-24
> **Response to Reviewer 7eBU**
>
> We sincerely thank the reviewer for their constructive suggestions and comments. As the deadline approaches, we hope the reviewer will have an opportunity to review our response. If there are any further comments or additional concerns, we would be grateful to address them. Please do not hesitate to let us know if any further clarifications or discussions are needed to assist in the evaluation.

---

### Official Review · Reviewer_kSFM · 2024-11-08

**Soundness:** 3
**Presentation:** 2
**Contribution:** 2
**Rating:** 5
**Confidence:** 2

**Summary:**

The paper considers a setting where there is a global decision making agent and there are many local agents. The paper frame this problem as a MDP problem with a global decision making agent and $n$ local agents. The paper proposes a subsampling based Q learning algorithm named SUBSAMPLE-Q and provide theoretical guarantees for the performance gap between the learned policy and the optimal policy. The paper also provides some numerical simulation experiments.

**Strengths:**

The paper considers a rather interesting RL setting with a global decision maker and local agents. The setting itself is fairly novel to me.

The theoretical results also seem to be interesting especially making the bound dependent on $k$.

**Weaknesses:**

In assumption 2.1, it assumes state space for local and global agent and the action space — all are finite. Essentially it’s a tabular setting. Given the RL theory these days mostly moved away from tabular setting and at the very minimum considers linear MDP setting (Jin et al 2020), the paper seems to lack generality beyond this finite setting.

The paper is very dense in notation and it’s very difficult to follow. I must add as a disclaimer that, I am not familiar with many of the related literature around this paper. However, even after coming from RL theory background, I found it tiring and difficult to follow the notations and setups.

The significance of the theoretical result is not clear. I think the paper would benefit more if the significance and importance are highlighted more.

**Questions:**

What would be the main difficulty in setting this problem up in non-tabular setting?

---

> ### Author Response · Authors · 2024-11-18
> **Response to Reviewer kSFM**
>
> We thank the reviewer for their constructive review. We have provided detailed responses to all the questions raised. In addition, we have simplified some notation, introduced a notation table for the essential variables in the Appendix, and added some additional explanations for the algorithms. We hope the reviewer can examine them and re-evaluate our paper. We look forward to addressing any further questions in the author-reviewer discussion period.
>
> >Given the RL theory these days mostly moved away from tabular setting and at the very minimum considers linear MDP setting, the paper seems to lack generality beyond this finite setting.
>
> While there have been tremendous steps in moving RL away from the tabular setting, our subsampling algorithm and analytic methods were intended to solve a different combinatorial problem of the curse of multiagency. As the reviewer points out, it is widely recognized that the issues of each agent having an infinitely large state/action space can be tackled using linear function approximation (FA) (see references [1,2]), but we emphasize that our paper tackles a _different_ sample complexity problem that generally emerges in RL (see references [3,4,5]): so, the two techniques can be combined to extend the paper beyond the finite tabular setting. As stated in discussion 3.7, the value iteration in our algorithm can be replaced with any arbitrary value-based RL algorithm, such as deep Q-networks where the price we pay is the loss of accuracy in estimating $\hat{Q}_k^*$, which appears as an additive error term in our bound for the optimality gap (which decays with the number of samples).
>
> Therefore, our subsampled $\hat{Q}_k^*$-function is amenable for learning under the linear function approximation (LFA) methodology that has been widely studied in the literature (as we stated in the last line of our submission). To elaborate, our subsampled $\hat{Q}_k$ function is an approximation of the ground truth $Q^*$ function. Therefore, with LFA to approximate $\hat{Q}_k$, the triangle inequality (combining our bound for subsampling with the existing bounds for approximating the $\hat{Q}_k^*$ function in the literature) gives us an immediate bound that readily extends our subsampling framework to the infinite non-tabular setting. To provide a concrete extension to the non-tabular setting, we have elaborated on the remarks in discussion 3.7, and introduced section E in the appendix which formalizes this intuition and derives a formal bound on the optimality gap in the extended non-tabular setting.
>
> >The significance of the theoretical result is not clear. I think the paper would benefit more if the significance and importance are highlighted more.
>
> As in the contributions section and discussion 3.6, we have included some statements regarding the implications of the theoretical results. We have now revised the discussion and provided a further exposition of the theoretical results in the updated manuscript. We summarize the statements below:
>
> -  When we set $k=O(\log n)$, we get an exponential speedup on the complexity from mean-field value iteration, from $\mathrm{poly}(n)$ to $\mathrm{poly}(\mathrm{log}(n))$, and a super-exponential speedup from traditional value-iteration from $\exp(n)$ to $\mathrm{poly}(\mathrm{log}(n))$, where the optimality gap $O(1/\sqrt{\log n})$ goes to $0$ as $n$ becomes large
> - This yields a learning algorithm with a polylogarithmic runtime, where the performance of the learned policy approaches the performance of the optimal policy as $k\to n$. In fact, the error bound on our optimality gap is _stronger_. The optimality gap of $O(1/\sqrt{k})$ decays to $0$, _regardless_ of the value of $n$.
> - When $|\mathcal{S}_l|\gg n$, our subsampling algorithm again reduces the complexity of value iteration in the tabular setup from $O(|\mathcal{S}_l|^n)$ to $O(|\mathcal{S}_l|^k)$, giving an exponential speedup.
> - Finally, as indicated in our numerical simulations, we can allow $k$ to be _much_ smaller than $n$, at a much smaller cost to still learn an approximately optimal policy. Given the success of subsampling in this setting, we believe sampling (and mean-field sampling) could be a potentially useful tool for networked multi-agent RL.
>
> References:
>
> [1] Linear Bellman Completeness Suffices for Efficient Online Reinforcement Learning with Few Actions (Golowich, Moitra, PMLR
> 2024)
>
> [2] Breaking the Curse of Multiagency: Provably Efficient Decentralized Multi-Agent RL with Function Approximation (Wang et. al., PMLR 2024)
>
> [3] Provable Representation with Efficient Planning for Partially Observable Reinforcement Learning (Zhang et. al., ICML 2024)
>
> [4] Near-Optimal Sample Complexity Bounds for Constrained MDPs (Vaswani et. al., NeurIPS, 2022)
>
> [5] V-Learning—A Simple, Efficient, Decentralized Algorithm for Multiagent RL (Jin et. al., ICLR 2022).

---

> > ### Comment · Reviewer_kSFM · 2024-11-26
> >
> > I thank the authors for making the contribution and discussion section clearer. Regarding tabular setting vs linear MDP setting,  I think the paper would benefit from adding such discussion.
> >
> > I increased the soundness score after reading the response. In my original review, I already provided a confidence score of 2 to indicate that I did not understand many parts of the submission as I am not familiar with some of the key related works. I hope the AC would keep that into consideration while making the final decision.

---

> > > ### Author Response · Authors · 2024-11-26
> > > **Reply to Reviewer kSFM**
> > >
> > > We thank the reviewer for their valuable feedback.
> > >
> > > We have updated the manuscript to include a detailed discussion of the tabular vs linear MDP setting. Specifically, we draw the reviewer's attention to Lines 378-381 (page 8) and Lines 1814-1875 (Appendix E, starting from page 34).
> > >
> > > In these changes, we have additionally considered the Linear MDP setting (actually, a stronger version of Linear Bellman completeness), and derived a theoretical performance guarantee in the setting where $\mathcal{S}_g$ is infinitely large, which we hope addresses the reviewer's concerns about extensions to the non-tabular setting.
> > >
> > > Additionally, we have revised the notation and wording in many other areas throughout the document to improve the clarity of the exposition. We are happy to provide further clarifications if you have any questions.

---

> ### Author Response · Authors · 2024-11-24
> **Response to Reviewer kSFM**
>
> We sincerely thank the reviewer for their constructive suggestions and comments. As the deadline approaches, we hope the reviewer will have an opportunity to review our response. If there are any further comments or additional concerns, we would be grateful to address them. Please do not hesitate to let us know if any further clarifications or discussions are needed to assist in the evaluation.

---

### Author Response · Authors · 2024-11-18
**Changes in the new version of the paper**

Dear reviewers and ACs,

We extend our deep thanks for the time you devoted to reviewing our manuscript and for the insightful comments and feedback provided. We are happy to hear that the reviewers
- Found our reinforcement learning setting to be interesting and novel in the context of modern applications,
- Found our theoretical bound (namely, the optimality gap of $O(1/\sqrt{k})$ in our subsampling framework) and the faster time complexity of the algorithm to be of interest, and
- Appreciated the numerical simulations to validate the theoretical results.

The reviewers identified a few weaknesses in our paper:
1) Reviewers kSFM and KC7o raised issues of clarity in our presentation and notation,
2) Reviewer kSFM raised a point on how our result could generalize to the non-tabular setting, and
3) Reviewer KC70 raised an issue that in some parameter regimes, our algorithm may not provide an actual speedup.

In response to these recommendations, we have implemented the following major modifications:

- Per reviewer KSFM and KC7o’s comments, we added a notation table in the Appendix and removed verbiage from the main body, making the notation less dense. Furthermore, we added a more detailed explanation of the purpose of Algorithm 3 (which serves to convert the optimality of the global agent’s action on the subsampled $k$-local-agent system to an approximate optimality on the full system),
- Per reviewer kSFM’s comment, we added Appendix E which extends our work to the non-tabular setting, where the global agent’s and state space can be an infinite (but compact) set, under a general assumption of Linear Bellman completeness, and
- Per reviewer KC70’s comment, we added Algorithm 1 which is a simple reparameterization of Algorithm 2, but is more efficient in the case where each local agent’s state space is large (relative to the number of agents).

We are very grateful for this constructive criticism. They have been crucial in enhancing the quality of the paper. We hope that our revision communicates our work better and addresses your points. Should you have any remaining queries, please do raise them. We are more than willing to provide further clarifications. We deeply value your support and are thankful for your role in furthering research in our field.


Warm regards,

Authors

---

### Author Response · Authors · 2024-11-25
**Official Comment by Authors**

We sincerely thank the reviewer for their constructive suggestions and comments. As the deadline approaches, we hope the reviewer will have an opportunity to review our response. If there are any further comments or additional concerns, we would be grateful to address them. Please do not hesitate to let us know if any further clarifications or discussions are needed to assist in the evaluation.

---

### Public Comment · ~Sai_Advaith_Maddipatla1 · 2024-11-28

Pros:

- The problem for the global agent appears interesting and is closely related to some other practical control-theory scenarios such as general resource allocation and cooperative path planning.
- The algorithm seems to be especially useful if the global agent does not get complete observability during the execution.

Questions:

- Algorithm 1 works with a fixed sample of local agents (i.e., same agents used in every value iteration), while Algorithm 2 draws a new random sample of local agents for each step. What is the intuition behind this difference, i.e., fixed sample in one case but drawing new samples in the other?
- Does this algorithm extend beyond Q-learning. For example, does this extend to TD learning?
- It is mentioned multiple times in the manuscript that the policy converges to the optimal policy as m \to \infty and k \to n. I'm curious if there is any relevant bound when only one of the previous conditions holds? For example, if only k \to n, then the setting is a sample-deficient estimation of the global agent's Q-function, but if only m \to \infty, is there any relevant idea of what may happen?

---

> ### Author Response · Authors · 2024-11-29
> **Reply to Public Comment**
>
> Thanks for your comments!
>
> >Algorithm 1 works with a fixed sample of local agents (i.e., same agents used in every value iteration), while Algorithm 2 draws a new random sample of local agents for each step. What is the intuition behind this difference, i.e., fixed sample in one case but drawing new samples in the other?
>
> Thanks for this question! The intuition for this difference can be split into two parts. Firstly, for the value iteration, a key observation is that the $\hat{Q}_k$-function converges to a fixed-point, irrespective of the actual choice of the $k$ agents. This allows us to fix a sample for the first case. For the policy execution, the purpose is that we want to use a sample of $k$ agents to be representative of all agents, and for this purpose, we do need to draw fresh samples at each time such that on expectation, we have a good representation of all agents. This is inspired by the widely popular _power-of-2-choices_ from the queueing theory literature, where two queues are sampled and a job is assigned to one of them. Intuitively, our algorithm generalizes this notion, where we sample from different agents at each step (as opposed to fixing the choice of $k$ agents at the start) to provide signals to different agents. To the best of our knowledge, this is the first multi-agent RL algorithm that utilizes such a sub-sampling method to estimate the ${Q}^*$-function.
>
>
> >Does this algorithm extend beyond $Q$-learning. For example, does this extend to TD learning?
>
> In this paper, we have proposed an algorithm that does $Q$-learning for the $k$ subsampled agents. We can replace the $Q$-learning algorithm with any arbitrary value-based RL method that learns the $Q$-function (which could include function approximation, and, indeed, TD learning). The price we pay is the term in our final bound which will depend on the approximation error of the underlying method. We have a remark in Discussion 3.7 which addresses this point. Appendix E in the manuscript computes this approximation error for the case where function approximation is used to learn the $Q$-function.
>
> >It is mentioned multiple times in the manuscript that the policy converges to the optimal policy as $m \to \infty$ and $k \to n$. I'm curious if there is any relevant bound when only one of the previous conditions holds? For example, if only $k \to n$, then the setting is a sample-deficient estimation of the global agent's Q-function, but if only $m \to \infty$, is there any relevant idea of what may happen?
>
> This is an interesting question. If $k\to n$ (independent of $m$), then the setting is indeed a sample-deficient estimation of the global agent’s $Q$-function. From our final bound, we know that the value function of the derived policy will still approach the value function of the optimal policy. On the other hand, if $m\to\infty$ (independent of $k$), then the Bellman update in each iteration becomes less stochastic (though this depends on the stochasticity of the transition functions).
>
> Specifically, if the transition functions are completely deterministic: increasing $m$ has no effect. If the transition functions are highly random: increasing $m$ ensures that the empirical average of samples in the Bellman updates corresponds to its expected value (by the law of large numbers). This would cause $\hat{Q}_{k,m}^*$ to approach $\hat{Q}_k^*$. Therefore, the optimality gap of the learned policy $\tilde{O}(1/\sqrt{k})$ only corresponds to the gap incurred by the sub-sampling mechanism.

---

### Meta-Review · Area_Chair_PVcn · 2024-12-21

**Metareview:**

The paper examines global decision-making in the presence of local agents by leveraging subsampled agents, and provide convergence guarantees as a function of the number of subsampled agents as well as numerical experiments.

The paper consider a fairly novel and interesting decision making setting. However, it is limited by woking in tabular MDPs.
Perhaps the main complaint, which is shared among all the reviewers, is that the paper is quite difficult to follow in terms of concepts and notations.
The presentation of this work is judged to be improvable as several aspects remains unclear.

**Additional Comments On Reviewer Discussion:**

The reviewers recommend improving the presentation of paper, as several aspects remain unclear

---

### Decision · Program_Chairs · 2025-01-22

Reject